# GeoSplat: A Deep Dive into Geometry-Constrained Gaussian Splatting

## Abstract

A few recent works explored incorporating geometric priors to regularize the optimization of Gaussian splatting, further improving its performance. However, those early studies mainly focused on the use of low-order geometric priors (e.g., normal vector), and they might also be unreliably estimated by noise-sensitive methods, like local principal component analysis. To address their limitations, we first present *GeoSplat*, a general geometry-constrained optimization framework that exploits both first-order and second-order geometric quantities to improve the entire training pipeline of Gaussian splatting, including Gaussian initialization, gradient update, and densification. As an example, we initialize the scales of 3D Gaussian primitives in terms of principal curvatures, leading to a better coverage of the object surface than random initialization. Secondly, based on certain geometric structures (e.g., local manifold), we introduce efficient and noise-robust estimation methods that provide dynamic geometric priors for our framework. We conduct extensive experiments on multiple datasets for novel view synthesis, showing that our framework: *GeoSplat*, significantly improves the performance of Gaussian splatting and outperforms previous baselines.

## 1 Introduction

The photorealistic rendering quality and high efficiency of Gaussian splatting (Kerbl et al., 2023) have fueled a surge of recent studies, which further investigate it from various perspectives, such as memory consumption (Niedermayr et al., 2024) and texture (Gao et al., 2024).

**Limitations in recent geometric regularization.** A notable perspective from those studies is to treat geometric priors as a form of regularization for improving Gaussian splatting, given that the Gaussian primitives should approximate the surfaces of 3D objects. For example, Li et al. (2025) places new primitives in the tangent planes of original primitives, with the aim to reduce outlier artifacts. While obtaining performance gains, previous methods mainly focus on low-order geometric quantities (e.g., normal vector) and do not much consider higher-order ones (i.e., curvature). The two types of geometric information characterize very different aspects of 2D object surfaces (Lee, 2018), so the absence of either one might limit their potential.

Another key limitation of prior works is that they sometimes estimate geometric information using methods that lack robustness. For example, local principal component analysis (PCA) (Kambhatla & Leen, 1997; Li et al., 2025) is highly sensitive to noise, and the trained StableNormal model (Ye et al., 2024; Wang et al., 2024) often fails on rarely seen data. The geometric priors in previous methods are also static, which would gradually become inaccurate in characterizing the dynamic geometry of Gaussian primitives during optimization.

**Our geometric optimization framework: GeoSplat.** To address the limitations of prior works, we first propose a general geometry-constrained optimization framework, *GeoSplat*, exploiting both low-order and higher-order geometric priors to regularize the training pipeline of Gaussian splatting. While being largely ignored by previous methods, the higher-order information (e.g., curvature) characterizes a critical property of 2D surface: how it bends in the 3D space. In light of this, we adopt such information to regularize the shapes of Gaussian primitives. For instance, an area with low curvature indicates that it approximates a flat plane, so we can initialize the scales of Gaussian

primitives in this area as a large number. Low-order geometric quantities, such as tangent and normal vectors, have already played key roles in recent works, and we further extend their application scopes in our framework. For example, to reduce floating artifacts, we truncate the gradient update of a Gaussian primitive in terms of its normal direction.

Secondly, we present an estimation method that can provide noise-robust geometric information for our framework. Specifically, we assume that each Gaussian primitive resides on an underlying surface that is locally a manifold (Lee, 2018), and we derive the analytical form of the shape operator, which contains the curvature information. We also investigate another approach, based on the notion of varifold from geometric measure theory (Allard, 1972; Simon, 1983; Menne, 2017), which in some cases outperforms our manifold-based method. Both methods are highly efficient, and thus can provide our framework with dynamic geometric priors during training.

Extensive experiments have been conducted on multiple benchmark datasets in novel view synthesis. The results demonstrate that our framework, *GeoSplat*, significantly improves the performance of Gaussian splatting and outperforms previous baselines.

## 2 Preliminaries

In this section, we mainly revisit the basics of Gaussian splatting and formulate our problem setup, establishing necessary notations. We also review basic geometry in Appendix A to explain some geometric terminologies (e.g., tangent space) used in the main text.

### 2.1 Gaussian Splatting

The core of Gaussian splatting is to represent a 3D scene as a number of Gaussian primitives $\{\mathcal{G}_{\mathrm{obj}}^i\}_{i \in \mathcal{I}}$, with each primitive parameterized by its opacity $\alpha_{\mathrm{obj}}^i \in \mathbb{R}$, RGB color $\mathbf{c}_{\mathrm{obj}}^i \in \mathbb{R}^3$, mean vector $\boldsymbol{\mu}_{\mathrm{obj}}^i \in \mathbb{R}^3$, and covariance matrix $\boldsymbol{\Sigma}_{\mathrm{obj}}^i \in \mathbb{R}^{3 \times 3}$. The matrix $\boldsymbol{\Sigma}_{\mathrm{obj}}^i$ is semipositive definite and can be decomposed as $\mathbf{R}_{\mathrm{obj}}^i \mathbf{S}_{\mathrm{obj}}^i (\mathbf{R}_{\mathrm{obj}}^i \mathbf{S}_{\mathrm{obj}}^i)^\top$, where $\mathbf{S}_{\mathrm{obj}}^i = \mathrm{diag}([s_1^i, s_2^i, s_3^i]^\top)$ is a diagonal scale matrix and $\mathbf{R}_{\mathrm{obj}}^i = [\mathbf{r}_1^i; \mathbf{r}_2^i; \mathbf{r}_3^i] \in \mathrm{SO}(3)$ is a rotation matrix.

A pixel at some image coordinate $\mathbf{p} \in \mathbb{R} \times \mathbb{R}$ is synthesized by splatting and blending Gaussian primitives. Specifically, the splatting operation first casts the 3D Gaussian primitive $\mathcal{N}(\boldsymbol{\mu}_{\mathrm{obj}}^i, \boldsymbol{\Sigma}_{\mathrm{obj}}^i)$ in the word coordinate to a 2D primitive $\mathcal{N}(\boldsymbol{\mu}_{\mathrm{plane}}^i, \boldsymbol{\Sigma}_{\mathrm{plane}}^i)$ in the image plane:

$$\boldsymbol{\mu}_{\mathrm{plane}}^i = \Pi(\mathbf{T}_{\mathrm{pose}} \boldsymbol{\mu}_{\mathrm{obj}}^i), \quad \boldsymbol{\Sigma}_{\mathrm{plane}}^i = \mathbf{J} \mathbf{W}_{\mathrm{pose}} \boldsymbol{\Sigma}_{\mathrm{obj}}^i (\mathbf{J} \mathbf{W}_{\mathrm{pose}})^\top, \tag{1}$$

where $\mathbf{T}_{\mathrm{pose}} \in \mathrm{SE}(3)$ is the camera pose consisting of rotation matrix $\mathbf{W}_{\mathrm{pose}}$ and translation vector $\mathbf{b}_{\mathrm{pose}}$, and $\Pi$ is the 3D to 2D coordinate projection, with $\mathbf{J}$ its Jacobian matrix. Then, the blending operation renders the color of pixel $\mathbf{p}$ as

$$\mathbf{c}_{\mathrm{img}}(\mathbf{p}) = \sum_i \mathbf{c}_{\mathrm{obj}}^i \beta^i(\mathbf{p}) \prod_{j < i} (1 - \beta^j(\mathbf{p})), \tag{2}$$

where $\beta^k(\mathbf{p}) := \alpha_{\mathrm{obj}}^k \mathcal{N}(\mathbf{p}; \boldsymbol{\mu}_{\mathrm{plane}}^k, \boldsymbol{\Sigma}_{\mathrm{plane}}^k)$ quantifies the radiance of a Gaussian primitive. Under this scheme, the optimization objective is to match rendered images with real images.

### 2.2 Problem Formulation

Our primary goal is to leverage diverse geometric priors to improve Gaussian splatting optimization, especially higher-order ones (e.g., curvature). Meanwhile, we also aim to estimate the priors in a noise-robust manner. These two aspects are largely neglected in recent explorations (Wang et al., 2024; Li et al., 2025) on geometric regularization.

**Manifold-based setup.** Manifold (Lee, 2018) is a popular geometric language for characterizing spatial objects. We provide an overview on it in Appendix A.1, mentioning a strict assumption that the objects need to be everywhere smooth. It is not hard to find real 3D objects that violate this assumption. For example, an apple has a singular point at the intersection between its stem and body. To generalize this geometric concept, we consider a weaker setting.

**Definition 2.1** (Merged Local Manifolds). Gaussian primitives $\{\mathcal{G}^i_{\mathrm{obj}}\}_{i \in \mathcal{I}}$ reside in an underlying 3D scene formed by finitely many continuous manifold $\{\mathcal{M}_j\}_{j \in \mathcal{J}}$. Notably, each manifold $\mathcal{M}_j$ is smooth up to a zero-measure set of singular points.

Compared with vanilla manifold, this new setting can accommodate typical 3D scenes, where there are multiple independent spatial objects that might not be fully smooth. The zero-measure condition is also important, permitting almost every point $\mathbf{q} \in \mathbb{R}^3$ in the scene to locate at a local region that can be treated as a smooth manifold. For novel view synthesis, the surface $\mathcal{M}_j$ is normally 2D (we denote its dimension as $D_{\mathcal{M}_j} = 2$), embedded in the 3D ambient space $\mathcal{E} = \mathbb{R}^3$ ($D_{\mathcal{E}} = 3$).

Most geometric quantities are locally defined, and thus fit our setting. Specifically, every Gaussian primitive $\mathcal{G}^i_{\mathrm{obj}}, i \in \mathcal{I}$ will lie on a certain underlying object surface $\mathcal{M}_j, j \in \mathcal{J}$ at position $\mathbf{q} = \boldsymbol{\mu}^i_{\mathrm{obj}}$, with tangent space $\mathcal{T}_{\mathbf{q}}\mathcal{M}_j$ spanned by orthonormal basis $\{\mathbf{u}^i_d\}_{1 \leqslant d \leqslant D_{\mathcal{M}_j}}$ and normal space $\mathcal{N}_{\mathbf{q}}\mathcal{M}_j$ formed by another basis $\{\widetilde{\mathbf{u}}^i_d\}_{D_{\mathcal{M}_j}+1 \leqslant d \leqslant D_{\mathcal{E}}}$. These vector bases represent first-order geometric quantities, and we also aim to consider higher-order ones, e.g. principal curvatures and directions $\{(\tau^i_d, \widetilde{\mathbf{w}}^i_d)\}_{1 \leqslant d \leqslant D_{\mathcal{M}_j}}$, with each pair indicating that the underlying surface is bent with a magnitude of $\tau^i_d$ along direction $\widetilde{\mathbf{w}}^i_d$. For notational convenience, we will omit the subscript $j$ of local manifold $\mathsf{M}_j$, since it is unique for each primitive $\mathcal{G}^i_{\mathrm{obj}}$.

**Another geometric tool: varifolds.** We also explore adopting another tool from geometric measure theory, the notion of varifold (Allard, 1972; Simon, 1983; Menne, 2017), which is free from the smoothness assumption of manifolds. Generally speaking, a varifold is a nonnegative Radon measure (Rudin, 1987) on the product space $\mathcal{E} \times \mathsf{G}_{D_{\mathsf{M}}, D_{\mathcal{E}}}$ where the second space denotes the Grassmannian of $D_{\mathsf{M}}$-dimensional vector subspaces of $\mathcal{E}$ (Lee, 2018). Two classes of varifolds useful in our context are rectifiable varifolds and point cloud varifolds. A rectifiable varifold has the form $\mathcal{W} = \theta \mathcal{H}^{D_{\mathsf{M}}}_{|\mathsf{M}} \otimes \delta_{\mathcal{T}_x \mathsf{M}}$ where $\mathcal{H}^{D_{\mathsf{M}}}_{|\mathsf{M}}$ is the $D_{\mathsf{M}}$-dimensional Hausdorff measure restricted to a countably rectifiable set $\mathsf{M}$ (Federer, 1969; Simon, 1983), $\delta_{\mathcal{T}_x \mathsf{M}}$ is the Dirac measure in $\mathsf{G}_{D_{\mathsf{M}}, D_{\mathcal{E}}}$ supported on the tangent plane at $x \in \mathsf{M}$, and $\theta : \mathsf{M} \to \mathbb{R}$ is a nonnegative multiplicity function with $\theta > 0$ $\mathcal{H}^{D_{\mathsf{M}}}$-a.e. on $\mathsf{M}$. A point cloud $d$-varifold has the form $\mathcal{W} = \sum_{1 \leqslant j \leqslant N} m_j \delta_{x_j} \otimes \delta_{P_j}$ where $\{x_j\}_{j=1}^N$ is a finite set of points associated with $d$-planes $\{P_j\}_{j=1}^N$ and nonnegative masses $\{m_j\}_{j=1}^N$.

The notion of varifold is so flexible that it can be associated with the discrete locations of Gaussian primitives $\{\mathcal{G}^i_{\mathrm{obj}}\}_{i \in \mathcal{I}}$ without supposing there is any smooth interpolation. More importantly, geometric quantities valid for manifolds can be weakly or approximately defined for varifolds, such as the approximate tangent space or the (approximate) generalized second fundamental form and mean curvature. Due to the limited space of main text, we provide more details in Appendix A.2, with a concise review of varifold theory.

## 3 METHOD: GEOMETRY-CONSTRAINED GAUSSIAN SPLATTING

In this section, we first present *GeoSplat*, a framework of geometry-constrained Gaussian splatting, which exploits higher-order geometric priors that are largely neglected by previous works. Then, we present noise-robust methods to estimate dynamic geometric priors. In contrast, previous methods obtained static priors through noise-sensitive or domain-specific techniques.

### 3.1 GEOMETRY-CONSTRAINED OPTIMIZATION

Training a Gaussian splatting model involves initialization, optimization, and densification. In the following, we show how each of these components can be geometrically improved.

#### 3.1.1 CURVATURE-GUIDED GAUSSIAN INITIALIZATION

**Curvature-guided covariance warm-up.** In common practices, most attributes of a Gaussian primitive $\mathcal{G}^i_{\mathrm{obj}}$ (e.g., the position vector $\boldsymbol{\mu}^i_{\mathrm{obj}}$) are properly initialized before training to ensure stable optimization and high performance. However, this is not the case for covariance matrix $\boldsymbol{\Sigma}^i_{\mathrm{obj}}$, with its rotation part $\mathbf{R}^i_{\mathrm{obj}}$ arbitrarily initialized and scale part simply set to be isotropic: $\mathbf{S}^i_{\mathrm{obj}} = (s^i_{\mathrm{nbr}}/2)\mathbf{I}$. Here $s^i_{\mathrm{nbr}}$ denotes the average distance to a few neighbors.

We assume that the primitive $\mathcal{G}_{\mathrm{obj}}^i$ locates at an underlying surface $\mathcal{M}^1$, so it is intuitive to warm up its initial shape $\boldsymbol{\Sigma}_{\mathrm{obj}}^i$ in terms of geometric priors. To begin with, we set one rotation direction as the normal vector to $\mathcal{M}$, with the corresponding scale specified as a tiny value:

$$\mathbf{r}_3^i = \widetilde{\mathbf{u}}_3^i, \quad s_3^i = \xi_{\min} \ll 1. \tag{3}$$

In this manner, Gaussian primitive $\mathcal{G}_{\mathrm{obj}}^i$ will get largely squashed along the normal direction $\widetilde{\mathbf{u}}_3^i$, and thus tightly fit the 2D surface: $\{\mathbf{r}_1^i, \mathbf{r}_2^i\} \subset \mathcal{T}_{\mathbf{q}}\mathcal{M}$.

Secondly and more importantly, other rotation directions $\mathbf{r}_1^i, \mathbf{r}_2^i$ are the eigenvectors of covariance matrix $\boldsymbol{\Sigma}_{\mathrm{obj}}^i$, so they correspond to the tangent directions that lead to the maximum and minimum variances $(s_1^i)^2, (s_2^i)^2$ (Anderson et al., 1958). In this regard, we respectively configure them as principal directions, with a curvature-based scale constraint:

$$\mathbf{r}_d^i = \widetilde{\mathbf{w}}_{3-d}^i, d \in \{1, 2\}, \quad s_1^i / s_2^i = |\tau_2^i|^{-1} / |\tau_1^i|^{-1}. \tag{4}$$

In this expression, the maximum standard deviation $s_1^i$ (or minimum one $s_2^i$) in statistics is linked with the low curvature $\tau_2^i$ (or high one $\tau_1^i$) in geometry. The core idea is that the points sampled on a flat curve (along the low curvature direction $\mathbf{w}_2^i$) tend to be less concentrated (in the high-variance direction $\mathbf{r}_1^i$) than a twisted one, given the same lengths.

The two scale entries $s_1^i, s_2^i$ are underdetermined with respect to a single constraint, so we impose an extra area-invariant condition to finally solve them.

**Proposition 3.1** (Curvature-constrained Primitive Scales). *If the curvature-based constraint (i.e., Eq. (4)) and a new constraint: $s_1^i s_2^i = (s_{\mathrm{nbr}}^i)^2 / 4$, both hold, then the scale variables can be solved as $s_1^i = \frac{s_{\mathrm{nbr}}^i}{2}\sqrt{|\tau_1^i / \tau_2^i|}, s_2^i = \frac{s_{\mathrm{nbr}}^i}{2}\sqrt{|\tau_2^i / \tau_1^i|}$. Notably, this solution ensures that the one-sigma region of projected Gaussian primitive in the tangent plane $\mathcal{T}_{\mathbf{q}}\mathcal{M}$ are area-invariant.*

The significance of area invariance is to ensure that the primitive $\mathcal{G}_{\mathrm{obj}}^i$ still has a sufficient coverage over the surface $\mathcal{M}$ after re-scaling. We put the proof for this conclusion in Appendix C. To summarize, the covariance matrix $\boldsymbol{\Sigma}_{\mathrm{obj}}^i$ of Gaussian primitive $\mathcal{G}_{\mathrm{obj}}^i$ can be fully warmed up in terms of normal and curvature priors, with its components as

$$\mathbf{R}_{\mathrm{obj}}^i = [\widetilde{\mathbf{w}}_2^i; \widetilde{\mathbf{w}}_1^i; \widetilde{\mathbf{u}}_3^i], \quad \mathbf{S}_{\mathrm{obj}}^i = \mathrm{diag}\left(\left[\frac{s_{\mathrm{nbr}}^i}{2}\sqrt{|\tau_1^i / \tau_2^i|}, \frac{s_{\mathrm{nbr}}^i}{2}\sqrt{|\tau_2^i / \tau_1^i|}, \xi_{\min}\right]^\top\right). \tag{5}$$

This solution assumes non-zero curvatures: $\tau_d^i \neq 0, d \in \{1, 2\}$, but we can fix the zero case by clamping it as $\min(\max(\tau_d^i, \xi_{\min}), \xi_{\max})$, where $\xi_{\max}$ is a predefined large value.

**Curvature-guided primitive upsampling.** The initial set of Gaussian primitives $\{\mathcal{G}_{\mathrm{obj}}^i\}_{i \in \mathcal{I}}$ are not large enough, so Ververas et al. (2025) populated the low-curvature areas that had few primitives before training. The mean curvature was used to identify such areas, though it suffered from an inaccurate approximation using low-order quantities.

Our noise-robust methods (Sec. 3.2) provide full curvature information $\{\tau_d^i\}_{1 \leqslant d \leqslant D_{\mathcal{M}}}$, so we are able to directly compute the mean curvature as $(\tau_1^i + \tau_2^i)/2$, without any approximation. It is not proper to identify the low-curvature regions via mean curvature. A counter-example is the helical surface, which has a zero mean curvature but is curved (i.e., $\prod_d \tau_d^i \neq 0$). In this regard, we adopt a more reasonable quantity: mean absolute curvature (MAC), as $\bar{\tau}^i = (\sum_d |\tau_d^i|)/D_{\mathcal{M}}$.

If MAC is quite small, i.e. $\bar{\tau}^i < \xi_{\min}$, which means the curvature is basically negligible, then we can identify that the corresponding Gaussian primitive $\mathcal{G}_{\mathrm{obj}}^i$ resides in an almost flat region. In this case, we upsample its position through midpoint interpolation:

$$\boldsymbol{\mu}_{\mathrm{obj}}^{i'} = (\boldsymbol{\mu}_{\mathrm{obj}}^i + \boldsymbol{\mu}_{\mathrm{obj}}^\star)/2, \star \in \mathrm{KNN}(i), \tag{6}$$

which also applies to other attributes (e.g., color). Here KNN denotes the nearest neighbors in index set $\mathcal{I}$, and $i'$ is a new index for the set. Due to the flatness, this linear interpolation will locate the new primitive $\mathcal{G}_{\mathrm{obj}}^{i'}$ near the underlying surface $\mathcal{M}$.

---

[1]With abuse of notation, we mostly denote the 2D surface as $\mathcal{M}$, potentially assuming it is a local manifold. However, it should be considered as a rectifiable set M in the context of rectifiable varifolds.

### 3.1.2 SHAPE-CONSTRAINED OPTIMIZATION

**Truncated gradient update.** The rendering quality of Gaussian splatting is largely affected by floating artifacts (Ungermann et al., 2024; Turkulainen et al., 2025). From a geometric point of view, we can define a Gaussian primitive $\mathcal{G}_{\mathrm{obj}}^i$ as an outlier if it significantly deviates from the underlying surface $\mathcal{M}$. In this sense, we propose to reduce the outliers through truncating the gradient update with respect to the normal direction $\tilde{\mathbf{u}}_3^i$ as

$$\boldsymbol{\mu}_{\mathrm{obj}}^i \leftarrow \boldsymbol{\mu}_{\mathrm{obj}}^i - \omega\big((\nabla^i \mathfrak{L})^\top + \min\big(\xi_{\min}/\|(\nabla^i \mathfrak{L})^\perp\|, 1\big)(\nabla^i \mathfrak{L})^\perp\big), \tag{7}$$

where $\omega$ is the learning rate, $\nabla^i \mathfrak{L}$ denotes the gradient of loss function $\mathfrak{L}$ regarding position $\boldsymbol{\mu}_{\mathrm{obj}}^i$, $(\cdot)^\top$ denotes the orthogonal projection onto the tangent plane $\mathcal{T}_\mathbf{q} \mathcal{M}$, and $(\cdot)^\perp$ represents the projection onto the normal direction $\tilde{\mathbf{u}}_3^i$. With this truncation, a Gaussian primitive $\mathcal{G}_{\mathrm{obj}}^i$ that lies on the underlying surface $\mathcal{M}$ will still stay close to it after an aggressive gradient update $\nabla^i \mathfrak{L}$.

**Shape regularization.** Another type of artifact that might degrade the performance of Gaussian splatting is the Gaussian primitive $\mathcal{G}_{\mathrm{obj}}^i$ with a needle-like shape (Hyung et al., 2024): covariance matrix $\boldsymbol{\Sigma}_{\mathrm{obj}}^i$ degenerates to be rank one. From a geometric perspective, the main scales $s_1^i, s_2^i$ of the primitive $\mathcal{G}_{\mathrm{obj}}^i$ must be regularized to ensure a rank two covariance matrix $\boldsymbol{\Sigma}_{\mathrm{obj}}^i$, so that it will have a sufficient coverage over the underlying manifold $\mathcal{M}$.

Based on the above insight and our prior discussion (i.e., Eq. 4), we present a hinge-like regularization that will impose a penalty if the minor scale $s_2^i$ is too small:

$$\mathcal{L}_{\mathrm{scale}} = \max\big(0, s_1^i/s_2^i - |\tau_1^i/\tau_2^i| - \xi_{\min}\big) + \big(s_3^i\big)^2, \tag{8}$$

with the last term to ensure the rank of matrix $\boldsymbol{\Sigma}_{\mathrm{obj}}^i$ is smaller than 3, hence the Gaussian primitive $\mathcal{G}_{\mathrm{obj}}^i$ is closely attached to the surface $\mathcal{M}$. For the hinge term, we again regard the curvature information $\{\tau_d^1, \tau_d^2\}$ as natural guidance for shaping the scale matrix $\mathbf{S}_{\mathrm{obj}}^i$. In a similar spirit, we introduce another regularization for the rotation matrix $\mathbf{R}_{\mathrm{obj}}^i$ as

$$\mathcal{L}_{\mathrm{rot}} = \big(1 - \langle \mathbf{r}_1^i, \tilde{\mathbf{w}}_2^i \rangle\big)^2 + \big(1 - \langle \mathbf{r}_2^i, \tilde{\mathbf{w}}_1^i \rangle\big)^2 + \big(1 - \langle \mathbf{r}_3^i, \tilde{\mathbf{u}}_3^i \rangle\big)^2, \tag{9}$$

aligning its components with the principal directions $\{\tilde{\mathbf{w}}_d^i\}_{1 \leqslant d \leqslant D_\mathcal{M}}$.

### 3.1.3 CURVATURE-REGULARIZED PRIMITIVE DENSIFICATION

Densification operations (i.e., split and clone) are indispensable in Gaussian splatting for rendering high-texture regions, though they might generate floating artifacts (Li et al., 2025). From a geometric viewpoint, that problem is caused by an inappropriate placement of new Gaussian primitive $\mathcal{G}_{\mathrm{obj}}^{i'}$, $i' \notin \mathcal{I}$, making it an outlier for the underlying surface $\mathcal{M}$.

To address this problem, we first restrict the split operation to produce the new primitive $\mathcal{G}_{\mathrm{obj}}^{i'}$ close to the surface $\mathcal{M}$. Specifically, we interpolate the principal directions $\{\tilde{\mathbf{w}}_d^i\}_{1 \leqslant d \leqslant D_\mathcal{M}}$ with a random vector $\boldsymbol{\rho} = [\rho_1, \rho_2, \rho_3]^\top$ sampled from standard normal $\mathcal{N}(\mathbf{0}, \mathbf{I})$ as

$$\boldsymbol{\mu}_{\mathrm{obj}}^{i'} = \boldsymbol{\mu}_{\mathrm{obj}}^i + (\rho_2/\tau_2^i)\tilde{\mathbf{w}}_2^i + (\rho_1/\tau_1^i)\tilde{\mathbf{w}}_1^i + \rho_3 \xi_{\min} \tilde{\mathbf{u}}_3^i. \tag{10}$$

Here the principal terms ensure the new primitive $\mathcal{G}_{\mathrm{obj}}^{i'}$ stays near the tangent plane $\mathcal{T}_\mathbf{q} \mathcal{M}$, while the last term allows a slight oscillation in the normal direction $\tilde{\mathbf{u}}_3^i$. For using curvatures $\{\tau_d^i\}_{1 \leqslant d \leqslant D_\mathcal{M}}$ as weights, it aims to bias the sampling towards sparse areas.

Secondly, in a similar way as our previous gradient truncation (i.e., Eq. (7)), we regularize the clone operation by clipping the gradient accumulation $\bar{\nabla}^i \mathfrak{L}$ as

$$\boldsymbol{\mu}_{\mathrm{obj}}^{i'} = \boldsymbol{\mu}_{\mathrm{obj}}^i + (\bar{\nabla}^i \mathfrak{L})^\top + \min\big(\xi_{\min}/\|(\bar{\nabla}^i \mathfrak{L})^\perp\|, 1\big)(\bar{\nabla}^i \mathfrak{L})^\perp. \tag{11}$$

The truncated shift in the normal direction $(\cdot)^\perp$ will lead to fewer new outlier primitives.

## 3.2 Noise-robust and Efficient Geometric Estimations

Previous works obtained noisy geometric priors through potentially unreliable methods (e.g., local PCA), and the priors were also kept static during optimization. As Gaussian primitives got denser, those priors became even more unreliable. To address their limitations, we present noise-robust and efficient estimation methods that can provide dynamic geometric information.

**Manifold-based method.** Our estimation method assumes that every Gaussian primitive $\mathcal{G}_{\text{obj}}^i$ lies on a local manifold $\mathcal{M}$ (i.e., Definition 2.1). We summarize below how it works, and due to the space limitation, we provide derivation details in Appendix E.

**Theorem 3.2** (Simplified from Proposition E.2 and Theorem E.7). *For a Gaussian primitive $\mathcal{G}_{\text{obj}}^i$ that lies on the embedded local manifold $\mathcal{M} \hookrightarrow \mathcal{E}$ at position $\mathbf{q} = \boldsymbol{\mu}_{\text{obj}}^i$, the eigenvectors of its tangential kernel matrix $\mathcal{K}^i$ (as formulated by Eq. (42)) that correspond to non-zero eigenvalues form an orthonormal basis $\{\mathbf{u}_d^i\}_{1 \leqslant d \leqslant D_{\mathcal{M}}}$ in the tangent space $\mathcal{T}_{\mathbf{q}}\mathcal{M}$.*

*Likewise, another shape operator matrix $\boldsymbol{\mathcal{S}}^i$ (as formulated by Eq. (51)) can be decomposed into pairs of eigenvalue and realigned eigenvector, which correspond to the principal curvatures and directions $\{(\tau_d^i, \widetilde{\mathbf{w}}_d^i)\}_{1 \leqslant d \leqslant D_{\mathcal{M}}}$ for the primitive $\mathcal{G}_{\text{obj}}^i$.*

For the normal information $\widetilde{\mathbf{u}}_3^i$ in 3D reconstruction, we can easily compute its unnormalized version through the cross product of two tangent vectors: $\mathbf{u}_{d_1}^i \times \mathbf{u}_{d_2}^i, d_1 \neq d_2$.

**Varifold-based method.** We have explored a different geometric approach based on varifolds. Generalizing the notion of second fundamental form valid for manifolds, Buet et al. (2022) introduced for point cloud varifolds an approximate second fundamental form (WSFF) as a matrix whose eigendecomposition provides full approximate curvature information. In Appendix D, we adapt that matrix to our setting, with its expression $\mathcal{B}^i$ for a Gaussian primitive $\mathcal{G}_{\text{obj}}^i$ as

$$\begin{cases} \mathcal{B}^i = \Big( \sum_{j \in \text{KNN}(i)} m_j \chi_\epsilon(\|\boldsymbol{\mu}_{\text{obj}}^i - \boldsymbol{\mu}_{\text{obj}}^j\|) \Big)^{-1} \sum_{j \in \text{KNN}(i)} \frac{m_j \Upsilon_\epsilon'(\|\boldsymbol{\mu}_{\text{obj}}^i - \boldsymbol{\mu}_{\text{obj}}^j\|)}{3\|\boldsymbol{\mu}_{\text{obj}}^i - \boldsymbol{\mu}_{\text{obj}}^j\|} \mathbf{B}^{i,j} \\ \mathbf{B}^{i,j} = 2(\widetilde{\mathbf{V}}^j \mathbf{V}^i)^\top \text{sym}\big(\widetilde{\mathbf{u}}_3^i(\boldsymbol{\mu}_{\text{obj}}^i - \boldsymbol{\mu}_{\text{obj}}^j)^\top\big)\widetilde{\mathbf{V}}^j \mathbf{V}^i + (\boldsymbol{\mu}_{\text{obj}}^i - \boldsymbol{\mu}_{\text{obj}}^j)^\top \widetilde{\mathbf{V}}^j \widetilde{\mathbf{u}}_3^i((\mathbf{V}^i)^\top \widetilde{\mathbf{V}}^j \mathbf{V}^i - \mathbf{I}) \end{cases}, \quad (12)$$

where $m_j$ denotes a weight, $\chi_\epsilon, \Upsilon_\epsilon$ are kernel functions that depend on the approximation scale $\epsilon$, $\text{sym}(\square) = (\square + \square^\top)/2$ is a symmetrization operation for any matrix $\square$, and $\widetilde{\mathbf{V}}^j = \mathbf{I} - \widetilde{\mathbf{u}}_3^i(\widetilde{\mathbf{u}}_3^i)^T$ is the orthogonal projection matrix onto the tangent plane $\mathcal{T}_{\mathbf{q}}\mathcal{M}$, with its basis as $\mathbf{V}^i = [\mathbf{u}_1^i; \mathbf{u}_2^i]$.

**Dynamic estimations during optimization.** Our estimation methods are both noise-robust and run-time efficient, which can process million-level Gaussian primitives in only tens of seconds. To update geometric information for the growing Gaussian primitives, we perform estimations at evenly spaced iterations during training, incurring a minor time increase.

## 4 Related Work

Common 3D scenes have a clear geometric structure (e.g., smoothness), so there is an emerging field in the literature (Wang et al., 2024; Bonilla et al., 2024; Li et al., 2025) that aims to regularize the optimization of Gaussian splatting in terms of certain geometric information. For example, Wang et al. (2024); Turkulainen et al. (2025) aligned the Gaussian primitive towards the normal direction, and Li et al. (2025) not only considered that, but also placed the densified primitives in the tangent plane. While achieving noticeable performance improvements, those earlier studies mainly considered low-order geometric quantities (e.g., normal vector), neglecting the valuable information contained in higher-order ones (e.g., curvature). As a rare case, Ververas et al. (2025) adopted the mean curvature to identify flat areas, though this was not a fully correct strategy (as explained in Sec. 3.1.1) and their implementation was still based on low-order quantities. A major contribution of this paper is the introduction of *GeoSplat*: a geometry-constrained optimization framework that exploits the higher-order geometric information neglected by prior works.

Previous studies were limited by potentially unreliable estimation methods, which yielded static and noise-sensitive geometric priors. For example, Wang et al. (2024) adopted the StableNormal

| Method | Metric | R0 | R1 | R2 | OFF0 | OFF1 | OFF2 | OFF3 | OFF4 |
|---|---|---|---|---|---|---|---|---|---|
| Vox-Fusion (Yang et al., 2022) | PSNR↑ | 22.39 | 22.36 | 23.92 | 27.79 | 29.83 | 20.33 | 23.47 | 25.21 |
| | SSIM↑ | 0.683 | 0.751 | 0.798 | 0.857 | 0.876 | 0.794 | 0.803 | 0.847 |
| | LPIPS↓ | 0.303 | 0.269 | 0.234 | 0.241 | 0.184 | 0.243 | 0.213 | 0.199 |
| Point-SLAM (Sandström et al., 2023) | PSNR↑ | 32.40 | 34.08 | 35.50 | 38.26 | 39.16 | 33.99 | 33.48 | 33.49 |
| | SSIM↑ | 0.974 | 0.977 | 0.982 | 0.983 | 0.986 | 0.960 | 0.960 | 0.979 |
| | LPIPS↓ | 0.113 | 0.116 | 0.111 | 0.100 | 0.118 | 0.156 | 0.132 | 0.142 |
| Gaussian-Splatting SLAM (Matsuki et al., 2024) | PSNR↑ | 34.83 | 36.43 | 37.49 | 39.95 | 42.09 | 36.24 | 36.70 | 36.07 |
| | SSIM↑ | 0.954 | 0.959 | 0.965 | 0.971 | 0.977 | 0.964 | 0.963 | 0.957 |
| | LPIPS↓ | 0.068 | 0.076 | 0.070 | 0.072 | 0.055 | 0.078 | 0.065 | 0.099 |
| GeoGaussian (Li et al., 2025) | PSNR↑ | 35.20 | 38.24 | 39.14 | 42.74 | 42.20 | 37.31 | 36.66 | 38.74 |
| | SSIM↑ | 0.952 | 0.979 | 0.970 | 0.981 | 0.970 | 0.970 | 0.964 | 0.967 |
| | LPIPS↓ | 0.029 | 0.021 | 0.024 | 0.016 | 0.040 | 0.029 | 0.029 | 0.031 |
| Our Model: *GeoSplat*, w/ Manifold-based Priors | PSNR↑ | **36.37** | **39.55** | **40.36** | 43.38 | **42.70** | **37.74** | 37.06 | 39.31 |
| | SSIM↑ | **0.976** | **0.982** | **0.985** | 0.984 | **0.988** | **0.975** | **0.969** | 0.972 |
| | LPIPS↓ | **0.024** | **0.018** | 0.021 | 0.015 | 0.037 | **0.027** | **0.027** | 0.028 |
| Our Model: *GeoSplat*, w/ Varifold-based Priors | PSNR↑ | 36.35 | 39.54 | 40.05 | **43.41** | 42.64 | 37.72 | **37.08** | **39.52** |
| | SSIM↑ | 0.973 | 0.975 | 0.981 | **0.985** | 0.986 | 0.974 | 0.968 | **0.981** |
| | LPIPS↓ | 0.025 | 0.019 | 0.022 | **0.013** | 0.971 | 0.028 | 0.028 | **0.027** |

Table 1: Performance comparison on a number of Replica datasets.

| Method | Metric | Room-1 | Room-2 | Office-2 | Office-3 |
|---|---|---|---|---|---|
| 3D Gaussian Splatting (3DGS) (Kerbl et al., 2023) | PSNR↑ | 40.79 | 39.10 | 37.88 | 36.04 |
| | SSIM↑ | 0.973 | 0.974 | 0.962 | 0.975 |
| | LPIPS↓ | 0.025 | 0.017 | 0.024 | 0.017 |
| LightGS (Fan et al., 2024) | PSNR↑ | 41.26 | 39.23 | 37.99 | 36.06 |
| | SSIM↑ | 0.974 | 0.974 | 0.962 | 0.975 |
| | LPIPS↓ | 0.023 | 0.017 | 0.023 | 0.016 |
| GeoGaussian (Li et al., 2025) | PSNR↑ | 41.43 | 39.46 | 38.54 | 36.19 |
| | SSIM↑ | 0.976 | 0.975 | 0.967 | 0.977 |
| | LPIPS↓ | 0.019 | 0.018 | 0.017 | 0.015 |
| Our Model: *GeoSplat*, w/ Manifold-based Priors | PSNR↑ | 41.81 | **39.97** | **38.75** | **36.61** |
| | SSIM↑ | 0.978 | **0.979** | **0.971** | 0.978 |
| | LPIPS↓ | 0.016 | **0.013** | **0.016** | 0.013 |
| Our Model: *GeoSplat*, w/ Varifold-based Priors | PSNR↑ | **41.92** | 39.80 | 38.68 | 36.60 |
| | SSIM↑ | **0.980** | 0.977 | 0.969 | **0.979** |
| | LPIPS↓ | **0.015** | 0.015 | 0.017 | **0.012** |

Table 2: Performance comparison on multiple ICL datasets.

model (Ye et al., 2024) to predict normal vectors, which tends to fail on unseen data, while Li et al. (2025) employed Agglomerative Hierarchical Clustering (AHC) (Feng et al., 2014), a classical clustering method that is not robust to sparsity. Even worse, as Gaussian primitives become denser during optimization, these static priors become increasingly unreliable. Based on inherent geometric structures (e.g., local manifolds), we propose efficient estimation methods to provide dynamic and reliable geometric information during optimization.

## 5 EXPERIMENTS

In this section, we show extensive experiment results that verify the effectiveness of our framework. We provide the details of our experiment setup and more results in Appendix F.

### 5.1 MAIN RESULTS

We compare our models with a number of baselines (e.g., GeoGaussian) on two types of view synthesis datasets, Replica (Straub et al., 2019) and ICL (Handa et al., 2014), across 12 different scenes. The results are provided in Table 1 and Table 2, showing that our optimization framework, *GeoSplat*, is able to significantly improve Gaussian splatting and outperform previous baselines. For the first point, we can see that our varifold-based model achieves consistent noticeable performance gains relative to 3DGS (Kerbl et al., 2023), which are 2.77% on ICL Room-1 in terms of PSNR, and our manifold-based model also outperforms it by 0.93% on ICL Office-2 in terms of SSIM. For the second point, we can see that both our manifold-based and varifold-based models significantly outperform the key baseline, GeoGaussian, on every Replica dataset. For example, we achieve higher PSNR scores by 3.42% on Replica R1 and 2.01% on Replica OFF4. The consistent performance gains over a dozen of datasets indicate that our framework is indeed effective.

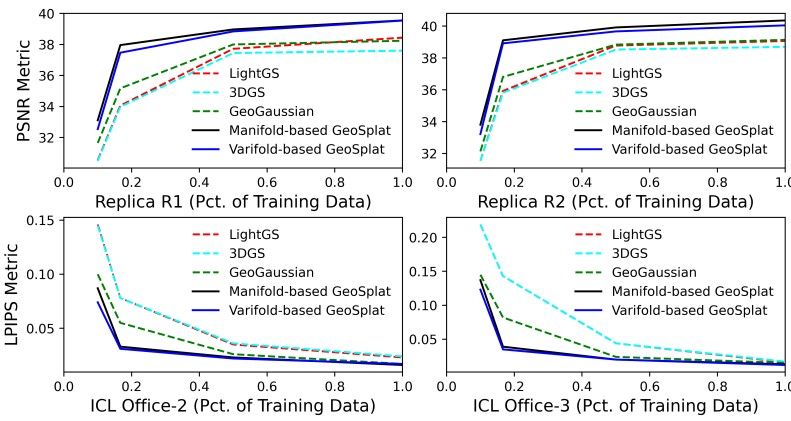

Figure 1: The performance of our models and baselines in low-resource settings.

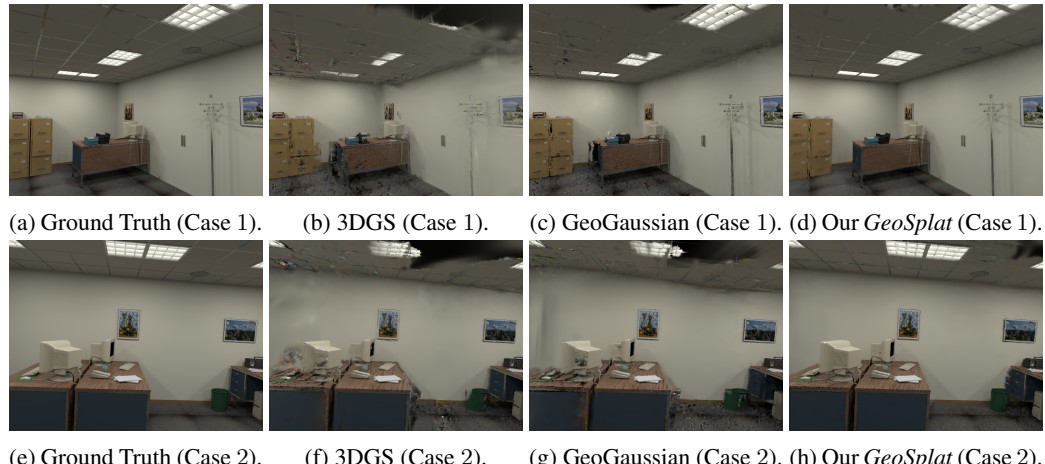

(a) Ground Truth (Case 1).   (b) 3DGS (Case 1).   (c) GeoGaussian (Case 1).   (d) Our *GeoSplat* (Case 1).

(e) Ground Truth (Case 2).   (f) 3DGS (Case 2).   (g) GeoGaussian (Case 2).   (h) Our *GeoSplat* (Case 2).

Figure 2: Ground-truth and rendered images on the low-resource ICL Office-2 dataset. The cases 1, 2 are respectively generated by our manifold-based and varifold-based models.

## 5.2 LOW-RESOURCE SETTING

Since our models are regularized by various types of geometric priors, it is intuitive that we might obtain even higher performance gains in low-resource settings, where observed views are sparse in the dataset. To verify this intuition, we compare our models with the baselines on 4 datasets from both Replica and ICL, with some percentage (pct.) of views excluded. The results are provided in Fig. 2, showing that our models have much slower performance decreases when views got sparser, even compared with GeoGaussian (which was regularized by low-order normal information). For example, our performance gain in terms of PSNR is $3.11\%$ on the full Replica R2 dataset, and can be further enlarged as $7.93\%$ when only $1/6$ views are observed.

## 5.3 CASE STUDIES

To visualize the effect of geometric regularization, we first compare the images rendered from our model with those from the baselines on sparse ICL Office-2 ($80\%$ views excluded). The results are demonstrated in Fig. 2. We can see that even the images generated by GeoGaussian (Subfig. 2c and Subfig. (2g)) contain perceivable floating artifacts, exhibiting large holes on the ceiling. In contrast, the images rendered by our models are noticeably cleaner and smoother, which benefits from our geometric regularization strategies. Figure 3 further shows the primitive clouds of our models and the baselines. We can see that the latter contain substantially more outlier primitives and holes in low-texture regions. These observations confirm the effectiveness of our framework.

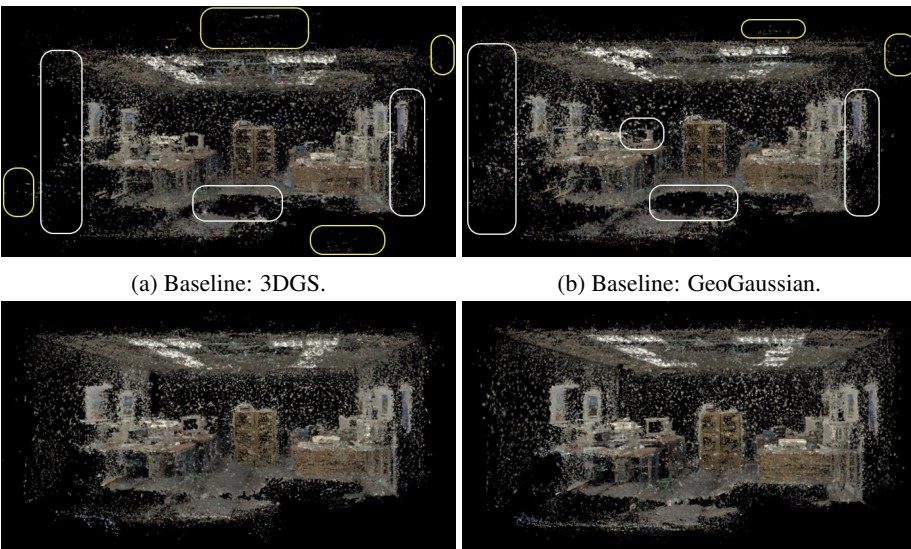

(a) Baseline: 3DGS.  (b) Baseline: GeoGaussian.

(c) Our Model: Varifold-based *GeoSplat*.  (d) Our Model: Manifold-based *GeoSplat*.

Figure 3: Colored Gaussian primitives on the low-resource ICL Office-2 dataset. The boxes in white and yellow respectively indicate sparse areas and outlier artifacts.

| Method | Replica R0 | | | ICL Office-3 | | |
|---|---|---|---|---|---|---|
| | PSNR↑ | SSIM↑ | LPIPS↓ | PSNR↑ | SSIM↑ | LPIPS↓ |
| Our Model: Manifold-based *GeoSplat* | **36.37** | **0.976** | **0.024** | **36.61** | **0.978** | **0.013** |
| w/o Curvature-guided Covariance Warm-up (Eq. (5)) | 36.01 | 0.969 | 0.026 | 36.43 | 0.976 | 0.014 |
| w/o Curvature-guided Primitive Upsampling (Eq. (6)) | 36.12 | 0.971 | 0.025 | 36.51 | 0.976 | 0.015 |
| w/o MAC, w/ Mean Curvature (Ververas et al., 2025) | 36.25 | 0.973 | 0.025 | 36.58 | 0.977 | 0.014 |
| w/o Truncated Gradient Update (Eq. (7)) | 36.09 | 0.970 | 0.026 | 36.47 | 0.975 | 0.015 |
| w/o Shape Regularization (Eq. (8), Eq. (9)) | 35.98 | 0.965 | 0.027 | 36.39 | 0.973 | 0.016 |
| w/o Curvature-regularized Densification (Eq. (10), Eq. (11)) | 36.05 | 0.970 | 0.026 | 36.48 | 0.975 | 0.015 |
| Our Model: Varifold-based *GeoSplat* | **36.35** | **0.973** | **0.025** | **36.60** | **0.979** | **0.012** |
| w/o Curvature-guided Covariance Warm-up (Eq. (5)) | 36.00 | 0.963 | 0.027 | 36.41 | 0.976 | 0.015 |
| w/o Curvature-guided Primitive Upsampling (Eq. (6)) | 36.17 | 0.968 | 0.026 | 36.53 | 0.977 | 0.013 |
| w/o MAC, w/ Mean Curvature (Ververas et al., 2025) | 36.23 | 0.971 | 0.026 | 36.57 | 0.978 | 0.013 |
| w/o Truncated Gradient Update (Eq. (7)) | 36.05 | 0.967 | 0.028 | 36.45 | 0.977 | 0.014 |
| w/o Shape Regularization (Eq. (8), Eq. (9)) | 35.93 | 0.961 | 0.029 | 36.37 | 0.975 | 0.016 |
| w/o Curvature-regularized Densification (Eq. (10), Eq. (11)) | 36.03 | 0.964 | 0.028 | 36.43 | 0.977 | 0.014 |

Table 3: Experiment results from the ablation studies on our models.

## 5.4 ABLATION STUDIES

We conduct ablation experiments to quantitatively measure the impacts of our regularization strategies, further confirming their effectiveness. As shown in Table 3, the performance of our model gets degraded after taking any component out from our framework. For example, without shape regularization, the PSNR scores of our manifold-based model noticeably decrease by $1.07\%$ on Replica R0 and $0.60\%$ on ICL Office-3. We can also see that using MAC as the flat-area identifier, instead of the mean curvature, makes our primitive upsampling strategy work better.

## 6 CONCLUSION

In this work, we introduce a general framework of geometry-constrained Gaussian splatting, with an emphasize to exploit higher-order geometric information (e.g., curvature) that is largely neglected by previous methods. The experiment results show that our framework significantly improves Gaussian splatting and outperforms previous baselines, with very few artifacts in low-resource settings. Building on inherent geometric structures (e.g., local manifolds), we present efficient estimation methods that provide our framework with noise-robust and dynamic geometric priors, thereby overcoming the reliance on static and noise-sensitive priors in existing approaches.

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

## A    REVIEW OF BASIC GEOMETRY

In this section, we will briefly review two different geometric structures: manifolds (Lee, 2018) in differential geometry and varifolds (Allard, 1972; Menne, 2017) in geometric measure theory.

### A.1    MANIFOLD IN RIEMANNIAN GEOMETRY

A smooth manifold $\mathcal{M}$ in Riemannian geometry (Lee, 2018) is associated with Riemannian metric $g$. An informal understanding is to view it as a point set that locally looks like a vector space, with inner products defined variously for different local spaces. In the rigorous formulation, such a local space is conceptualized as the tangent plane $\mathcal{T}_{\mathbf{q}}\mathcal{M}$ at every point $\mathbf{q} \in \mathcal{M}$, with metric $g_{\mathbf{q}} : \mathcal{T}_{\mathbf{q}}\mathcal{M} \times \mathcal{T}_{\mathbf{q}}\mathcal{M} \to \mathbb{R}$ as the inner product to measure both vector length $\|\mathbf{v}\|^2 = g_{\mathbf{q}}(\mathbf{v}, \mathbf{v}), \mathbf{v} \in \mathcal{T}_{\mathbf{q}}\mathcal{M}$ and similarity $g_{\mathbf{q}}(\mathbf{v}_1, \mathbf{v}_2), \mathbf{v}_1, \mathbf{v}_2 \in \mathcal{T}_{\mathbf{q}}\mathcal{M}$. All tangent planes share the same dimension, which is consistent with the defined dimension $D_{\mathcal{M}} \in \mathbb{N}$ of the global manifold, and their disjoint union, the tangent bundle $\sqcup_{\mathbf{q} \in \mathcal{M}} \mathcal{T}_{\mathbf{q}}\mathcal{M}$, is denoted as $\mathcal{T}\mathcal{M}$. A smooth section $\mathbf{V}$ of the bundle (i.e., vector field on the manifold) is a map that pairs every point $\mathbf{q}$ with a tangent vector $\mathbf{V}(\mathbf{q}) \in \mathcal{T}_{\mathbf{q}}\mathcal{M}$. We denote the space of all smooth sections as $\mathcal{V}(\mathcal{M})$, and represent the set containing all smooth function $f : \mathcal{M} \to \mathbb{R}$ as $\mathcal{C}^{\infty}(\mathcal{M})$. For the normal space $\mathcal{N}_{\mathbf{q}}\mathcal{M}$ at each point $\mathbf{q}$ on the manifold $\mathcal{M}$, it is easier to define it in a Euclidean ambient space $\mathcal{E} = \mathbb{R}^{D_{\mathcal{E}}}$ that isometrically embeds manifold $\mathcal{M}$ and has a larger dimension $D_{\mathcal{E}} > D_{\mathcal{M}}$. In that sense, the space $\mathcal{N}_{\mathbf{q}}\mathcal{M}$ consists of normal vector $\mathbf{n} \in \mathcal{E}$ that is perpendicular to the tangent space $\mathcal{T}_{\mathbf{q}}\mathcal{M}$.

A nice property of the Riemannian manifold is that its metric $g$ uniquely determines an affine connection $\nabla_{\mathbf{V}_1} \mathbf{V}_2 : \mathcal{V}(\mathcal{M}) \times \mathcal{V}(\mathcal{M}) \to \mathcal{V}(\mathcal{M})$, which can be applied to further induce the Riemannian curvature tensor $\mathcal{R}(\mathbf{V}_1, \mathbf{V}_2)\mathbf{V}_3$ as

$$\mathcal{R}(\mathbf{V}_1, \mathbf{V}_2)\mathbf{V}_3 = \nabla_{\mathbf{V}_1}\nabla_{\mathbf{V}_2}\mathbf{V}_3 - \nabla_{\mathbf{V}_2}\nabla_{\mathbf{V}_1}\mathbf{V}_3 - \nabla_{[\mathbf{V}_1, \mathbf{V}_2]}\mathbf{V}_3, \tag{13}$$

where operator $[,]$ represents the Lie bracket (Hall, 2013). This tensor defines all other types of curvature. For example, the sectional curvature:

$$\mathrm{Sec}(\mathbf{V}_1, \mathbf{V}_2) = \frac{g(\mathcal{R}(\mathbf{V}_1, \mathbf{V}_2)\mathbf{V}_1, \mathbf{V}_2)}{g(\mathbf{V}_1, \mathbf{V}_1)g(\mathbf{V}_2, \mathbf{V}_2) - g(\mathbf{V}_1, \mathbf{V}_2)^2}, \tag{14}$$

which measures how the manifold is curved in a 2D subspace.

### A.2    ANOTHER TOOL: VARIFOLDS

While manifold is a popular geometric language, it relies on a strong assumption of smoothness. The notion of varifold (Allard, 1972; Simon, 1983; Menne, 2017) is another mathematical tool that requires only weak regularity. For the sake of clarity, we first focus on the simpler subclass of rectifiable varifolds. Given a countably $D_{\mathsf{M}}$-rectifiable set (Federer, 1969) $\mathsf{M}$ that is contained in an open set $\Omega \subset \mathcal{E}$ and a nonnegative function $\theta : \mathsf{M} \to \mathbb{R}$ that is positive $\mathcal{H}^{D_{\mathsf{M}}}$-almost everywhere, the rectifiable varifold $\mathcal{W}$ is defined as the nonnegative Radon measure $\theta \mathcal{H}_{|\mathsf{M}}^{D_{\mathsf{M}}} \otimes \delta_{\mathcal{T}_x \mathsf{M}}$ on the product space $\mathsf{M} \times \mathsf{G}_{D_{\mathsf{M}}, D_{\mathcal{E}}}$. Here $\mathcal{H}_{|\mathsf{M}}^{D_{\mathsf{M}}}$ denotes the $D_{\mathsf{M}}$-dimensional Hausdorff measure supported on $\mathsf{M}$ (Federer, 1969), $\mathsf{G}_{D_{\mathsf{M}}, D_{\mathcal{E}}}$ is the Grassmannian manifold (Lee, 2018) formed by all $D_{\mathsf{M}}$-dimensional linear subspaces of the ambient space $\mathcal{E}$, and $\delta_{\mathcal{T}_x \mathsf{M}}$ is the Dirac measure in $\mathsf{G}_{D_{\mathsf{M}}, D_{\mathcal{E}}}$ supported on the tangent plane $\mathcal{T}_x \mathsf{M}$ at $x \in \mathsf{M}$.

The action of $\mathcal{W}$ on a continuous function $\varphi$ compactly supported in $\Omega \times \mathsf{G}_{D_{\mathsf{M}}, D_{\mathcal{E}}}$ is

$$\int_{\Omega \times \mathsf{G}_{D_{\mathsf{M}}, D_{\mathcal{E}}}} \varphi(\mathbf{q}, \mathsf{T}) \mathrm{d}\mathcal{W}(\mathbf{q}, \mathsf{T}) = \int_{\mathsf{M}} \varphi(\mathbf{q}, \mathcal{T}_{\mathbf{q}}\mathsf{M})\theta(\mathbf{q})\mathrm{d}\mathcal{H}^{D_{\mathsf{M}}}(\mathbf{q}). \tag{15}$$

Denoting as $\mathsf{P}$ the canonical projection from the product space $\mathsf{M} \times \mathsf{G}_{D_{\mathsf{M}}, D_{\mathcal{E}}}$ onto $\mathsf{M}$, we define the mass $\|\mathcal{W}\|$ of $\mathcal{W}$ as the Radon measure such that, for any Borel set $B \subset \Omega$,

$$\|\mathcal{W}\|(B) = \mathcal{W}(\mathsf{P}^{-1}(B)) = \int_{\mathsf{M} \cap B} \theta(\mathbf{q})\mathrm{d}\mathcal{H}^{D_{\mathsf{M}}}(\mathbf{q}). \tag{16}$$

A key concept in the theory of varifolds is the notion of first variation $\delta\mathcal{W}$ of a varifold $\mathcal{W}$: for any smooth compactly supported vector field $\mathsf{X}$

$$\delta\mathcal{W}(\mathsf{X}) = \int_{\Omega \times \mathsf{G}_{D_{\mathsf{M}}, D_{\mathcal{E}}}} \mathrm{div}^{\mathsf{T}}\mathsf{X}(\mathbf{q})\mathrm{d}\mathcal{W}(\mathbf{q}, \mathsf{T}), \tag{17}$$

where $\mathrm{div}^{\mathsf{T}}$ denotes the tangential divergence with respect to $T$. If the linear functional $\delta\mathcal{W}$ is locally bounded, it follows from Riesz theorem (Yosida, 2012) and Radon-Nikodym theorem (Halmos, 2013) that it can be associated with a unique Radon measure that admits the decomposition

$$\delta\mathcal{W} = -\mathsf{H}\|\mathcal{W}\| + \delta^S\mathcal{W}, \tag{18}$$

where $\delta^S\mathcal{W}$ is a vector-valued measure singular with respect to the mass $\|\mathcal{W}\|$, and $\mathsf{H}$ is called the generalized mean curvature. The latter coincides with the classical notion of mean curvature when $\mathcal{W}$ is a rectifiable varifold associated with a smooth set and a locally constant multiplicity. More generally, a generalized notion of second fundamental form can be defined for varifolds, see Buet et al. (2022). The first variation of a point varifold is not locally bounded in general, so the above decomposition does not hold. However, it is possible to define for point cloud varifolds consistent notions of approximate mean curvature Buet et al. (2017) and approximate weak second fundamental form, see Buet et al. (2022) and Appendix D.

## B    More Preliminaries on Manifold: Unique Connection

The affine connection $\nabla^{\mathcal{M}}$ that is compatible with the metric $g^{\mathcal{M}}$ turns out to be unique. The Koszul identity (Lee, 2018) provides an explicit formula for this connection. We show below its derivation process. Based on the metric compatibility, we have

$$\begin{cases} \partial_{\mathbf{U}_1}^{\mathcal{M}}g^{\mathcal{M}}(\mathbf{U}_2, \mathbf{U}_3) = g^{\mathcal{M}}(\nabla_{\mathbf{U}_1}^{\mathcal{M}}\mathbf{U}_2, \mathbf{U}_3) + g^{\mathcal{M}}(\mathbf{U}_2, \nabla_{\mathbf{U}_1}^{\mathcal{M}}\mathbf{U}_3) \\ \partial_{\mathbf{U}_2}^{\mathcal{M}}g^{\mathcal{M}}(\mathbf{U}_3, \mathbf{U}_1) = g^{\mathcal{M}}(\nabla_{\mathbf{U}_2}^{\mathcal{M}}\mathbf{U}_3, \mathbf{U}_1) + g^{\mathcal{M}}(\mathbf{U}_3, \nabla_{\mathbf{U}_2}^{\mathcal{M}}\mathbf{U}_1) \\ \partial_{\mathbf{U}_3}^{\mathcal{M}}g^{\mathcal{M}}(\mathbf{U}_1, \mathbf{U}_2) = g^{\mathcal{M}}(\nabla_{\mathbf{U}_3}^{\mathcal{M}}\mathbf{U}_1, \mathbf{U}_2) + g^{\mathcal{M}}(\mathbf{U}_1, \nabla_{\mathbf{U}_3}^{\mathcal{M}}\mathbf{U}_2)) \end{cases} . \tag{19}$$

By adding the first two equations and subtracting the last one, we get

$$\partial_{\mathbf{U}_1}^{\mathcal{M}}g^{\mathcal{M}}(\mathbf{U}_2, \mathbf{U}_3) + \partial_{\mathbf{U}_2}^{\mathcal{M}}g^{\mathcal{M}}(\mathbf{U}_3, \mathbf{U}_1) - \partial_{\mathbf{U}_3}^{\mathcal{M}}g^{\mathcal{M}}(\mathbf{U}_1, \mathbf{U}_2) = g^{\mathcal{M}}(\nabla_{\mathbf{U}_1}^{\mathcal{M}}\mathbf{U}_2 + \nabla_{\mathbf{U}_2}^{\mathcal{M}}\mathbf{U}_1, \mathbf{U}_3)$$
$$+ g^{\mathcal{M}}(\nabla_{\mathbf{U}_1}^{\mathcal{M}}\mathbf{U}_3 - \nabla_{\mathbf{U}_3}^{\mathcal{M}}\mathbf{U}_1, \mathbf{U}_2) + g^{\mathcal{M}}(\nabla_{\mathbf{U}_2}^{\mathcal{M}}\mathbf{U}_3 - \nabla_{\mathbf{U}_3}^{\mathcal{M}}\mathbf{U}_2, \mathbf{U}_1). \tag{20}$$

In terms of the torsion-free condition:

$$\nabla_{\mathbf{U}}^{\mathcal{M}}\mathbf{U}' - \nabla_{\mathbf{U}'}^{\mathcal{M}}\mathbf{U} = [\mathbf{U}, \mathbf{U}'], \tag{21}$$

for any two vector fields $\mathbf{U}, \mathbf{U}'$, we can simplify the merged equation as

$$\partial_{\mathbf{U}_1}^{\mathcal{M}}g^{\mathcal{M}}(\mathbf{U}_2, \mathbf{U}_3) + \partial_{\mathbf{U}_2}^{\mathcal{M}}g^{\mathcal{M}}(\mathbf{U}_3, \mathbf{U}_1) - \partial_{\mathbf{U}_3}^{\mathcal{M}}g^{\mathcal{M}}(\mathbf{U}_1, \mathbf{U}_2) =$$
$$g^{\mathcal{M}}(2\nabla_{\mathbf{U}_1}^{\mathcal{M}}\mathbf{U}_2 + [\mathbf{U}_2, \mathbf{U}_1], \mathbf{U}_3) + g^{\mathcal{M}}([\mathbf{U}_1, \mathbf{U}_3], \mathbf{U}_2) + g^{\mathcal{M}}([\mathbf{U}_2, \mathbf{U}_3], \mathbf{U}_1). \tag{22}$$

By reorganizing this equality, we finally get

$$g^{\mathcal{M}}(\nabla_{\mathbf{U}_1}^{\mathcal{M}}\mathbf{U}_2, \mathbf{U}_3) = \frac{1}{2}\Big(\partial_{\mathbf{U}_1}^{\mathcal{M}}g^{\mathcal{M}}(\mathbf{U}_2, \mathbf{U}_3) + \partial_{\mathbf{U}_2}^{\mathcal{M}}g^{\mathcal{M}}(\mathbf{U}_3, \mathbf{U}_1) - \partial_{\mathbf{U}_3}^{\mathcal{M}}g^{\mathcal{M}}(\mathbf{U}_1, \mathbf{U}_2)+$$
$$g^{\mathcal{M}}([\mathbf{U}_1, \mathbf{U}_2], \mathbf{U}_3) - g^{\mathcal{M}}([\mathbf{U}_1, \mathbf{U}_3], \mathbf{U}_2) - g^{\mathcal{M}}([\mathbf{U}_2, \mathbf{U}_3], \mathbf{U}_1)\Big), \tag{23}$$

which holds for any smooth vector fields $\mathbf{U}_1, \mathbf{U}_2, \mathbf{U}_3$.

## C    Proof: Curvature-based Scale Initialization

Given the curvature-based and area constraints as

$$\frac{s_1^i}{s_2^i} = \left|\frac{\tau_1^i}{\tau_2^i}\right|, \quad s_1^i s_2^i = \frac{(s_{\mathrm{nbr}}^i)^2}{4}, \tag{24}$$

let us first solve the scale variables $s_1^i, s_2^i$. For the unknown scale $s_1^i$, we have

$$s_1^i = \sqrt{\left(s_1^i s_2^i\right)\left(\frac{s_1^i}{s_2^i}\right)} = \sqrt{\frac{(s_{\mathrm{nbr}}^i)^2}{4}\left|\frac{\tau_1^i}{\tau_2^i}\right|} = \frac{s_{\mathrm{nbr}}^i}{2}\sqrt{\left|\frac{\tau_1^i}{\tau_2^i}\right|}. \tag{25}$$

Similarly, for the other one, we can derive

$$s_2^i = \sqrt{\left(s_1^i s_2^i\right)\left(1\Big/\frac{s_1^i}{s_2^i}\right)} = \sqrt{\frac{(s_{\mathrm{nbr}}^i)^2}{4}\left|\frac{\tau_2^i}{\tau_1^i}\right|} = \frac{s_{\mathrm{nbr}}^i}{2}\sqrt{\left|\frac{\tau_2^i}{\tau_1^i}\right|}. \tag{26}$$

Then, we aim to analyze the second claim. In statistics, the one-sigma region of Gaussian primitive $\mathcal{G}_{\mathrm{obj}}^i$ is defined as the ellipsoid spanned by the rotation directions $\mathbf{r}_1^i, \mathbf{r}_2^i, \mathbf{r}_3^i$ of covariance matrix $\boldsymbol{\Sigma}_{\mathrm{obj}}^i$, with its volume $\mathfrak{V}$ measured by the scales $s_1^i, s_2^i, s_3^i$ as

$$\mathfrak{V} = s_1^i s_2^i s_3^i = s_1^i s_2^i \xi_{\min}. \tag{27}$$

As Gaussian primitive $\mathcal{G}_{\mathrm{obj}}^i$ in our setting is oriented in terms of the normal direction: $\mathbf{r}_3^i = \widetilde{\mathbf{u}}_3^i$, the area $\mathfrak{A}$ of its projection to the tangent plane $\mathcal{T}_{\mathbf{q}}\mathcal{M}$ can be simply measured as

$$\mathfrak{A} = \frac{\mathfrak{V}}{s_3^i} = s_1^i s_2^i \frac{1}{4}(s_{\mathrm{nbr}}^i)^2 = \left(\frac{s_{\mathrm{nbr}}^i}{2}\right)\left(\frac{s_{\mathrm{nbr}}^i}{2}\right), \tag{28}$$

which is exactly the area before scale warm-up.

## D  ADAPTED EXPRESSION OF VARIFOLD-BASED CURVATURES

Buet et al. (2022) introduced a notion of approximate second fundamental form (WSFF) for point cloud varifolds, with convergence and consistency properties. The aim of this section is to adapt that notion to our setting.

**Method recap.**  To obtain full curvature information from a varifold $\mathcal{W}$, Buet et al. (2022) first introduced a collection of G-linear variations indexed by $i, j, k$ and defined by their action on $C_c^1(\Omega)$ functions as

$$\delta_{i,j,k}\mathcal{W}(\varphi) = \int_{\Omega \times \mathsf{G}_{D_{\mathsf{M}}, D_{\mathcal{E}}}} \mathsf{T}_{j,k}\langle \nabla^{\mathsf{T}}\varphi(\mathbf{q}), \mathbf{e}_i\rangle \mathrm{d}\mathcal{W}(\mathbf{q}, \mathsf{T}), \tag{29}$$

where every element $T$ of the Grassmannian is identified with its orthogonal projection matrix $(T_{jk})$, $\nabla^{\mathsf{T}}$ represents the tangential gradient with respect to $T$, and $\mathbf{e}_i$ denotes the $i$-th Euclidean basis vector. We say that $\mathcal{W}$ has locally bounded variations whenever all G-linear variations are Radon measures, in which case they can be decomposed as

$$\delta_{i,j,k}\mathcal{W} = -b_{i,j,k}\|\mathcal{W}\| + \delta_{i,j,k}^S\mathcal{W}. \tag{30}$$

where $b_{i,j,k} \in L^1(\|\mathcal{W}\|)$ and $\delta_{i,j,k}^S\mathcal{W}$ are Radon measures singular with respect to $\|\mathcal{W}\|$. When W is a point cloud varifold associated with (point, tangent plane, mass) triplets $\{(\mathbf{q}_i, \mathsf{T}_i, m_i)\}_{i \in \mathcal{I}}$, Buet et al. (2022) introduced a consistent notion of approximate weak second fundamental form (WSFF) defined as

$$\begin{cases} \mathsf{B}_{i,j,k}(\mathbf{q}_{l_1}) = \dfrac{\frac{D_{\mathcal{M}}}{2D_{\mathcal{E}}}\sum_{l_2} m_{l_2}\Upsilon'\left(\frac{\|\mathbf{q}_{l_1}-\mathbf{q}_{l_2}\|}{\epsilon}\right)\left\langle \frac{\mathsf{T}_{l_2}(\mathbf{q}_{l_1}-\mathbf{q}_{l_2})}{\|\mathbf{q}_{l_1}-\mathbf{q}_{l_2}\|}, \mathbf{t}_{l_1,l_2,i,j,k}\right\rangle}{\sum_{l_2} m_{l_2}\chi\left(\frac{\|\mathbf{q}_{l_1}-\mathbf{q}_{l_2}\|}{\epsilon}\right)}, \\ \mathbf{t}_{l_1,l_2,i,j,k} = (\mathsf{T}_{l_2} - \mathsf{T}_{l_1})_{j,k}\mathbf{e}_i + (\mathsf{T}_{l_2} - \mathsf{T}_{l_1})_{i,k}\mathbf{e}_j - (\mathsf{T}_{l_2} - \mathsf{T}_{l_1})_{i,j}\mathbf{e}_k \end{cases}, \tag{31}$$

where $\Upsilon, \chi$ are suitable kernel functions, and $\epsilon > 0$ is the approximation scale. The approximate curvature quantities can be finally obtained through the eigendecomposition of the matrix

$$\mathcal{B}(\mathbf{q}_{l_1}) = \bar{\mathsf{T}}_{l_1}^\top \{\langle \mathsf{B}_{i,j,:}(\mathbf{q}_{l_1}), \mathbf{n}_{l_1}\rangle\}_{1\leqslant i,j\leqslant D_{\mathcal{M}}} \bar{\mathsf{T}}_{l_1}, \tag{32}$$

where $i, j, :$ means taking all entries along the last index to form a vector, $\bar{\mathsf{T}}_{l_1}$ denotes an orthonormal basis of the tangent space $\mathsf{T}_{l_1}$, and $\mathbf{n}_{l_1}$ is a normal vector.

**Adaptation to our setting.** Now, let us adapt the above approximate WSFF $\mathcal{B}$ to our setting, facilitating practical computation. As the first step, we specify the points as the positions of Gaussian primitives, $\mathsf{M} = \{\boldsymbol{\mu}_{\text{obj}}^i\}_{i \in \mathcal{I}}$, with each element $\boldsymbol{\mu}_{\text{obj}}^i$ paired with a tangent plane $\widetilde{\mathbf{V}}^i$ that is determined by the normal vector $\widetilde{\mathbf{u}}_3^i$ as $\mathbf{I} - \widetilde{\mathbf{u}}_3^i(\widetilde{\mathbf{u}}_3^i)^\top$. Here $D_\mathcal{M} = 2$ for novel view rendering in the 3D world $\mathcal{E} = \mathbb{R}^3$. For better notational consistencies, we also relabel the middle term $\mathbf{B}_{i,j,k}(\mathbf{q}_{l_1})$ as $\mathsf{B}_{d_1,d_2,d_3}^i$ (given $\mathbf{q}_{l_1} = \boldsymbol{\mu}_{\text{obj}}^i$) and the subscript $l_2$ as $j$.

Then, we reshape the inner product $\langle \cdot, \mathbf{t}_{i,j,d_1,d_2,d_3} \rangle$ of term $\mathsf{B}^i$ based on its linearity:

$$\frac{(\widetilde{\mathbf{V}}^j - \widetilde{\mathbf{V}}^i)_{d_2,d_3} \langle \widetilde{\mathbf{V}}^j(\boldsymbol{\mu}_{\text{obj}}^i - \boldsymbol{\mu}_{\text{obj}}^j), \mathbf{e}_{d_1} \rangle + (\widetilde{\mathbf{V}}^j - \widetilde{\mathbf{V}}^i)_{d_1,d_3}\langle \cdot, \mathbf{e}_{d_2} \rangle - (\widetilde{\mathbf{V}}^j - \widetilde{\mathbf{V}}^i)_{d_1,d_2}\langle \cdot, \mathbf{e}_{d_3} \rangle}{\|\boldsymbol{\mu}_{\text{obj}}^i - \boldsymbol{\mu}_{\text{obj}}^j\|}. \tag{33}$$

As the tangential projection of a normal vector is zero, $\widetilde{\mathbf{V}}^i \widetilde{\mathbf{u}}_3^i = \mathbf{0}$, we have

$$\bar{t}_{i,j,d_1,d_2} = \left\langle \left\{ \langle \cdot, \mathbf{t}_{i,j,d_1,d_2,d_3} \rangle \|\boldsymbol{\mu}_{\text{obj}}^i - \boldsymbol{\mu}_{\text{obj}}^j\| \right\}_{1 \le d_3 \le D_\mathcal{E}}, \widetilde{\mathbf{u}}_3^i \right\rangle$$

$$= \langle (\widetilde{\mathbf{V}}^j - \widetilde{\mathbf{V}}^i)_{d_2,:}, \widetilde{\mathbf{u}}_3^i \rangle \langle \cdot, \mathbf{e}_{d_1} \rangle + \langle (\cdot)_{d_1,:}, \widetilde{\mathbf{u}}_3^i \rangle \langle \cdot, \mathbf{e}_{d_2} \rangle - (\widetilde{\mathbf{V}}^j - \widetilde{\mathbf{V}}^i)_{d_1,d_2}\langle \widetilde{\mathbf{V}}^j(\boldsymbol{\mu}_{\text{obj}}^i - \boldsymbol{\mu}_{\text{obj}}^j), \widetilde{\mathbf{u}}_3^i \rangle \tag{34}$$

$$= \langle \widetilde{\mathbf{V}}_{d_2,:}^j, \widetilde{\mathbf{u}}_3^i \rangle \langle \cdot, \mathbf{e}_{d_1} \rangle + \langle \widetilde{\mathbf{V}}_{d_1,:}^j, \widetilde{\mathbf{u}}_3^i \rangle \langle \cdot, \mathbf{e}_{d_2} \rangle - (\widetilde{\mathbf{V}}^j - \widetilde{\mathbf{V}}^i)_{d_1,d_2} \langle \widetilde{\mathbf{V}}^j(\boldsymbol{\mu}_{\text{obj}}^i - \boldsymbol{\mu}_{\text{obj}}^j), \widetilde{\mathbf{u}}_3^i \rangle.$$

With the above new term, we can convert the entry $\langle \mathsf{B}_{d_1,d_2,:}^i, \widetilde{\mathbf{u}}_3^i \rangle$ into

$$\sum_j \frac{m_j \Upsilon'(\|\boldsymbol{\mu}_{\text{obj}}^i - \boldsymbol{\mu}_{\text{obj}}^j\|/\epsilon) \bar{t}_{i,j,d_1,d_2}}{3\|\boldsymbol{\mu}_{\text{obj}}^i - \boldsymbol{\mu}_{\text{obj}}^j\| \sum_k m_k \chi(\|\boldsymbol{\mu}_{\text{obj}}^i - \boldsymbol{\mu}_{\text{obj}}^k\|/\epsilon)}, \tag{35}$$

given that $D_\mathcal{M} = 2, D_\mathcal{E} = 3$. Therefore, we express the approximate WSFF as an interpolated matrix

$$\mathcal{B}^i = \sum_j \frac{m_j \Upsilon'(\|\boldsymbol{\mu}_{\text{obj}}^i - \boldsymbol{\mu}_{\text{obj}}^j\|/\epsilon)\left((\mathbf{V}^i)^\top \bar{t}_{i,j,:,:} \mathbf{V}^i\right)}{3\|\boldsymbol{\mu}_{\text{obj}}^i - \boldsymbol{\mu}_{\text{obj}}^j\| \sum_k m_k \chi(\|\boldsymbol{\mu}_{\text{obj}}^i - \boldsymbol{\mu}_{\text{obj}}^k\|/\epsilon)}, \tag{36}$$

where $\mathbf{V}^i$ represents the tangential basis matrix $[\mathbf{u}_1^i; \mathbf{u}_2^i]$ in our setting. The last task is to simplify the matrix multiplication term in the numerator:

$$\mathbf{B}^{i,j} = (\mathbf{V}^i)^\top \bar{t}_{i,j,:,:} \mathbf{V}^i$$

$$= (\mathbf{V}^i)^\top \left( \widetilde{\mathbf{V}}^j(\boldsymbol{\mu}_{\text{obj}}^i - \boldsymbol{\mu}_{\text{obj}}^j)(\widetilde{\mathbf{V}}^j \widetilde{\mathbf{u}}_3^i)^\top + \widetilde{\mathbf{V}}^j \widetilde{\mathbf{u}}_3^i (\cdot)^\top + (\widetilde{\mathbf{V}}^j - \widetilde{\mathbf{V}}^i)\langle \widetilde{\mathbf{V}}^j(\boldsymbol{\mu}_{\text{obj}}^i - \boldsymbol{\mu}_{\text{obj}}^j), \widetilde{\mathbf{u}}_3^i \rangle \right) \mathbf{V}^i \tag{37}$$

$$= 2(\widetilde{\mathbf{V}}^j \mathbf{V}^i)^\top \text{sym}\left( \widetilde{\mathbf{u}}_3^i(\boldsymbol{\mu}_{\text{obj}}^i - \boldsymbol{\mu}_{\text{obj}}^j)^\top \right)\widetilde{\mathbf{V}}^j \mathbf{V}^i + (\boldsymbol{\mu}_{\text{obj}}^i - \boldsymbol{\mu}_{\text{obj}}^j)^\top \widetilde{\mathbf{V}}^j \widetilde{\mathbf{u}}_3^i((\mathbf{V}^i)^\top \widetilde{\mathbf{V}}^j \mathbf{V}^i - \mathbf{I}),$$

where $\text{sym}(\mathbf{A})$ denotes for any matrix $A$ the operation $\frac{1}{2}(\mathbf{A} + \mathbf{A}^\top)$.

To summarize, the approximate WSFF adapted for our setting is

$$\mathcal{B}^i = \left( \sum_{j \in \text{KNN}(i)} m_j \chi_\epsilon(\|\boldsymbol{\mu}_{\text{obj}}^i - \boldsymbol{\mu}_{\text{obj}}^j\|) \right)^{-1} \sum_{j \in \text{KNN}(i)} \frac{m_j \Upsilon'_\epsilon(\|\boldsymbol{\mu}_{\text{obj}}^i - \boldsymbol{\mu}_{\text{obj}}^j\|)}{3\|\boldsymbol{\mu}_{\text{obj}}^i - \boldsymbol{\mu}_{\text{obj}}^j\|} \mathbf{B}^{i,j}, \tag{38}$$

where we relabel the original kernel functions as $\chi_\epsilon, \Upsilon'_\epsilon$ that depend on the approximation scale $\epsilon$.

## E  MANIFOLD STRUCTURE ESTIMATION FOR GAUSSIAN SPLATTING

Based on local manifold assumption (i.e., Definition 2.1), we aim to estimate multiple key geometric properties (e.g., tangent vector) of the spatial object that hides behind 3D Gaussians $\{\mathcal{G}_{\text{obj}}^i\}_{i \in \mathcal{I}}$. Such geometric information can be used to improve Gaussian splatting, as shown in Sec. 3.

There are several types of classical frameworks for estimating manifold structures from discrete data. For example, moving least squares (Lancaster & Salkauskas, 1981) and heat equations (van den Berg & Le Gall, 1994; Coifman & Lafon, 2006). Following the direction of heat equations, we provide a full derivation procedure that shows how to practically estimate the manifold-related geometric quantities mentioned in Sec. 2.2, considering both the positions and shapes of Gaussian primitives in our problem setting. While this part should have overlaps with some previous works in this classical direction, our main goal here is to make our method self-contained, providing practical calculation guidance for future research in computer vision and graphics.

### E.1 TANGENT VECTOR ESTIMATION

A 3D Gaussian $\mathcal{G}_{\text{obj}}^i$ essentially characterizes an uncertain point $\mathbf{q} = \boldsymbol{\mu}_{\text{obj}}^i$ on the object surface, with a covariance ellipse as $\boldsymbol{\Sigma}_{\text{obj}}^i$. We will first study the estimation of tangent space $\mathcal{T}_{\mathbf{q}}\mathcal{M}$, which is the basis of computing other geometric quantities (e.g., normal vectors).

#### E.1.1 DEFINITION OF CURVE-DERIVED TANGENT VECTORS

From the viewpoint of intrinsic geometry (Lee, 2018), the tangent vector $\mathbf{v}_\star^i \in \mathcal{T}_{\mathbf{q}}\mathcal{M}$ at an arbitrary point $\mathbf{q} = \boldsymbol{\mu}_{\text{obj}}^i$ on the manifold $\mathcal{M}$ can be defined by a curve $\boldsymbol{\gamma}_\star^i$ that passes through the point. This definition in a strict sense relies on the concept of charts: which invertibly map a local manifold to the Euclidean space, as the manifold $\mathcal{M}$ might be curved (i.e., non-zero curvature $\mathcal{R}$). In the scenario of Gaussian splatting, the potential manifold $\mathcal{M}$ naturally resides in an ambient space $\mathcal{E} = \mathbb{R}^3$, so that we can simplify the definition as below.

**Definition E.1** (Curve-based Tangent Vector Formulation). For an arbitrary vector $\mathbf{v}_\star^i$ that is tangent to the embedded manifold $\mathcal{M} \subset \mathcal{E}$ at a point $\boldsymbol{\mu}_{\text{obj}}^i \in \mathcal{M}$, there always exits a continuous curve $\boldsymbol{\gamma}_\star^i : \mathbb{R} \to \mathcal{M}$ that satisfies two conditions:

$$\boldsymbol{\gamma}_\star^i(0) = \boldsymbol{\mu}_{\text{obj}}^i, \quad \partial_s \boldsymbol{\gamma}_\star^i(s)|_{s=0} = \mathbf{v}_\star^i. \tag{39}$$

Here the differential operator $\partial_s$ is inherited from the Euclidean space $\mathcal{E}$.

The significance of this definition is that we can convert the estimation of tangent vector $\mathbf{v}_\star^i$ to the approximation of curve $\boldsymbol{\gamma}_\star^i(s)$ on manifold $\mathcal{M}$. The curve in differential calculus is zero-order, and thus more tractable than the first-order tangent vector for a discrete point cloud.

There are infinitely many curves that can derive a tangent vector $\mathbf{v}_\star^i \in \mathcal{T}_{\mathbf{q}}\mathcal{M}$ at point $\mathbf{q} = \boldsymbol{\mu}_{\text{obj}}^i$. In the case of flat space, a natural choice is the coordinate curve $\boldsymbol{\gamma}_d^i$ that centers at point $\boldsymbol{\mu}_{\text{obj}}^i$ and moves along some dimension $d \in \{1, 2, D_{\mathcal{E}} = 3\}$. Formally speaking, this type of curve satisfies both Eq. (39) and a new condition as

$$\phi_{d'}(\boldsymbol{\gamma}_d^i(s)) = \phi_{d'}(\boldsymbol{\gamma}_d^i(0)) = \phi_{d'}(\boldsymbol{\mu}_{\text{obj}}^i), \forall d' \neq d, s \in \mathbb{R}, \tag{40}$$

where operation $\phi_{d'}(\cdot) = [\cdot]_{d'}$ is to takes out the $d'$-th coordinate value from the input vector. If the manifold $\mathcal{M}$ is curved, the above condition needs to be generalized as

$$\boldsymbol{\gamma}_d^i = \operatorname*{arg\,max}_{\boldsymbol{\gamma}_\star^i : \|\mathbf{v}_\star^i\| = c_d} \left( \partial_s \phi_d(\boldsymbol{\gamma}_\star^i(s))|_{s=0} \right), \tag{41}$$

where $c_d \in \mathbb{R}^+$ is an arbitrary positive constant. This equation means that the curve heads in the direction of maximizing its value at the $d$-th dimension. For the tangent vector derived by such a coordinate curve $\boldsymbol{\gamma}_d^i$, we denote it as $\mathbf{v}_d^i$.

#### E.1.2 ANALYSIS OF THE TANGENT SPACE STRUCTURE

For any point $\mathbf{q} = \boldsymbol{\mu}_{\text{obj}}^i \in \mathcal{M}$, the set of tangent vectors $\{\mathbf{v}_d^i\}_{1 \leqslant d \leqslant D_{\mathcal{E}}}$ derived by coordinate curve $\boldsymbol{\gamma}_\star^i(s)$ has a size of $D_{\mathcal{E}} = 3$, which is larger than the tangent space dimension: $D_{\mathcal{T}_{\mathbf{q}}\mathcal{M}} = D_{\mathcal{M}} < 3$. While this inequality is necessary, it is not sufficient to guarantee that the curve-derived vectors $\{\mathbf{v}_d^i\}_d$ span the entire tangent space $\mathcal{T}_{\mathbf{q}}\mathcal{M}$. A counter-example is that any two vectors differ only by a scale factor, which cannot form a 2D plane.

From the perspective of linear algebra (Roman et al., 2005), to determine the rank of vector set $\{\mathbf{v}_d^i\}_{1 \leqslant d \leqslant D_{\mathcal{E}}}$ and extract independent components, it is critical to study the relation between two tangent vectors $\mathbf{v}_{d_1}^i, \mathbf{v}_{d_2}^i, d_1 \neq d_2$: inner product $g_{\mathbf{q}}$. In this regard, we construct the kernel matrix $\mathcal{K}^i$ of inner product $g_{\mathbf{q}}(\mathbf{v}_{d_1}^i, \mathbf{v}_{d_2}^i)$, and confirm that the tangent space $\mathcal{T}_{\mathbf{q}}$ is indeed spanned by curve-derived vectors $\{\mathbf{v}_d^i\}_d$. The details are as follows.

**Proposition E.2** (Tangential Kernel Analysis). *For an arbitrary point $\mathbf{q} = \boldsymbol{\mu}_{\text{obj}}^i$ on the embedded Riemannian manifold $\mathcal{M} \subset \mathcal{E}$, the collection of curve-derived vectors $\{\mathbf{v}_d^i\}_{1 \leqslant d \leqslant D_{\mathcal{E}}}$ span the entire*

tangent space $\mathcal{T}_{\mathbf{q}}\mathcal{M}$, and the inner products between pairs of these vectors: $g_{\mathbf{q}}(\mathbf{v}_{d_1}^i, \mathbf{v}_{d_2}^i), 1 \leqslant d_1, d_2 \leqslant D_{\mathcal{E}} = 3$, form a tangential kernel matrix $\mathcal{K}^i$ as

$$\mathcal{K}^i := \begin{Bmatrix} g_{\mathbf{q}}(\mathbf{v}_1^i, \mathbf{v}_1^i) & g_{\mathbf{q}}(\mathbf{v}_1^i, \mathbf{v}_2^i) & g_{\mathbf{q}}(\mathbf{v}_1^i, \mathbf{v}_3^i) \\ g_{\mathbf{q}}(\mathbf{v}_2^i, \mathbf{v}_1^i) & g_{\mathbf{q}}(\mathbf{v}_2^i, \mathbf{v}_2^i) & g_{\mathbf{q}}(\mathbf{v}_2^i, \mathbf{v}_3^i) \\ g_{\mathbf{q}}(\mathbf{v}_3^i, \mathbf{v}_1^i) & g_{\mathbf{q}}(\mathbf{v}_3^i, \mathbf{v}_2^i) & g_{\mathbf{q}}(\mathbf{v}_3^i, \mathbf{v}_3^i) \end{Bmatrix}, \tag{42}$$

with its eigenvectors $\{\mathbf{u}_d^i\}_{1 \leqslant d \leqslant D_{\mathcal{M}}}$ that are paired with a positive eigenvalue $\lambda_d^i > 0$ constituting the basis of tangent space $\mathcal{T}_{\mathbf{q}}\mathcal{M}$.

*Proof.* The proof is provided in Appendix E.4. $\qquad\square$

We can see that the entire tangent space $\mathcal{T}_{\mathbf{q}}\mathcal{M}$ is accessible once we can construct the kernel matrix $\mathcal{K}^i$. More significantly, we show that the unknown entries of this matrix can be easily computed based on the zero-order curve $\gamma_d^i$ that defines the first-order tangent vector $\mathbf{v}_d^i$.

**Proposition E.3** (Curve-derived Riemannian Metric). *For an embedded manifold $\mathcal{M} \hookrightarrow \mathcal{E}$ and the coordinate-derived tangent vectors $\{\mathbf{v}_d^i\}_{1 \leqslant d \leqslant D_{\mathcal{E}}}$ (as formulated in Definition E.1 and Eq. (41)) at any point $\mathbf{q} = \boldsymbol{\mu}_{\mathrm{obj}}^i \in \mathcal{M}$, the metric $g_{\mathbf{q}}$ that takes two tangent vectors $\mathbf{v}_{d_1}^i, \mathbf{v}_{d_2}^i, 1 \leqslant d_1, d_2 \leqslant D_{\mathcal{E}}$ as its arguments can be expressed as*

$$g_{\mathbf{q}}(\mathbf{v}_{d_1}^i, \mathbf{v}_{d_2}^i) = \frac{1}{2}\mathrm{Lbnz}_\Delta[\phi_{d_1}, \phi_{d_2}](\mathbf{q}), \tag{43}$$

*where $\Delta$ denotes the Laplacian operator on the function space $\mathcal{C}^\infty(\mathcal{M})$, and $\mathrm{Lbnz}_\square$ characterizes how much it violates the Leibniz rule: $\square[\phi_{d_1}\phi_{d_2}] - \phi_{d_1}\square[\phi_{d_2}] - \phi_{d_2}\square[\phi_{d_1}]$.*

*Proof.* The proof is provided in Appendix E.5. $\qquad\square$

The significance of this conclusion is a special expression of metric $g_{\mathbf{q}}$ that only depends on the accessible zero-order curve coordinate (e.g., $\phi_{d_1}(\mathbf{q}) = [\gamma_{d_1}^i(0)]_{d_1} = [\boldsymbol{\mu}_{\mathrm{obj}}^i]_{d_1}$), without involving the intractable first-order tangent vectors (e.g., $\mathbf{v}_{d_2}^i$). While this expression still introduces a new unknown term: Laplacian operator $\Delta$, it can be efficiently estimated from the raw data $\{\mathcal{G}_{\mathrm{obj}}^i\}_{i \in \mathcal{I}}$ through many mature techniques (Perry & Mahoney, 2011; Cheng & Yang, 2005; Chuang et al., 2009; Xu & Dunn, 2019) in the literature.

### E.1.3 LAPLACIAN OPERATOR ESTIMATION

The Laplacian operator $\Delta$ in the metric reformulation (i.e., Eq. (43)) is a type of linear operator $\mathscr{L}$ in the function space $\mathcal{C}^\infty(\mathcal{M})$, satisfying

$$\mathscr{L}[af + bh] = a\mathscr{L}[f] + b\mathscr{L}[h], \forall a, b \in \mathbb{R}, \forall f, g \in \mathcal{C}^\infty(\mathcal{M}). \tag{44}$$

The estimation of linear operator $\mathscr{L}$ on discrete data points is a well-studied field. In the same spirit as the Feynman-Kac formula (Bertini & Cancrini, 1995), we adopt the following Monte Carlo approximation to efficiently evaluating the linear operator $\mathscr{L}[f], f \in \mathcal{C}^\infty(\mathcal{M})$.

**Proposition E.4** (Markov-chain Linear Operator Estimation). *Suppose that the Gaussian centers $\{\mathbf{c}_{\mathrm{obj}}^i\}_{i \in \mathcal{I}}$ are uniformly sampled from the Riemannian manifold $\mathcal{M}$, and linear operator $\mathscr{L}$ is the infinitesimal generator of a homogeneous Markov chain. For any smooth function $f \in \mathcal{C}^\infty(\mathcal{M})$, the evaluation of operator $\mathscr{L}$ has an unbiased approximation:*

$$\mathscr{L}[f](\mathbf{q}) \approx \frac{C}{t|\mathcal{I}|} \sum_{i \in \mathcal{I}} \Big(f(\mathbf{c}_{\mathrm{obj}}^i) - f(\mathbf{q})\Big) k_t(\mathbf{q}, \mathbf{c}_{\mathrm{obj}}^i), \tag{45}$$

*as $t$ converges to $0$. Here $C > 0$ is a constant that only depends on the manifold $\mathcal{M}$, and $k_t(\cdot)$ is the transition kernel of the Markov chain.*

*Proof.* The proof is provided in Appendix E.6. $\qquad\square$

This conclusion is to estimate the linear operator $\mathscr{L}$ specified by its kernel $k_t$. For approximating the Laplacian operator $\mathscr{L} = \Delta$, we can set the kernel $k_t$ as Gaussian (Davies, 1989):

$$k_t(\mathbf{q}_1, \mathbf{q}_2) \propto \exp\left( -\frac{1}{t} \|\mathbf{q}_1 - \mathbf{q}_2\|_2^2 \right), \tag{46}$$

which is normalized by the integral over $\mathbf{q}_1$ or $\mathbf{q}_2$. In practical implementation, the sum operation $\sum$ of Eq. (45) will be time-consuming if it runs over a large index set $I$. As the Gaussian kernel $k_t$ exponentially decays with respect to the point distance $\| \cdot \|_2^2$, we can limit the set as a few nearest neighbors of the point $\mathbf{q}$ to improve efficiency and reduce noise.

Besides, a point set $\{\mathcal{G}_{\mathrm{obj}}^i\}_{i \in \mathcal{I}}$ might not be evenly distributed, with some sparse regions. This problem can be addressed by adopting a bandwidth-adaptive kernel:

$$\begin{cases} k_t(\mathbf{q}_1, \mathbf{q}_2) \propto \exp\left( -(\mathbf{q}_1 - \mathbf{q}_2)^\top \mathbf{\Lambda}_t(\mathbf{q}_1, \mathbf{q}_2)^{-1}(\mathbf{q}_1 - \mathbf{q}_2) \right) \\ \mathbf{\Lambda}_t(\mathbf{q}_1 = \mathbf{c}_{\mathrm{obj}}^i, \mathbf{q}_2 = \mathbf{c}_{\mathrm{obj}}^j) = t \mathbf{\Sigma}_{\mathrm{obj}}^i \mathbf{\Sigma}_{\mathrm{obj}}^j \end{cases}. \tag{47}$$

Intuitively, this kernel $k_t$ is more sensitive to the neighbor $\mathbf{q}_2$ of a point $\mathbf{q}_1 = \mathbf{c}_{\mathrm{obj}}^i$ if the corresponding covariance matrix $\mathbf{\Sigma}_{\mathrm{obj}}^i$ is of a large scale.

## E.2 DIMENSION AND NORMAL VECTOR ESTIMATIONS

Starting with the curve-derived tangent vectors $\{\mathbf{v}_d^i\}_{1 \leq d \leq D_{\mathcal{E}}}$ (as defined in Sec. E.1.1), we can easily induce the normal vector $\mathbf{n} \in \mathcal{N}_{\mathbf{q}}\mathcal{M}$ at any point $\mathbf{q} = \boldsymbol{\mu}_{\mathrm{obj}}^i$ on the manifold $\mathcal{M}$. For example, if the manifold is 2-dimensional: $D_{\mathcal{M}} = 2$, then the unit norm $\mathbf{n}$ is unique and we can derive it from any two tangent vectors $\mathbf{v}_{d_1}^i, \mathbf{v}_{d_2}^i, 1 \leq d_1, d_2 \leq D_{\mathcal{E}}$ that are not co-linear:

$$\mathbf{n} = (\mathbf{v}_{d_1}^i \times \mathbf{v}_{d_2}^i) / \|\mathbf{v}_{d_1}^i \times \mathbf{v}_{d_2}^i\|_2, \tag{48}$$

where $\times$ denotes the vector cross product in three dimensions ($\mathcal{D}_{\mathcal{E}} = 3$). More conveniently, both manifold dimension $D_{\mathcal{M}}$ and normal space $\mathcal{N}_{\mathbf{q}}\mathcal{M}$ are just side products from the kernel matrix decomposition in Theorem E.2. The details are below.

**Corollary E.5** (Induced Orthogonal Space). *For any point $\mathbf{q} = \boldsymbol{\mu}_{\mathrm{obj}}^i$ on the embedded Riemannian manifold $\mathcal{M} \hookrightarrow \mathcal{E}$, the number of positive eigenvalue $\lambda_d > 0$ of kernel matrix $\mathcal{K}^i$ (as specified by Eq. (42)) is equal to the manifold dimension $D_{\mathcal{M}}$. More notably, the eigenvectors $\{\mathbf{u}_d^i\}_{D_{\mathcal{M}}+1 \leq d \leq D_{\mathcal{E}}}$ that correspond to zero eigenvalue $\lambda_d = 0$ form the basis of normal space $\mathcal{N}_{\mathbf{q}}\mathcal{M}$.*

*Proof.* The proof is provided in Appendix E.7. $\qquad\square$

While the tangential kernel matrix $\mathcal{K}^i$ seems to incur more computations than the cross product $\times$, the matrix is of a very small shape (i.e., $3 \times 3$) and its factorization can be analytical solved, instead of resorting to inefficient iteration algorithms (e.g., Alternating Least Squares (Comon et al., 2009)). Plus, to distinguish more from the notations of tangent basis $\{\mathbf{u}_d^i\}_{1 \leq d \leq D_{\mathcal{M}}}$, we might denote the normal basis as $\{\widetilde{\mathbf{u}}_d^i\}_{D_{\mathcal{M}}+1 \leq d \leq D_{\mathcal{E}}}$ to emphasize its directions.

## E.3 CURVATURE ESTIMATION

From the angle of extrinsic differential geometry, the curvature is a geometric quantity that characterizes how a hypersurface $\mathcal{M}$ bends within its ambient space $\mathcal{E}$. There are various types of curvature definitions (e.g., sectional curvature Sec), and they are all determined by the shape operator (Lee, 2018; Do Carmo, 2016) at every point $\mathbf{q} = \boldsymbol{\mu}_{\mathrm{obj}}^i$ on the maniold $\mathcal{M}$:

$$s^i(\mathbf{n}, \mathbf{u}_1, \mathbf{u}_2) : \mathcal{N}_{\mathbf{q}}\mathcal{M} \times \mathcal{T}_{\mathbf{q}}\mathcal{M} \times \mathcal{T}_{\mathbf{q}}\mathcal{M} \to \mathbb{R}, \tag{49}$$

which conditions on a normal vector $\mathbf{n}$ and maps a pair of tangent vectors $\mathbf{u}_1, \mathbf{u}_2$ to a scalar. In the case of Euclidean space $\mathcal{E}$, it is intuitive that this operator $\int^i$ is always valued as 0, indicating that the entire space is flat. For a curved manifold $\mathcal{M}$, the operator for curved-derived tangent vectors $\{\mathbf{v}_d^i\}_{1 \leq d \leq D_{\mathcal{E}}}$ can be expressed as follows.

**Proposition E.6** (Curve-derived Shape Operator). *For an arbitrary point $\mathbf{q} = \boldsymbol{\mu}_{\text{obj}}^i$ on the embedded Riemannian manifold $\mathcal{M} \hookrightarrow \mathcal{E}$ and any pair of curved-derived tangent vectors $\mathbf{v}_{d_1}^i, \mathbf{v}_{d_2}^i, 1 \leqslant d_1, d_2 \leqslant D_{\mathcal{E}}$, the shape operator $s^i$ for a normal vector $\mathbf{n} \in \mathcal{N}_{\mathbf{q}}\mathcal{M}$ takes the form as*

$$s^i(\mathbf{n}, \mathbf{v}_{d_1}^i, \mathbf{v}_{d_2}^i) = \frac{1}{8}\Big(\mathcal{A}[\phi_{d_1}, \eta, \phi_{d_2}](\mathbf{q}) + \mathcal{A}[\phi_{d_2}, \eta, \phi_{d_1}](\mathbf{q}) - \mathcal{A}[\eta, \phi_{d_1}, \phi_{d_2}](\mathbf{q})\Big), \qquad (50)$$

*where $\mathcal{A}[\cdot, \cdot, \cdot]$ is a second-order differential operator, with its value for any three functions $f_1, f_2, f_3$ as nested Leibniz rules $\text{Lbnz}_{\Delta}[f_1, \text{Lbnz}_{\Delta}[f_2, f_3]]$, and $\eta \in \mathcal{C}^{\infty}(\mathcal{M})$ is the height function, with its value at any point $\mathbf{q}' \in \mathcal{M}$ as the Euclidean inner product $\langle \mathbf{n}, \mathbf{q}' - \mathbf{q} \rangle_{\mathcal{E}}$.*

*Proof.* The proof is provided in Appendix E.8. □

In the same spirit as Proposition E.3, this conclusion permits us to compute the shape operator $s^i$ with only the zero-order curves $\{\mathbf{v}_d^i\}_{d \in [1, D_{\mathcal{E}}]}$. The Laplacian operator is typically unknown as in the case of tangent space estimation, and we can get it from Proposition E.4.

With the shape operator $s^i$, we can compute the principal curvatures of manifold $\mathcal{M}$ through matrix decomposition. The formal conclusion is as follows.

**Theorem E.7** (Shape Operator Factorization). *For any point $\mathbf{q} = \boldsymbol{\mu}_{\text{obj}}^i$ on the embedded Riemannian manifold $\mathcal{M} \hookrightarrow \mathcal{E}$ and a fixed normal vector $\mathbf{n} \in \mathcal{N}_{\mathbf{q}}\mathcal{M}$, the matrix:*

$$\boldsymbol{\mathcal{S}}^i := \{s^i(\mathbf{n}, \mathbf{u}_{d_1}^i, \mathbf{u}_{d_2}^i)\}_{1 \leqslant d_1, d_2 \leqslant D_{\mathcal{M}}}, \qquad (51)$$

*formed by the shape operator $s^i$ and orthonormal tangent vectors $\boldsymbol{\mathcal{U}}^i = \{\mathbf{u}_d^i\}_{1 \leqslant d \leqslant D_{\mathcal{M}}}$, is with eigenvalue $\tau_d^i, 1 \leqslant d \leqslant D_{\mathcal{M}}$ corresponding to the curvature value and realigned eigenvector $\tilde{\mathbf{w}}_d^i := \boldsymbol{\mathcal{U}}^i \mathbf{w}_d^i, 1 \leqslant d \leqslant D_{\mathcal{M}}$ associated with the principle direction.*

*Proof.* The proof is provided in Appendix E.9. □

The dimension $D_{\mathcal{M}}$ in 3D rendering is typically small than 3, so the eigen-decomposition of matrix $\boldsymbol{\mathcal{S}}^i$ can be solved even analytically. The orthonormal matrix $\boldsymbol{\mathcal{U}}^i$ involved in computation can also be obtained from previous Proposition E.2.

### E.4 PROOF: ESTIMATION OF THE TANGENT SPACE

The geometric analysis in this part is divided into three parts. We will first derive some basic terms (e.g., metric) in terms of embedded Riemannian geometry. Then, we will characterize the set of coordinate-derived tangent vectors $\{\mathbf{v}_d^i\}_{1 \leqslant d \leqslant D_{\mathcal{E}}}$ and their tangential kernel matrix $\mathcal{K}^i$.

#### E.4.1 PREPARATIONS

Let us first respectively denote the connections of Riemannian manifold $\mathcal{M}$ and Euclidean space $\mathcal{E}$ as $\nabla^{\mathcal{M}}, \nabla^{\mathcal{E}}$. Given that the manifold $\mathcal{M}$ is embedded in the ambient space $\mathcal{E}$, an important conclusion in differential geometry (Lee, 2018) is that

$$\nabla_{\mathbf{U}_1}^{\mathcal{M}} \mathbf{U}_2 = (\nabla_{\mathbf{U}_1}^{\mathcal{E}} \mathbf{U}_2)^{\pi}, \forall \mathbf{U}_1, \mathbf{U}_2 \in \mathcal{V}. \qquad (52)$$

Another key fact is that the metric $g_{\mathbf{q}}^{\mathcal{M}}$ of any tangent space $\mathcal{T}_{\mathbf{q}}\mathcal{M}$ can be induced by the inner product $g_{\mathbf{q}}^{\mathcal{E}}$ of the ambient space $\mathcal{E}$ as

$$g_{\mathbf{q}}^{\mathcal{M}}(\mathbf{u}_1, \mathbf{u}_2) = g_{\iota(\mathbf{q})}^{\mathcal{E}}(\partial \iota_{\mathbf{q}}[\mathbf{u}_1], \partial \iota_{\mathbf{q}}[\mathbf{u}_2]) = g_{\mathbf{q}}^{\mathcal{E}}(\mathbf{u}_1, \mathbf{u}_2) = \mathbf{u}_1^{\top} \mathbf{u}_2, \qquad (53)$$

where $\mathbf{u}_1, \mathbf{u}_2 \in \mathcal{T}_{\mathbf{q}}\mathcal{M}, \iota : \mathcal{M} \hookrightarrow \mathcal{E}$ denotes the inclusion map, and $\partial \iota_{\mathbf{q}}[\cdot]$ represents the pushforward operation at point $\mathbf{q}$. With the above two conclusions, we can relate the gradient operator $\text{grad}^{\mathcal{M}}$ of the embedded manifold $\mathcal{M}$ to that $\text{grad}^{\mathcal{E}}$ of the Euclidean space $\mathcal{E}$. For any function $f$ defined on the manifold $\mathcal{M}$, its gradient $\text{grad}^{\mathcal{M}} f$ is formulated as

$$\partial_{\mathbf{u}}^{\mathcal{M}} f(\mathbf{q}) = g_{\mathbf{q}}^{\mathcal{M}}(\text{grad}_{\mathbf{q}}^{\mathcal{M}} f, \mathbf{u}) = (\text{grad}_{\mathbf{q}}^{\mathcal{M}} f)^{\top} \mathbf{u}, \forall \mathbf{u} \in \mathcal{T}_{\mathbf{q}}\mathcal{M}, \qquad (54)$$

where the first term $\partial_{\mathbf{u}}^{\mathcal{M}} f(\mathbf{q})$ represents the derivative of function $f$ following the direction $\mathbf{u}$. The same conclusion applies to the Euclidean space $\mathcal{E}$ as

$$\partial_{\mathbf{u}}^{\mathcal{E}} f(\mathbf{q}) = g_{\mathbf{q}}^{\mathcal{E}}(\text{grad}_{\mathbf{q}}^{\mathcal{E}} f, \mathbf{u}) = (\text{grad}_{\mathbf{q}}^{\mathcal{E}} f)^{\top} \mathbf{u}, \forall \mathbf{u} \in \mathcal{E}. \tag{55}$$

Since the manifold $\mathcal{M}$ is embedded in the Euclidean space $\mathcal{E}$, the tangent space $\mathcal{T}_{\mathbf{q}}\mathcal{M}$ (i.e., a plane that is tangent to the manifold $\mathcal{M}$ at point $\mathbf{q}$) is also a subset of ambient space $\mathcal{E}$. For any tangent vector $\mathbf{u} \in \mathcal{T}_{\mathbf{q}}\mathcal{M} \subset \mathcal{E}$, we can prove that the two differential operations $\partial_{\mathbf{u}}^{\mathcal{M}}, \partial_{\mathbf{u}}^{\mathcal{E}}$ are equivalent based on Definition E.1. Suppose that some smooth curve $\boldsymbol{\gamma} : \mathbb{R} \to \mathcal{M}$ on the manifold $\mathcal{M}$ derives this tangent vector $\mathbf{u}$, which means

$$\boldsymbol{\gamma}(0) = \mathbf{q}, \quad \lim_{\epsilon \to 0} \frac{\boldsymbol{\gamma}(\epsilon) - \boldsymbol{\gamma}(0)}{\epsilon} = \mathbf{u}, \tag{56}$$

then the derivative $\partial_{\mathbf{u}}^{\mathcal{M}} f(\mathbf{q})$ of a smooth function $f$ can be formulated as

$$\partial_{\mathbf{u}}^{\mathcal{M}} f(\mathbf{q}) = \lim_{\epsilon \to 0} \frac{f(\boldsymbol{\gamma}(\epsilon)) - f(\boldsymbol{\gamma}(0))}{\epsilon}. \tag{57}$$

Because the manifold-valued curve $\boldsymbol{\gamma} \subseteq \mathcal{M}$ also resides in the ambient space $\mathcal{E}$, then the expression at the right hand side can also define the term $\partial_{\mathbf{u}}^{\mathcal{E}} f(\mathbf{q})$. Therefore, we have

$$(\text{grad}_{\mathbf{q}}^{\mathcal{M}} f)^{\top} \mathbf{u} = \partial_{\mathbf{u}}^{\mathcal{M}} f(\mathbf{q}) = \partial_{\mathbf{u}}^{\mathcal{E}} f(\mathbf{q}) = (\text{grad}_{\mathbf{q}}^{\mathcal{E}} f)^{\top} \mathbf{u}. \tag{58}$$

Because this equality holds for any tangent vector $\mathbf{u} \in \mathcal{T}_{\mathbf{q}}\mathcal{M}$, then we can infer that the leftmost term $\text{grad}_{\mathbf{q}}^{\mathcal{M}} f$ is the projection of the rightmost term $\text{grad}_{\mathbf{q}}^{\mathcal{E}} f$ on the tangent plane $\mathcal{T}_{\mathbf{q}}\mathcal{M}$. More formally, we specify this relation as

$$\text{grad}_{\mathbf{q}}^{\mathcal{M}} f = (\text{grad}_{\mathbf{q}}^{\mathcal{E}} f)^{\pi}. \tag{59}$$

The projection operation $\pi$ here is defined in terms of the square norm $\| \cdot \|_2^2$.

Secondly, let us derive the concrete form of coordinate-derived tangent vectors $\{\mathbf{v}_d^i\}_{1 \leqslant d \leqslant D_{\mathcal{E}}}$. For some coordinate dimension $d$, this type of tangent vector $\mathbf{v}_d^i$ represents the direction that the function $\phi_d(\cdot) = [\cdot]_d$ increases the most. For a flat space $\mathcal{E}$, the non-unit version $\tilde{\mathbf{v}}_d^i$ of the coordinate-derived vector can be computed with vanilla differential calculus:

$$\tilde{\mathbf{v}}_d^i = \text{grad}_{\mathbf{q}}^{\mathcal{E}} [\cdot]_d = \lim_{\epsilon \to 0} \frac{\mathbf{q} + \epsilon \cdot \mathbf{e}_d - \mathbf{q}}{\epsilon} = \mathbf{e}_d, \tag{60}$$

where $\mathbf{e}_d$ represents a basis vector in the Euclidean space $\mathcal{E}$ that has an entry as 1 at the $d$-th dimension and is with 0 in all other entries. For a curved embedded manifold $\mathcal{M} \hookrightarrow \mathcal{E}$, the gradient projection formula (i.e., Eq. (59)) suggests that the term $\tilde{\mathbf{v}}_d^i$ can be generalized as

$$\tilde{\mathbf{v}}_d^i = \text{grad}_{\mathbf{q}}^{\mathcal{M}} [\cdot]_d = (\text{grad}_{\mathbf{q}}^{\mathcal{E}} [\cdot]_d)^{\pi} = \mathbf{e}_d^{\pi}. \tag{61}$$

The last projection term $\mathbf{e}_d^{\pi}$ can be further expanded as

$$\mathbf{e}_d - \sum_{D_{\mathcal{M}} < d' \leqslant D_{\mathcal{E}}} g_{\mathbf{q}}^{\mathcal{M}}(\mathbf{n}_{d'}, \mathbf{e}_d) \mathbf{n}_{d'} = \mathbf{e}_d - \sum_{D_{\mathcal{M}} < d' \leqslant D_{\mathcal{E}}} (\mathbf{n}_{d'}^{\top} \mathbf{e}_d) \mathbf{n}_{d'}, \tag{62}$$

where $\{\mathbf{n}_{d'}\}_{d' \in (D_{\mathcal{M}}, D_{\mathcal{E}}]}$ is a orthonormal basis in the normal space $\mathcal{N}_{\mathbf{q}}\mathcal{M}$. Combining the above two equations, we get a concrete expression for the vector $\tilde{\mathbf{v}}_d^i$ as

$$\tilde{\mathbf{v}}_d^i = \mathbf{e}_d - \sum_{D_{\mathcal{M}} < d' \leqslant D_{\mathcal{E}}} (\mathbf{n}_{d'}^{\top} \mathbf{e}_d) \mathbf{n}_{d'}, \tag{63}$$

with its normalized version $\tilde{\mathbf{v}}_d^i / \|\tilde{\mathbf{v}}_d^i\|_2$ as the coordinate-derived tangent vector $\mathbf{v}_d^i$.

### E.4.2 RANK DETERMINATION

Then, we prove that coordinate-derived tangent vectors $\{\mathbf{v}_d^i\}_{1 \leqslant d \leqslant D_{\mathcal{E}}}$ span the entire tangent space $\mathcal{T}_{\mathbf{q}}\mathcal{M}$ at point $\mathbf{q} = \boldsymbol{\mu}_{\text{obj}}^i \in \mathcal{M}$. It suffices to verify this fact for the collection of unnormalized vectors $\{\tilde{\mathbf{v}}_d^i\}_{1 \leqslant d \leqslant D_{\mathcal{E}}}$. Note that the projection operation $\pi$ is linear, so it can be represented by a matrix. In terms of Eq. (63), we can reformulate it as

$$\tilde{\mathbf{v}}_d^i = \mathbf{I} \mathbf{e}_d - \sum_{D_{\mathcal{M}} < d' \leqslant D_{\mathcal{E}}} (\mathbf{n}_{d'} \mathbf{n}_{d'}^{\top}) \mathbf{e}_d = \left( \mathbf{I} - \sum_{D_{\mathcal{M}} < d' \leqslant D_{\mathcal{E}}} (\mathbf{n}_{d'} \mathbf{n}_{d'}^{\top}) \right) \mathbf{e}_d = \mathbf{L} \mathbf{e}_d, \tag{64}$$

where $\mathbf{I}$ denotes the identity matrix and $\mathbf{L}$ is the concrete form of projection $\pi$. This matrix $\mathbf{L}$ is of rank $D_{\mathcal{M}}$. Specifically, for any normal vector $\mathbf{n}_\star, \star \in (D_{\mathcal{M}}, D_{\mathcal{E}}]$, we have

$$\mathbf{L}\mathbf{n}_\star = \Big(\mathbf{I} - \sum_{D_{\mathcal{M}} < d' \leqslant D_{\mathcal{E}}} (\mathbf{n}_{d'}\mathbf{n}_{d'}^\top)\Big)\mathbf{n}_\star = \mathbf{n}_\star - \sum_{D_{\mathcal{M}} < d' \leqslant D_{\mathcal{E}}} (\mathbf{n}_{d'}^\top \mathbf{n}_\star)\mathbf{n}_{d'} = \mathbf{n}_\star - \mathbf{n}_\star = \mathbf{0}. \quad (65)$$

For any tangent vector $\mathbf{u} \in \mathcal{T}_\mathbf{q}\mathcal{M}$, we also have

$$\mathbf{L}\mathbf{u} = \Big(\mathbf{I} - \sum_{D_{\mathcal{M}} < d' \leqslant D_{\mathcal{E}}} (\mathbf{n}_{d'}\mathbf{n}_{d'}^\top)\Big)\mathbf{u} = \mathbf{u} - \sum_{D_{\mathcal{M}} < d' \leqslant D_{\mathcal{E}}} (\mathbf{n}_{d'}^\top \mathbf{u})\mathbf{n}_{d'} = \mathbf{u}. \quad (66)$$

Therefore, any normal vector $\mathbf{n}_\star$ is an eigenvector of matrix $\mathbf{L}$ with the eigenvalue as 0, and any tangent vector is also an eigenvector with the eigenvalue as 1. Clearly, the projection matrix $\mathbf{L}$ has a rank as $D_{\mathcal{M}}$. Unnormalized vectors $\{\widetilde{\mathbf{v}}_d^i\}_{1 \leqslant d \leqslant D_{\mathcal{E}}}$ are actually computed from the Euclidean basis $\{\mathbf{e}_d\}_{1 \leqslant d \leqslant D_{\mathcal{E}}}$ with respect to the projection matrix $\mathbf{L}$. Given that the basis $\{\mathbf{e}_d\}_d$ are mutually independent and the matrix $\mathbf{L}$ is of rank $D_{\mathcal{M}}$, the vectors $\{\widetilde{\mathbf{v}}_d^i\}_d$ must have a rank as $D_{\mathcal{M}}$, and thus they span the tangent plane $\mathcal{T}_\mathbf{q}\mathcal{M}$.

The last claim might not be very straightforward. We provide a simple proof for it to end this subsection. By contradiction, suppose that the set of vectors $\{\widetilde{\mathbf{v}}_d^i\}_{1 \leqslant d \leqslant D_{\mathcal{E}}}$ are not able to span the entire tangent plane $\mathcal{T}_\mathbf{q}\mathcal{M}$, then there exists a non-zero tangent vector $\mathbf{u} \in \mathcal{T}_\mathbf{q}\mathcal{M}$ that is orthogonal to them: $g_\mathbf{q}^\mathcal{M}(\mathbf{u}, \widetilde{\mathbf{v}}_d^i) = 0, \forall d \in [1, D_{\mathcal{E}}]$. Equivalently, we have

$$0 = \mathbf{u}^\top \widetilde{\mathbf{v}}_d^i = \mathbf{u}^\top (\mathbf{L}\mathbf{e}_d) = (\mathbf{u}^\top \mathbf{L})\mathbf{e}_d. \quad (67)$$

Since this equality is true for any Euclidean basis $\mathbf{e}_d$, then we can infer that $\mathbf{u}^\top \mathbf{L} = \mathbf{0}^\top$. Note that the matrix $\mathbf{L}$ is symmetric, we have

$$\mathbf{0} = \mathbf{L}^\top \mathbf{u} = \mathbf{L}\mathbf{u}, \quad (68)$$

which contradicts Eq. (66). Hence, the set $\{\widetilde{\mathbf{v}}_d^i\}_d$ must span the whole space $\mathcal{T}_\mathbf{q}\mathcal{M}$.

### E.4.3 KERNEL MATRIX ANALYSIS

Lastly, let us study the part with kernel matrix $\mathcal{K}^i$. Note that the constant $c_d$ in Eq. (41) can be set as an arbitrary number, so we do not distinguish the unit vector $\mathbf{v}_j^i, j \in [1, D_{\mathcal{E}}]$ from its unnormalized version $\widetilde{\mathbf{v}}_j^i$ in this subsection. A key property of this matrix is positive semidefiniteness. For any vector $\mathbf{u}^i = [u_1^i, u_2^i, \cdots, u_{D_{\mathcal{M}}}^i]^\top \in \mathbb{R}^{D_{\mathcal{E}}}$, please note that

$$\begin{aligned}
(\mathbf{u}^i)^\top \mathcal{K}^i \mathbf{u}^i &= \sum_{1 \leqslant j,k \leqslant D_{\mathcal{E}}} g_\mathbf{q}^\mathcal{M}(\mathbf{v}_j^i, \mathbf{v}_k^i)u_j^i u_k^i = \sum_{1 \leqslant j \leqslant D_{\mathcal{E}}} u_j^i \Big(\sum_{1 \leqslant k \leqslant D_{\mathcal{E}}} u_k^i g_\mathbf{q}^\mathcal{M}(\mathbf{v}_j^i, \mathbf{v}_k^i)\Big) \\
&= \sum_{1 \leqslant j \leqslant D_{\mathcal{E}}} u_j^i g_\mathbf{q}^\mathcal{M}(\mathbf{v}_j^i, \sum_{1 \leqslant k \leqslant D_{\mathcal{E}}} u_k^i \mathbf{v}_k^i) = g_\mathbf{q}^\mathcal{M}(\sum_{1 \leqslant j \leqslant D_{\mathcal{E}}} u_j^i \mathbf{v}_j^i, \sum_{1 \leqslant k \leqslant D_{\mathcal{E}}} u_k^i \mathbf{v}_k^i) \geqslant 0,
\end{aligned} \quad (69)$$

where the last inequality holds because the inner product is positive-definite. Therefore, the property is indeed true for kernel matrix $\mathcal{K}^i$. Furthermore, suppose this vector $\mathbf{u}^i$ is a non-zero eigenvector of the matrix $\mathcal{K}^i$, with the corresponding eigenvalue as $\lambda^i$, then we have

$$\mathcal{K}^i \mathbf{u}^i = \lambda^i \mathbf{u}^i. \quad (70)$$

Taking the Euclidean inner product with the same vector $\mathbf{u}^i$ on both sides, we get

$$(\mathbf{u}^i)^\top \mathcal{K}^i \mathbf{u}^i = \lambda^i (\mathbf{u}^i)^\top \mathbf{u}^i \implies \lambda^i = \frac{(\mathbf{u}^i)^\top \mathcal{K}^i \mathbf{u}^i}{(\mathbf{u}^i)^\top \mathbf{u}^i} \geqslant 0. \quad (71)$$

Therefore, the eigenvalue $\lambda^i$ of tangential kernel matrix $\mathcal{K}^i$ is always positive or zero.

For a pair of eigenvector $\mathbf{u}_\star^i \in \mathbb{R}^{D_{\mathcal{E}}}$ and eigenvalue $\lambda_\star^i \in \mathbb{R}$, we will show that this eigenvector corresponds to a tangent vector if the eigenvalue is positive (i.e., $\lambda_\star^i > 0$), otherwise it is paired with a normal vector. Let us first look into the second case. Based on Eq. (70), we have

$$0 = [0 \cdot \mathbf{u}_\star^i]_d = [\mathcal{K}^i \mathbf{u}_\star^i]_d = \sum_{1 \leqslant j \leqslant D_{\mathcal{E}}} g_\mathbf{q}^\mathcal{M}(\mathbf{v}_d^i, \mathbf{v}_j^i)u_{\star,j}^i = g_\mathbf{q}^\mathcal{M}(\mathbf{v}_j^i, \sum_{1 \leqslant j \leqslant D_{\mathcal{E}}} u_{\star,j}^i \mathbf{v}_j^i), \quad (72)$$

where $1 \leqslant d \leqslant D_{\mathcal{E}}$ and $u_{\star,j}^i = [\mathbf{u}_{\star}^i]_d$. As this equality holds for any subscript $d$ and the vector set $\{\mathbf{v}_d^i\}_{1 \leqslant d \leqslant D_{\mathcal{E}}}$ span the tangent space $\mathcal{T}_{\mathbf{q}}\mathcal{M}$, we can infer

$$\sum_{1 \leqslant j \leqslant D_{\mathcal{E}}} u_{\star,j}^i \mathbf{v}_j^i = \mathbf{0}. \tag{73}$$

By incorporating Eq. (63), we can expand the right hand side as

$$\mathbf{0} = \sum_{1 \leqslant j \leqslant D_{\mathcal{E}}} u_{\star,j}^i \Big( \mathbf{e}_j - \sum_{D_{\mathcal{M}} < j' \leqslant D_{\mathcal{E}}} (\mathbf{n}_{j'}^\top \mathbf{e}_j) \mathbf{n}_{j'} \Big). \tag{74}$$

By further reshaping the equation, we can get

$$\sum_{1 \leqslant j \leqslant D_{\mathcal{E}}} \sum_{D_{\mathcal{M}} < j' \leqslant D_{\mathcal{E}}} u_{\star,j}^i (\mathbf{n}_{j'}^\top \mathbf{e}_j) \mathbf{n}_{j'} = \sum_{1 \leqslant j \leqslant D_{\mathcal{E}}} u_{\star,j}^i \mathbf{e}_j = \mathbf{u}_{\star}^i, \tag{75}$$

indicating that the eigenvector $\mathbf{u}_{\star}^i$ with a zero eigenvalue $\lambda_{\star}^i = 0$ is a normal vector in the space $\mathcal{N}_{\mathbf{q}}\mathcal{M}$. On the other hand, note that any normal vector $\mathbf{u}_{\star}^i \in \mathcal{N}_{\mathbf{q}}\mathcal{M}$ makes Eq. (74) hold, so it is an eigenvector of matrix $\mathcal{K}^i$ with a zero eigenvalue. In light of this fact, we can infer that every eigenvector paired with a positive eigenvalue $\lambda_{\star}^i > 0$ is perpendicular to the normal space, and thus tangent to the manifold: $\mathbf{u}_{\star}^i \in \mathcal{T}_{\mathbf{q}}\mathcal{M}$, which proves the first case.

Finally, we show that the positive eigenvalue can only be 1. If $\lambda_{\star}^i > 0$, we have

$$\lambda_{\star}^i = \lambda_{\star}^i (\mathbf{u}_{\star}^i)^\top \mathbf{u}_{\star}^i = (\mathbf{u}_{\star}^i)^\top \mathcal{K}^i \mathbf{u}_{\star}^i = \sum_{1 \leqslant j,k \leqslant D_{\mathcal{E}}} g_{\mathbf{q}}^{\mathcal{M}} (\mathbf{v}_j^i, \mathbf{v}_k^i) u_{\star,j}^i u_{\star,k}^i$$

$$= g_{\mathbf{q}}^{\mathcal{M}} \Big( \sum_{1 \leqslant j \leqslant D_{\mathcal{E}}} u_{\star,j}^i \mathbf{v}_j^i, \sum_{1 \leqslant k \leqslant D_{\mathcal{E}}} u_{\star,k}^i \mathbf{v}_k^i \Big). \tag{76}$$

In this regard, there exists a unit tangent vector $\bar{\mathbf{u}}_{\star}^i$ such that

$$\sqrt{\lambda_{\star}^i} \bar{\mathbf{u}}_{\star}^i = \sum_{1 \leqslant d \leqslant D_{\mathcal{E}}} u_{\star,d}^i \mathbf{v}_d^i. \tag{77}$$

In terms of Eq. (70), the right hand side can be expanded as

$$\sqrt{\lambda_{\star}^i} \bar{\mathbf{u}}_{\star}^i = \sum_{1 \leqslant d \leqslant D_{\mathcal{E}}} u_{\star,d}^i \Big( \mathbf{e}_d - \sum_{D_{\mathcal{M}} < d' \leqslant D_{\mathcal{E}}} (\mathbf{n}_{d'}^\top \mathbf{e}_d) \mathbf{n}_{d'} \Big)$$

$$= \sum_{1 \leqslant d \leqslant D_{\mathcal{E}}} u_{\star,d}^i \mathbf{e}_d - \sum_{1 \leqslant d \leqslant D_{\mathcal{E}}} \sum_{D_{\mathcal{M}} < d' \leqslant D_{\mathcal{E}}} u_{\star,d}^i (\mathbf{n}_{d'}^\top \mathbf{e}_d) \mathbf{n}_{d'} \tag{78}$$

$$= \mathbf{u}_{\star}^i - \sum_{D_{\mathcal{M}} < d' \leqslant D_{\mathcal{E}}} (\mathbf{n}_{d'}^\top \mathbf{u}_{\star}^i) \mathbf{n}_{d'}.$$

Note that the last sum term should be zero, as previously we prove that $\mathbf{u}_{\star}^i \in \mathcal{T}_{\mathbf{q}}\mathcal{M}$. Therefore, we get $\sqrt{\lambda_{\star}^i} \bar{\mathbf{u}}_{\star}^i = \mathbf{u}_{\star}^i$, indicating that $\lambda_{\star}^i = 1$. The key here is that the vectors $\bar{\mathbf{u}}_{\star}^i, \mathbf{u}_{\star}^i$ in both sides are normalized: $\|\bar{\mathbf{u}}_{\star}^i\|_2 = 1, \|\mathbf{u}_{\star}^i\|_2 = 1$.

### E.5 Proof: Curve-based Formulation of the Riemannian Metric

To make our proof more readable, we will first derive the second-order form of the Riemannian metric for the Euclidean space, and then dive into the more general manifold.

#### E.5.1 Analysis on the Euclidean Space

Let us first study the metric form in the flat space, and then dive into the curved case. In the Euclidean case $\mathcal{E}$, the metric $g^{\mathcal{E}}(\mathbf{u}_1, \mathbf{u}_2)$ is the same everywhere: inner product $\mathbf{u}_1^\top \mathbf{u}_2 = \sum_i u_{1,i} u_{2,i}$, and the Laplacian operator $\Delta^{\mathcal{E}}$ is a sum of the second-order differential operators:

$$g^{\mathcal{E}}(\mathbf{u}_1, \mathbf{u}_2) = \mathbf{u}_1^\top \mathbf{u}_2 = \sum_{1 \leqslant i \leqslant D_{\mathcal{E}}} u_{1,i} u_{2,i}, \quad \Delta^{\mathcal{E}} = \sum_{1 \leqslant i \leqslant D_{\mathcal{E}}} \partial_{\mathbf{e}_i}^2, \tag{79}$$

where $\mathbf{e}_i$ represents the $i$-th Euclidean basis. For any two smooth functions $f, h : \mathcal{E} \to \mathbb{R}$, we iteratively apply the Leibniz rule as

$$
\begin{aligned}
\Delta^{\mathcal{E}}[fh] &= \sum_{1 \leqslant i \leqslant D_{\mathcal{E}}} \partial_{\mathbf{e}_i}^2(fh) = \sum_{1 \leqslant i \leqslant D_{\mathcal{E}}} \partial_{\mathbf{e}_i}(h\partial_{\mathbf{e}_i}f + f\partial_{\mathbf{e}_i}h) \\
&= \sum_{1 \leqslant i \leqslant D_{\mathcal{E}}} \partial_{\mathbf{e}_i}(h\partial_{\mathbf{e}_i}f) + \partial_{\mathbf{e}_i}(f\partial_{\mathbf{e}_i}h) \\
&= \sum_{1 \leqslant i \leqslant D_{\mathcal{E}}} \left( \partial_{\mathbf{e}_i}h\partial_{\mathbf{e}_i}f + h\partial_{\mathbf{e}_i}^2 f \right) + \sum_{1 \leqslant i \leqslant D_{\mathcal{E}}} \left( \partial_{\mathbf{e}_i}f\partial_{\mathbf{e}_i}h + f\partial_{\mathbf{e}_i}^2 h \right) \\
&= 2 \sum_{1 \leqslant i \leqslant D_{\mathcal{E}}} \partial_{\mathbf{e}_i}f\partial_{\mathbf{e}_i}h + h \sum_{1 \leqslant i \leqslant D_{\mathcal{E}}} \partial_{\mathbf{e}_i}^2 f + f \sum_{1 \leqslant i \leqslant D_{\mathcal{E}}} \partial_{\mathbf{e}_i}^2 h \\
&= 2g^{\mathcal{E}}(\mathrm{grad}^{\mathcal{E}}f, \mathrm{grad}^{\mathcal{E}}h) + h\Delta^{\mathcal{E}}[f] + f\Delta^{\mathcal{E}}[h],
\end{aligned}
\tag{80}
$$

By rearranging the terms, we arrive at an equation as

$$
g^{\mathcal{E}}(\mathrm{grad}^{\mathcal{E}}f, \mathrm{grad}^{\mathcal{E}}h) = \frac{1}{2}\Big( \Delta^{\mathcal{E}}[fh] - h\Delta^{\mathcal{E}}[f] - f\Delta^{\mathcal{E}}[h] \Big).
\tag{81}
$$

In essence, the right hand side measures how much the linear operator $\Delta^{\mathcal{E}}$ violates the Leibniz rule. Based on this equality, our previous conclusion (i.e., Eq. (60)) indicates that

$$
\begin{aligned}
g_{\mathbf{q}}^{\mathcal{E}}(\mathbf{v}_{d_1}^i, \mathbf{v}_{d_2}^i) &= g_{\mathbf{q}}^{\mathcal{E}}(\mathrm{grad}_{\mathbf{q}}^{\mathcal{E}}\phi_{d_1}, \mathrm{grad}_{\mathbf{q}}^{\mathcal{E}}\phi_{d_2}) \\
&= \frac{1}{2}\Big( \Delta^{\mathcal{E}}[\phi_{d_1}\phi_{d_2}](\mathbf{q}) - \phi_{d_1}(\mathbf{q})\Delta^{\mathcal{E}}[\phi_{d_2}](\mathbf{q}) - \phi_{d_2}(\mathbf{q})\Delta^{\mathcal{E}}[\phi_{d_1}](\mathbf{q}) \Big),
\end{aligned}
\tag{82}
$$

where $\mathbf{v}_{d_1}^i, \mathbf{v}_{d_2}^i$ are two coordinate-derived tangent vectors at point $\mathbf{q} = \boldsymbol{\mu}_{\mathrm{obj}}^i$.

Importantly, we can see that the final conclusion is similar to the well-known Leibniz rule. To simplify the notation, we specify the Leibniz operator $\mathrm{Lbnz}_{\square}$ as

$$
\mathrm{Lbnz}_{\square}[f_1, f_2] = \square[f_1 f_2] - f_1\square[f_2] - f_2\square[f_1],
\tag{83}
$$

while $\square$ is a certain linear operator. Under this scheme, the conclusion now is as

$$
g_{\mathbf{q}}^{\mathcal{E}}(\mathbf{v}_{d_1}^i, \mathbf{v}_{d_2}^i) = \frac{1}{2}\mathrm{Lbnz}_{\Delta^{\mathcal{E}}}[\phi_{d_1}, \phi_{d_2}](\mathbf{q}),
\tag{84}
$$

where $d_1, d_2$ are two arbitrary numbers from $\{1, 2, \cdots, D_{\mathcal{E}}\}$.

### E.5.2 ANALYSIS ON THE CURVED MANIFOLD

Then, for a curved manifold $\mathcal{M}$, the metric $g_{\mathbf{q}}^{\mathcal{M}}$ varies point by point in notation and we need to generalize the Laplacian operator. Specifically, a general definition of this operator is that it stands as the divergence $\mathrm{div}^{\mathcal{M}}$ of the gradient field $\mathrm{grad}^{\mathcal{M}}f$ of a function $f$:

$$
\Delta^{\mathcal{M}}[f] = \mathrm{div}^{\mathcal{M}}(\mathrm{grad}^{\mathcal{M}}f).
\tag{85}
$$

Let us begin with deriving the concrete form of gradient term $\mathrm{grad}^{\mathcal{M}}f$. Formally speaking, suppose that the basis fields are as $\{\mathbf{X}_d^{\mathcal{M}} : \mathbf{q} \mapsto \mathcal{T}_{\mathbf{q}}\mathcal{M}\}_{1 \leqslant d \leqslant D_{\mathcal{M}}}$ and the coefficient fields for this gradient term are as $\{b_d : \mathcal{M} \to \mathbb{R}\}_{1 \leqslant d \leqslant D_{\mathcal{M}}}$, then we can get the expanded form as

$$
\mathrm{grad}^{\mathcal{M}}f = \sum_{1 \leqslant d \leqslant D_{\mathcal{M}}} b_d\mathbf{X}_d.
\tag{86}
$$

In terms of Eq. (54), we further have

$$
\begin{aligned}
\partial_{\mathbf{X}_{d'}}^{\mathcal{M}}f = g^{\mathcal{M}}(\mathrm{grad}^{\mathcal{M}}f, \mathbf{X}_{d'}) &= \sum_{1 \leqslant d, d' \leqslant D_{\mathcal{M}}} b_d g^{\mathcal{M}}(\mathbf{X}_d, \mathbf{X}_{d'}) \\
&= \sum_{1 \leqslant d, d' \leqslant D_{\mathcal{M}}} b_d\bar{g}_{d,d'} = [\mathbf{b}^{\top}\bar{\mathbf{G}}]_{d'},
\end{aligned}
\tag{87}
$$

where $d'$ is an integer from $[1, D_{\mathcal{M}}]$ and $\bar{\mathbf{G}}$ is a 2-dimensional tensor of metric coefficients $\{\bar{g}_{d,d'} := g^{\mathcal{M}}(\mathbf{X}_d, \mathbf{X}_{d'})\}_{1 \leqslant d, d' \leqslant D_{\mathcal{M}}}$. By raising the metric index, we finally get

$$\sum_{1 \leqslant d' \leqslant D_{\mathcal{M}}} \partial^{\mathcal{M}}_{\mathbf{X}_{d'}} f \cdot \bar{g}^{d',d} = \sum_{1 \leqslant d' \leqslant D_{\mathcal{M}}} [\mathbf{b}^{\top} \bar{\mathbf{G}}]_{d'} \bar{g}^{d',d} = [\mathbf{b}^{\top} \bar{\mathbf{G}} \bar{\mathbf{G}}^{-1}]_d = b_d, \tag{88}$$

where $\bar{g}^{d',d}$ is the entry of inverse matrix $\bar{\mathbf{G}}^{-1}$ at the $d'$-th row and $d$-th column. For an arbitrary vector field $\mathbf{U} \in \mathcal{V}(\mathcal{M})$, the general definition of divergence operator $\mathrm{div}^{\mathcal{M}}$ relying on its expanded form $\sum_{1 \leqslant d \leqslant D_{\mathcal{M}}} u_d \mathbf{X}_d$ is as

$$\begin{aligned}
\mathrm{div}^{\mathcal{M}} \mathbf{U} &= \sum_{1 \leqslant j,k \leqslant D_{\mathcal{M}}} \bar{g}^{j,k} g^{\mathcal{M}}(\nabla^{\mathcal{M}}_{\mathbf{X}_j} \mathbf{U}, \mathbf{X}_k) \\
&= \sum_{1 \leqslant j,k \leqslant D_{\mathcal{M}}} \bar{g}^{j,k} g^{\mathcal{M}}\Big(\nabla^{\mathcal{M}}_{\mathbf{X}_j}\Big(\sum_{1 \leqslant d \leqslant D_{\mathcal{M}}} u_d \mathbf{X}_d\Big), \mathbf{X}_k\Big) \\
&= \sum_{1 \leqslant j,k \leqslant D_{\mathcal{M}}} \bar{g}^{j,k} g^{\mathcal{M}}\Big(\sum_{1 \leqslant d \leqslant D_{\mathcal{M}}} \Big(\partial^{\mathcal{M}}_{\mathbf{X}_j} u_d \cdot \mathbf{X}_d + u_d \nabla^{\mathcal{M}}_{\mathbf{X}_j} \mathbf{X}_d\Big), \mathbf{X}_k\Big) \\
&= \sum_{1 \leqslant j,k,d \leqslant D_{\mathcal{M}}} \bar{g}^{j,k}(\partial^{\mathcal{M}}_{\mathbf{X}_j} u_d) g^{\mathcal{M}}(\mathbf{X}_d, \mathbf{X}_k) + \sum_{1 \leqslant j,k,d \leqslant D_{\mathcal{M}}} \bar{g}^{j,k} u_d g^{\mathcal{M}}\Big(\nabla^{\mathcal{M}}_{\mathbf{X}_j} \mathbf{X}_d, \mathbf{X}_k\Big) \\
&= \sum_{1 \leqslant j,k,d \leqslant D_{\mathcal{M}}} \bar{g}^{j,k}(\partial^{\mathcal{M}}_{\mathbf{X}_j} u_d) \bar{g}_{d,k} + \sum_{1 \leqslant j,k,d \leqslant D_{\mathcal{M}}} \bar{g}^{j,k} u_d g^{\mathcal{M}}\Big(\nabla^{\mathcal{M}}_{\mathbf{X}_j} \mathbf{X}_d, \mathbf{X}_k\Big) \\
&= \sum_{1 \leqslant j,d \leqslant D_{\mathcal{M}}} \partial^{\mathcal{M}}_{\mathbf{X}_j} u_d \cdot \delta_{j,d} + \sum_{1 \leqslant j,k,d \leqslant D_{\mathcal{M}}} \bar{g}^{j,k} u_d g^{\mathcal{M}}\Big(\nabla^{\mathcal{M}}_{\mathbf{X}_j} \mathbf{X}_d, \mathbf{X}_k\Big).
\end{aligned} \tag{89}$$

where the third equality is derived based on the Leibniz rule and $\delta_{j,d}$ is the Kronecker delta function. The further analysis of term $\nabla^{\mathcal{M}}_{\mathbf{X}_j} \mathbf{X}_d$ relies on the Christoffel symbols $\Gamma^{\mathcal{M}}_{j,k,d}, 1 \leqslant j, k, d \leqslant D_{\mathcal{M}}$ for the connection $\nabla^{\mathcal{M}}$, which specifies

$$\nabla^{\mathcal{M}}_{\mathbf{X}_j} \mathbf{X}_k = \sum_{1 \leqslant d \leqslant D_{\mathcal{M}}} \Gamma^{\mathcal{M}}_{j,k,d} \mathbf{X}_d. \tag{90}$$

Combining the above two equations, we have

$$\mathrm{div}^{\mathcal{M}} \mathbf{U} = \sum_{1 \leqslant j \leqslant D_{\mathcal{M}}} \partial^{\mathcal{M}}_{\mathbf{X}_j} u_j + \sum_{1 \leqslant j,k,d \leqslant D_{\mathcal{M}}} \bar{g}^{j,k} u_d g^{\mathcal{M}}\Big(\sum_{1 \leqslant d' \leqslant D_{\mathcal{M}}} \Gamma^{\mathcal{M}}_{j,d,d'} \mathbf{X}_{d'}, \mathbf{X}_k\Big). \tag{91}$$

Similar to the first term in the right hand side, we can simplify the second term as

$$\begin{aligned}
&\sum_{1 \leqslant j,k,d,d' \leqslant D_{\mathcal{M}}} \bar{g}^{j,k} u_d \Gamma^{\mathcal{M}}_{j,d,d'} g^{\mathcal{M}}(\mathbf{X}_{d'}, \mathbf{X}_k) = \sum_{1 \leqslant j,k,d,d' \leqslant D_{\mathcal{M}}} \bar{g}^{j,k} u_d \Gamma^{\mathcal{M}}_{j,d,d'} \bar{g}_{d',k} \\
&= \sum_{1 \leqslant j,d,d' \leqslant D_{\mathcal{M}}} u_d \Gamma^{\mathcal{M}}_{j,d,d'} [\bar{\mathbf{G}}^{-1} \bar{\mathbf{G}}]_{j,d'} = \sum_{1 \leqslant j,d,d' \leqslant D_{\mathcal{M}}} u_d \Gamma^{\mathcal{M}}_{j,d,d'} \delta_{j,d'} = \sum_{1 \leqslant j,d \leqslant D_{\mathcal{M}}} u_d \Gamma^{\mathcal{M}}_{j,d,j}.
\end{aligned} \tag{92}$$

Therefore, we get the concrete form of divergence operator $\mathrm{div}^{\mathcal{M}}$ as

$$\mathrm{div}^{\mathcal{M}} \mathbf{U} = \sum_{1 \leqslant d \leqslant D_{\mathcal{M}}} \partial^{\mathcal{M}}_{\mathbf{X}_d} u_d + \sum_{1 \leqslant j,d \leqslant D_{\mathcal{M}}} \Gamma^{\mathcal{M}}_{j,d,j} u_d. \tag{93}$$

By incorporating this formula, we can compute the Laplacian of a function as

$$\begin{aligned}
\Delta^{\mathcal{M}}[f] &= \mathrm{div}^{\mathcal{M}}\Big(\sum_{1 \leqslant d \leqslant D_{\mathcal{M}}} b_d \mathbf{X}_d\Big) = \sum_{1 \leqslant d \leqslant D_{\mathcal{M}}} \partial^{\mathcal{M}}_{\mathbf{X}_d} b_d + \sum_{1 \leqslant j,d \leqslant D_{\mathcal{M}}} \Gamma^{\mathcal{M}}_{j,d,j} b_d \\
&= \sum_{1 \leqslant d \leqslant D_{\mathcal{M}}} \partial^{\mathcal{M}}_{\mathbf{X}_d}\Big(\sum_{1 \leqslant d' \leqslant D_{\mathcal{M}}} \partial^{\mathcal{M}}_{\mathbf{X}_{d'}} f \cdot \bar{g}^{d',d}\Big) + \sum_{1 \leqslant j,d \leqslant D_{\mathcal{M}}} \Gamma^{\mathcal{M}}_{j,d,j}\Big(\sum_{1 \leqslant d' \leqslant D_{\mathcal{M}}} \partial^{\mathcal{M}}_{\mathbf{X}_{d'}} f \cdot \bar{g}^{d',d}\Big) \\
&= \sum_{1 \leqslant d,d' \leqslant D_{\mathcal{M}}} \bar{g}^{d',d} \partial^{\mathcal{M}}_{\mathbf{X}_d} \partial^{\mathcal{M}}_{\mathbf{X}_{d'}} f + \sum_{1 \leqslant d,d' \leqslant D_{\mathcal{M}}} \Big(\partial^{\mathcal{M}}_{\mathbf{X}_d} \bar{g}^{d',d} + \sum_{1 \leqslant j \leqslant D_{\mathcal{M}}} \Gamma^{\mathcal{M}}_{j,d,j} \bar{g}^{d',d}\Big) \partial^{\mathcal{M}}_{\mathbf{X}_{d'}} f.
\end{aligned} \tag{94}$$

Like in the Euclidean space, we are interested in the following expression:

$$\Delta^{\mathcal{M}}[fh] - f\Delta^{\mathcal{M}}[h] - h\Delta^{\mathcal{M}}[f]. \tag{95}$$

We can anticipate that the second sum of every Laplacian term crosses out, since the first-order differential operator $\partial^{\mathcal{M}}_{\mathbf{X}_{d'}}$ follows the Leibniz rule:

$$\partial^{\mathcal{M}}_{\mathbf{X}_{d'}}(fh) = f\partial^{\mathcal{M}}_{\mathbf{X}_{d'}}h + h\partial^{\mathcal{M}}_{\mathbf{X}_{d'}}f. \tag{96}$$

For the first sum of each Laplacian term, we repeatedly apply the Leibniz rule to it as

$$\sum_{1\leqslant d,d'\leqslant D_{\mathcal{M}}} \bar{g}^{d',d}\Big(\partial^{\mathcal{M}}_{\mathbf{X}_d}\partial^{\mathcal{M}}_{\mathbf{X}_{d'}}(fh) - f\partial^{\mathcal{M}}_{\mathbf{X}_d}\partial^{\mathcal{M}}_{\mathbf{X}_{d'}}h - h\partial^{\mathcal{M}}_{\mathbf{X}_d}\partial^{\mathcal{M}}_{\mathbf{X}_{d'}}f\Big)$$

$$= \sum_{1\leqslant d,d'\leqslant D_{\mathcal{M}}} \bar{g}^{d',d}\Big(\partial^{\mathcal{M}}_{\mathbf{X}_d}\big(f\partial^{\mathcal{M}}_{\mathbf{X}_{d'}}h + h\partial^{\mathcal{M}}_{\mathbf{X}_{d'}}f\big) - f\partial^{\mathcal{M}}_{\mathbf{X}_d}\partial^{\mathcal{M}}_{\mathbf{X}_{d'}}h - h\partial^{\mathcal{M}}_{\mathbf{X}_d}\partial^{\mathcal{M}}_{\mathbf{X}_{d'}}f\Big)$$

$$= \sum_{1\leqslant d,d'\leqslant D_{\mathcal{M}}} \bar{g}^{d',d}\Big(\partial^{\mathcal{M}}_{\mathbf{X}_d}f \cdot \partial^{\mathcal{M}}_{\mathbf{X}_{d'}}h + \partial^{\mathcal{M}}_{\mathbf{X}_d}h \cdot \partial^{\mathcal{M}}_{\mathbf{X}_{d'}}f\Big) \tag{97}$$

$$= 2\sum_{1\leqslant d,d'\leqslant D_{\mathcal{M}}} \bar{g}^{d',d}(\partial^{\mathcal{M}}_{\mathbf{X}_d}f \cdot \partial^{\mathcal{M}}_{\mathbf{X}_{d'}}h),$$

where the second last equality is derived based on the symmetry of inverse Riemannian metric $\bar{g}^{d',d}$. Combining the above 4 equations, we have

$$\Delta^{\mathcal{M}}[fh] - f\Delta^{\mathcal{M}}[h] - h\Delta^{\mathcal{M}}[f] = 2\sum_{1\leqslant d,d'\leqslant D_{\mathcal{M}}} \bar{g}^{d',d}(\partial^{\mathcal{M}}_{\mathbf{X}_d}f \cdot \partial^{\mathcal{M}}_{\mathbf{X}_{d'}}h)$$

$$= 2\sum_{1\leqslant d,d'\leqslant D_{\mathcal{M}}} \bar{g}^{d',d}(g^{\mathcal{M}}(\text{grad}^{\mathcal{M}}f, \mathbf{X}_d) \cdot \partial^{\mathcal{M}}_{\mathbf{X}_{d'}}h) \tag{98}$$

$$= 2g^{\mathcal{M}}(\text{grad}^{\mathcal{M}}f, \sum_{1\leqslant d,d'\leqslant D_{\mathcal{M}}} \bar{g}^{d',d}(\partial^{\mathcal{M}}_{\mathbf{X}_{d'}}h)\mathbf{X}_d) = 2g^{\mathcal{M}}(\text{grad}^{\mathcal{M}}f, \text{grad}^{\mathcal{M}}g),$$

where the last equality is derived based on Eq. (88). By rearranging the terms, we get

$$g^{\mathcal{M}}(\text{grad}^{\mathcal{M}}f, \text{grad}^{\mathcal{M}}g) = \frac{1}{2}\Big(\Delta^{\mathcal{M}}[fh] - f\Delta^{\mathcal{M}}[h] - h\Delta^{\mathcal{M}}[f]\Big). \tag{99}$$

Note that this equality is the same as Eq. (81), despite the superscript difference. Hence, the conclusion (i.e., Eq. (82)) in the Euclidean space $\mathcal{E}$ can be generalized to the curved space $\mathcal{M}$. The above equation is field-based, and we can simplify it to a point-based expression as

$$g^{\mathcal{M}}_{\mathbf{q}}(\mathbf{v}^i_{d_1}, \mathbf{v}^i_{d_2}) = \Big(\Delta^{\mathcal{M}}[\phi_{d_1}\phi_{d_2}](\mathbf{q}) - \phi_{d_1}(\mathbf{q})\Delta^{\mathcal{M}}[\phi_{d_2}](\mathbf{q}) - \phi_{d_2}(\mathbf{q})\Delta^{\mathcal{M}}[\phi_{d_1}](\mathbf{q})\Big), \tag{100}$$

where $\phi_{d_1}, \phi_{d_2}$ correspond to $f, g$ and $\mathbf{v}^i_{d_1}, \mathbf{v}^i_{d_2}$ are their gradients at point $\mathbf{q}$. For notational convenience, we can further simplify the conclusion with the Leibniz symbol $\text{Lbnz}_{\square}$ as

$$g^{\mathcal{M}}_{\mathbf{q}}(\mathbf{v}^i_{d_1}, \mathbf{v}^i_{d_2}) = \frac{1}{2}\text{Lbnz}_{\Delta^{\mathcal{M}}}[\phi_{d_1}, \phi_{d_2}](\mathbf{q}), \tag{101}$$

where $d_1, d_2$ are any two numbers from $\{1, 2, \cdots, D_{\mathcal{E}}\}$.

### E.6 PROOF: MARKOVIAN CHARACTERIZATION OF THE LINEAR OPERATOR

**Markov-chain generator.** A key conclusion in the theory of Markov chain (Bertini & Cancrini, 1995; Anderson, 2012) is that any homogeneous continuous-time Markov process $\{Z_t\}_{0\leqslant t\leqslant <\infty}$ forms a semi-group in terms of its impact on a smooth function $f : \mathcal{S} \to \mathbb{R}$, where $\mathcal{S}$ denotes the state space. Formally speaking, if we define a class of operators $\{\mathcal{E}_t\}_{t\geqslant 0}$ on the function $f$ as

$$\mathcal{E}_t[f](z) = \mathbb{E}[f(Z_t) \mid Z_0 = z], \tag{102}$$

then those operators form a semi-group:

$$\mathcal{E}_0 = \text{Id}, \quad \mathcal{E}_{s+t} = \mathcal{E}_s \circ \mathcal{E}_t, \tag{103}$$

where $s, t$ are arbitrary non-negative numbers and symbol $\circ$ denotes the operator composition. In this situation, it is proved that such a class of operators have an analytical function:

$$\mathcal{E}_t = \exp(t\mathscr{L}) = \sum_{k\geqslant 0} \frac{t^k}{k!}\Big(\underbrace{\mathscr{L}\circ\mathscr{L}\circ\cdots\circ\mathscr{L}}_{k\text{-time composition}}\Big), \tag{104}$$

where $\mathscr{L}$ is a linear operator named as the infinitesimal generator. The evaluation of this linear operator $\mathscr{L}$ on function $f$ can also be computed as

$$\mathscr{L}[f](z) = \lim_{t \to 0} \frac{1}{t}\Big(\mathcal{E}_t[f](z) - f(z)\Big) = \lim_{t \to 0} \frac{1}{t}\Big(\mathbb{E}[f(Z_t) \mid Z_0 = z] - f(z)\Big). \tag{105}$$

Let us consider the expectation term on the right hand side: it can be expanded as

$$\mathbb{E}[\cdot] = \int f(z')P(Z_t = z' \mid Z_0 = z)\mu(\mathrm{d}z'), \tag{106}$$

where the inside conditional probability term $P(\cdot)$ is called the Markov kernel, which is typically re-symbolized as $k_t(z, z'), z, z' \in \mathcal{S}$, and $\mu$ is some volume measure defined on the abstract space $\mathcal{S}$ (e.g., smooth manifold $\mathcal{M}$). Obviously, the kernel is always non-negative and satisfies the normalization constraint: $\int k_t(z, z')\mathrm{d}\mu(z') = 1$.

**Monte Carlo approximation.** In light of the former background review of continuous-time Markov chain, we can see that the evaluation of linear operator $\mathscr{L}$ with any test function $f \in \mathcal{C}^\infty(\mathcal{M})$ can be approximated in an unbiased manner as

$$\mathscr{L}[f](z) = \lim_{t \to 0} \frac{1}{t}\Big(\int (f(z') - f(z))k_t(z, z')\mu(\mathrm{d}z')\Big), \tag{107}$$

when $t$ is close to 0. Finally, suppose that the state space $\mathcal{S}$ has a bounded measure $\mu(S) = \int_{z \in \mathcal{S}} \mu(\mathrm{d}z) < \infty$, and the observed data $\{z_i'\}_{i \in [1, N]}$ are uniformly sampled from the state space $\mathcal{S}$, then we can arrive at the below conclusion:

$$\begin{aligned}
\mathscr{L}[f](z) &= \lim_{t \to 0} \frac{1}{t}\Big(\int (f(z') - f(z))k_t(z, z')\mu(S)\frac{1}{\mu(S)}\mu(\mathrm{d}z')\Big) \\
&= \mu(S)\lim_{t \to 0} \frac{1}{t}\Big(\int (f(z') - f(z))k_t(z, z')\mathcal{U}_\mathcal{S}(z')\mu(\mathrm{d}z')\Big) \\
&= \mu(S)\lim_{t \to 0} \frac{1}{t}\mathbb{E}_{z' \sim \mathcal{U}_\mathcal{S}(z')}\Big[(f(z') - f(z))k_t(z, z')\Big] \\
&\approx \frac{\mu(S)}{tN}\Big(\sum_{1 \leqslant i \leqslant N}(f(z_i') - f(z))k_t(z, z_i')\Big),
\end{aligned} \tag{108}$$

where number $t$ in the final part is close to 0, and term $\mathcal{U}_\mathcal{S}$ denotes the density of a uniform distribution over space $\mathcal{S}$. The uniform assumption also makes sense since this Markov chain $\{Z_t\}_{t \geqslant 0}$ is just manually designed (i.e., not a stochastic process realized in the real world).

### E.7 PROOF: ESTIMATIONS OF THE DIMENSION AND NORMAL VECTORS

For an arbitrary point $\mathbf{q} = \boldsymbol{\mu}_{\mathrm{obj}}^i$ residing in the embedded Riemannian manifold $\mathcal{M} \hookrightarrow \mathcal{E}$, we have proved two key conclusions in Appendix E.4.3:

*Conclusion-1: eigenvectors as normal vectors.* Any normal vector $\mathbf{n} \in \mathcal{N}_\mathbf{q}\mathcal{M}$ is an eigenvector of the kernel matrix $\mathcal{K}^i$, with a zero eigenvalue $\lambda_d = 0$;

*Conclusion-2: eigenvectors as tangent vectors.* Any tangent vector $\mathbf{u} \in \mathcal{T}_\mathbf{q}\mathcal{M}$ is an eigenvector of the same matrix, with a unit eigenvalue $\lambda_d = 1$.

We can infer that the eigenvectors $\{\mathbf{u}_d^i \mid (\mathbf{u}_d^i, \lambda_d), \lambda_d = 1\}_d$ in the second case form an orthonormal basis of the tangent space $\mathcal{T}_\mathbf{q}\mathcal{M}$. Therefore, the number of positive eigenvalues is equal to the manifold dimension $D_\mathcal{M}$. In a similar sense, the eigenvectors $\{\mathbf{u}_d^i \mid (\mathbf{u}_d^i, \lambda_d), \lambda_d = 0\}_d$ in the first scenario constitute a basis for the normal space $\mathcal{N}_\mathbf{q}\mathcal{M}$.

### E.8 PROOF: CURVED-BASED FORMULATION OF THE SHAPE OPERATOR

The shape operator $s^i$ is a geometric quantity defined at every point $\mathbf{q} = \boldsymbol{\mu}_{\mathrm{obj}}^i$ on an embedded manifold $\mathcal{M} \hookrightarrow \mathcal{E}$. The operator $s^i$ has many forms, and we adopt its definition (Lee, 2018) as

$$s^i(\mathbf{n}, \mathbf{u}_1, \mathbf{u}_2) = -g_\mathbf{q}^\mathcal{E}(\nabla_{\mathbf{u}_1}^\mathcal{E}\mathbf{N}\mid_\mathbf{q}, \mathbf{u}_2), \tag{109}$$

where $\mathbf{N}$ is a normal section (which extends the normal vector $\mathbf{n}$ at point $\mathbf{q}$) and $(\cdot)\mid_{\mathbf{q}}$ means to take the value of expression $(\cdot)$ at point $\mathbf{q}$. In extrinsic differential geometry, the shape operator $s^i$ is closely related to the concept of height function:

$$\eta(\mathbf{q}') = g_{\mathbf{q}}^{\mathcal{E}}(\mathbf{n}, \mathbf{q}' - \mathbf{q}) = \mathbf{n}^{\top}(\mathbf{q}' - \mathbf{q}). \tag{110}$$

Now, Let us verify this point. Based on Eq. (59), the derivative of this point is as

$$\operatorname{grad}_{\mathbf{q}'}^{\mathcal{M}} \eta = (\operatorname{grad}_{\mathbf{q}'}^{\mathcal{E}} \eta)^{\pi} = \mathbf{n}^{\pi_{\mathbf{q}'}}, \tag{111}$$

where $\pi_{\mathbf{q}'}$ means the perpendicular projection to the tangent plane $\mathcal{T}_{\mathbf{q}'}\mathcal{M}$. Given that $\mathbf{n} \perp \mathcal{T}_{\mathbf{q}}\mathcal{M}$, it is obvious that $\operatorname{grad}_{\mathbf{q}}^{\mathcal{M}} \eta = \mathbf{0}$. Therefore, the point $\mathbf{q}$ is both a zero point ($\eta(\mathbf{q}) = \mathbf{n}^{\top}\mathbf{0} = 0$) and a critical point to the height function $\eta$.

Before proceeding further, we need to first derive the Hessian tensor $\operatorname{Hess}^{\mathcal{M}}$ of an arbitrary function $f \in \mathcal{C}^{\infty}(\mathcal{M})$. The formal definition of this tensor is as

$$\operatorname{Hess}^{\mathcal{M}} f(\mathbf{U}_1, \mathbf{U}_2) = g^{\mathcal{M}}(\nabla_{\mathbf{U}_1}^{\mathcal{M}} \operatorname{grad}^{\mathcal{M}} f, \mathbf{U}_2), \forall \mathbf{U}_1, \mathbf{U}_2 \in \mathcal{V}(\mathcal{M}). \tag{112}$$

Given that the affine connection $\nabla^{\mathcal{M}}$ is induced by the Riemannian metric $g^{\mathcal{M}}$, the Hessian tensor field can be expanded as follows:

$$\begin{aligned}
\operatorname{Hess}^{\mathcal{M}} f(\mathbf{U}_1, \mathbf{U}_2) &= \partial_{\mathbf{U}_1}^{\mathcal{M}} g^{\mathcal{M}}(\operatorname{grad}^{\mathcal{M}} f, \mathbf{U}_2) - g^{\mathcal{M}}(\operatorname{grad}^{\mathcal{M}} f, \nabla_{\mathbf{U}_1}^{\mathcal{M}} \mathbf{U}_2) \\
&= \nabla_{\mathbf{U}_1}^{\mathcal{M}} \nabla_{\mathbf{U}_2}^{\mathcal{M}} f - g^{\mathcal{M}}(\operatorname{grad}^{\mathcal{M}} f, \nabla_{\mathbf{U}_1}^{\mathcal{M}} \mathbf{U}_2) = \partial_{\mathbf{U}_1}^{\mathcal{M}} \partial_{\mathbf{U}_2}^{\mathcal{M}} f - \partial_{\nabla_{\mathbf{U}_1}^{\mathcal{M}} \mathbf{U}_2}^{\mathcal{M}} f.
\end{aligned} \tag{113}$$

Based on previous conclusions (i.e., Eq. (52) and Eq. (59)) on the Embedded manifold $\mathcal{M}$, the Riemannian Hessian $\operatorname{Hess}^{\mathcal{M}}$ can be related to the Euclidean Hessian $\operatorname{Hess}^{\mathcal{E}}$ as

$$\begin{aligned}
\operatorname{Hess}^{\mathcal{M}} f(\mathbf{U}_1, \mathbf{U}_2) &= \partial_{\mathbf{U}_1}^{\mathcal{M}} \partial_{\mathbf{U}_2}^{\mathcal{M}} f - g^{\mathcal{E}}(\operatorname{grad}^{\mathcal{M}} f, \nabla_{\mathbf{U}_1}^{\mathcal{M}} \mathbf{U}_2) \\
&= \partial_{\mathbf{U}_1}^{\mathcal{E}} \partial_{\mathbf{U}_2}^{\mathcal{E}} f - \langle \operatorname{grad}^{\mathcal{E}} f - (\operatorname{grad}^{\mathcal{E}} f)^{\perp}, \nabla_{\mathbf{U}_1}^{\mathcal{M}} \mathbf{U}_2 \rangle_{\mathcal{E}} \\
&= \partial_{\mathbf{U}_1}^{\mathcal{E}} \partial_{\mathbf{U}_2}^{\mathcal{E}} f - \langle \operatorname{grad}^{\mathcal{E}} f, \nabla_{\mathbf{U}_1}^{\mathcal{E}} \mathbf{U}_2 - (\nabla_{\mathbf{U}_1}^{\mathcal{E}} \mathbf{U}_2)^{\perp} \rangle_{\mathcal{E}} \\
&= \partial_{\mathbf{U}_1}^{\mathcal{E}} \partial_{\mathbf{U}_2}^{\mathcal{E}} f - \partial_{\nabla_{\mathbf{U}_1}^{\mathcal{E}} \mathbf{U}_2}^{\mathcal{E}} f + \partial_{(\nabla_{\mathbf{U}_1}^{\mathcal{E}} \mathbf{U}_2)^{\perp}}^{\mathcal{E}} f = \operatorname{Hess}^{\mathcal{E}} f(\mathbf{U}_1, \mathbf{U}_2) + \partial_{(\nabla_{\mathbf{U}_1}^{\mathcal{E}} \mathbf{U}_2)^{\perp}}^{\mathcal{E}} f,
\end{aligned} \tag{114}$$

where $\perp$ indicates the orthogonal projection to the normal bundle $\mathcal{N}\mathcal{M} = \sqcup_{\mathbf{q} \in \mathcal{M}} \mathcal{N}_{\mathbf{q}}\mathcal{M}$. For the metric-compatible connection $\nabla^{\mathcal{M}}$, we can also show that the Riemannian Hessian $\operatorname{Hess}^{\mathcal{M}}$ is symmetric, based on the torsion-free condition:

$$\operatorname{Hess}^{\mathcal{M}} f(\mathbf{U}_1, \mathbf{U}_2) = \partial_{\mathbf{U}_1}^{\mathcal{M}} \partial_{\mathbf{U}_2}^{\mathcal{M}} f - \partial_{\nabla_{\mathbf{U}_1}^{\mathcal{M}} \mathbf{U}_2}^{\mathcal{M}} f = \partial_{\mathbf{U}_2}^{\mathcal{M}} \partial_{\mathbf{U}_1}^{\mathcal{M}} f - \partial_{\nabla_{\mathbf{U}_2}^{\mathcal{M}} \mathbf{U}_1}^{\mathcal{M}} f = \operatorname{Hess}^{\mathcal{M}} f(\mathbf{U}_2, \mathbf{U}_1). \tag{115}$$

The central equality holds as we can apply the torsion-free condition to function $f$.

Importantly, we show that the Hessian tensor $\operatorname{Hess}^{\mathcal{M}} f$ is associated with the shape operator $s^i$. Based on Eq. (52) and Eq. (59), we have

$$\begin{aligned}
\operatorname{Hess}^{\mathcal{M}} f(\mathbf{U}_1, \mathbf{U}_2) &= g^{\mathcal{E}}((\nabla_{\mathbf{U}_1}^{\mathcal{E}} \operatorname{grad}^{\mathcal{M}} f)^{\pi}, \mathbf{U}_2) \\
&= g^{\mathcal{E}}(\nabla_{\mathbf{U}_1}^{\mathcal{E}} \operatorname{grad}^{\mathcal{M}} f, \mathbf{U}_2) - g^{\mathcal{E}}((\nabla_{\mathbf{U}_1}^{\mathcal{E}} \operatorname{grad}^{\mathcal{M}} f)^{\perp}, \mathbf{U}_2) = g^{\mathcal{E}}(\nabla_{\mathbf{U}_1}^{\mathcal{E}} \operatorname{grad}^{\mathcal{M}} f, \mathbf{U}_2) \\
&= g^{\mathcal{E}}(\nabla_{\mathbf{U}_1}^{\mathcal{E}} \mathbf{n}^{\Pi}, \mathbf{U}_2) = g^{\mathcal{E}}(\nabla_{\mathbf{U}_1}^{\mathcal{E}} (\mathbf{N}' - g^{\mathcal{E}}(\mathbf{N}', \mathbf{N})\mathbf{N}), \mathbf{U}_2),
\end{aligned} \tag{116}$$

where $\Pi$ represents the element-wise projection to the tangent bundle $\mathcal{T}\mathcal{M}$, and $\mathbf{N}'$ denotes a constant vector field valued as $\mathbf{n}$ at every point. By simplifying the last term, we get

$$\begin{aligned}
\operatorname{Hess}^{\mathcal{M}} f(\mathbf{U}_1, \mathbf{U}_2) &= g^{\mathcal{E}}(\nabla_{\mathbf{U}_1}^{\mathcal{E}} \mathbf{N}', \mathbf{U}_2) - g^{\mathcal{E}}(\nabla_{\mathbf{U}_1}^{\mathcal{E}} (g^{\mathcal{E}}(\mathbf{N}', \mathbf{N})\mathbf{N}), \mathbf{U}_2) \\
&= 0 - g^{\mathcal{E}}(\nabla_{\mathbf{U}_1}^{\mathcal{E}} (g^{\mathcal{E}}(\mathbf{N}', \mathbf{N}))\mathbf{N} + g^{\mathcal{E}}(\mathbf{N}', \mathbf{N})\nabla_{\mathbf{U}_1}^{\mathcal{E}} \mathbf{N}, \mathbf{U}_2) = -g^{\mathcal{E}}(\mathbf{N}', \mathbf{N})g^{\mathcal{E}}(\nabla_{\mathbf{U}_1}^{\mathcal{E}} \mathbf{N}, \mathbf{U}_2).
\end{aligned} \tag{117}$$

The above equality is tensorized, and its form at point $\mathbf{q} \in \mathcal{M}$ is as

$$\operatorname{Hess}_{\mathbf{q}}^{\mathcal{M}} f(\mathbf{u}_1, \mathbf{u}_2) = -g_{\mathbf{q}}^{\mathcal{E}}(\mathbf{n}', \mathbf{n})g_{\mathbf{q}}^{\mathcal{E}}(\nabla_{\mathbf{u}_1}^{\mathcal{E}} \mathbf{N}, \mathbf{u}_2) = -g_{\mathbf{q}}^{\mathcal{E}}(\nabla_{\mathbf{u}_1}^{\mathcal{E}} \mathbf{N} \mid_{\mathbf{q}}, \mathbf{u}_2), \tag{118}$$

which exactly matches the previous definition of the shape operator $s^i$.

As the last stage, we aim to simplify the below expression:

$$s^i(\mathbf{n}, \mathbf{v}_{d_1}^i, \mathbf{v}_{d_2}^i) = \text{Hess}_{\mathbf{q}}^{\mathcal{M}} \eta(\mathbf{v}_{d_1}^i, \mathbf{v}_{d_2}^i) = g^{\mathcal{M}}(\nabla_{\text{grad}^{\mathcal{M}} \phi_{d_1}}^{\mathcal{M}} \text{grad}^{\mathcal{M}} \eta, \text{grad}^{\mathcal{M}} \phi_{d_2}) \mid_{\mathbf{q}} . \quad (119)$$

Based on the explicit expression (i.e., Eq. (23)) of affine connection $\nabla^{\mathcal{M}}$, we can get

$$\begin{aligned}
s^i(\mathbf{n}, \mathbf{v}_{d_1}^i, \mathbf{v}_{d_2}^i) = \frac{1}{2} \Big( & \partial_{\text{grad}^{\mathcal{M}} \phi_{d_1}}^{\mathcal{M}} g^{\mathcal{M}}(\text{grad}^{\mathcal{M}} \eta, \text{grad}^{\mathcal{M}} \phi_{d_2}) \\
& + \partial_{\text{grad}^{\mathcal{M}} \eta}^{\mathcal{M}} g^{\mathcal{M}}(\text{grad}^{\mathcal{M}} \phi_{d_2}, \text{grad}^{\mathcal{M}} \phi_{d_1}) - \partial_{\text{grad}^{\mathcal{M}} \phi_{d_2}}^{\mathcal{M}} g^{\mathcal{M}}(\text{grad}^{\mathcal{M}} \phi_{d_1}, \text{grad}^{\mathcal{M}} \eta) \\
& + g^{\mathcal{M}}([\text{grad}^{\mathcal{M}} \phi_{d_1}, \text{grad}^{\mathcal{M}} \eta], \text{grad}^{\mathcal{M}} \phi_{d_2}) - g^{\mathcal{M}}([\text{grad}^{\mathcal{M}} \eta, \text{grad}^{\mathcal{M}} \phi_{d_2}], \text{grad}^{\mathcal{M}} \phi_{d_1}) \\
& + g^{\mathcal{M}}([\text{grad}^{\mathcal{M}} \phi_{d_2}, \text{grad}^{\mathcal{M}} \phi_{d_1}], \text{grad}^{\mathcal{M}} \eta) \Big) \mid_{\mathbf{q}} .
\end{aligned} \quad (120)$$

Inside the brackets of the right hand side, the last three terms are similar in form. For the forth term, we can expand it as follows:

$$\begin{aligned}
g^{\mathcal{M}}([\text{grad}^{\mathcal{M}} \phi_{d_1}, \text{grad}^{\mathcal{M}} \eta], \text{grad}^{\mathcal{M}} \phi_{d_2}) &= \partial_{[\text{grad}^{\mathcal{M}} \phi_{d_1}, \text{grad}^{\mathcal{M}} \eta]}^{\mathcal{M}} \phi_{d_2} = \\
&= \partial_{\text{grad}^{\mathcal{M}} \phi_{d_1}}^{\mathcal{M}} \partial_{\text{grad}^{\mathcal{M}} \eta}^{\mathcal{M}} \phi_{d_2} - \partial_{\text{grad}^{\mathcal{M}} \eta}^{\mathcal{M}} \partial_{\text{grad}^{\mathcal{M}} \phi_{d_1}}^{\mathcal{M}} \phi_{d_2} \\
&= \partial_{\text{grad}^{\mathcal{M}} \phi_{d_1}}^{\mathcal{M}} g^{\mathcal{M}}(\text{grad}^{\mathcal{M}} \eta, \text{grad}^{\mathcal{M}} \phi_{d_2}) - \partial_{\text{grad}^{\mathcal{M}} \eta}^{\mathcal{M}} g^{\mathcal{M}}(\text{grad}^{\mathcal{M}} \phi_{d_1}, \text{grad}^{\mathcal{M}} \phi_{d_2}).
\end{aligned} \quad (121)$$

Applying the same derivation to the last two terms, we have

$$\begin{aligned}
s^i(\mathbf{n}, \mathbf{v}_{d_1}^i, \mathbf{v}_{d_2}^i) = \frac{1}{2} \Big( & \partial_{\text{grad}^{\mathcal{M}} \phi_{d_1}}^{\mathcal{M}} g^{\mathcal{M}}(\cdot) + \partial_{\text{grad}^{\mathcal{M}} \eta}^{\mathcal{M}} g^{\mathcal{M}}(\cdot) - \partial_{\text{grad}^{\mathcal{M}} \phi_{d_2}}^{\mathcal{M}} g^{\mathcal{M}}(\cdot) \\
& + \partial_{\text{grad}^{\mathcal{M}} \phi_{d_1}}^{\mathcal{M}} g^{\mathcal{M}}(\text{grad}^{\mathcal{M}} \eta, \text{grad}^{\mathcal{M}} \phi_{d_2}) - \partial_{\text{grad}^{\mathcal{M}} \eta}^{\mathcal{M}} g^{\mathcal{M}}(\text{grad}^{\mathcal{M}} \phi_{d_1}, \text{grad}^{\mathcal{M}} \phi_{d_2}) \\
& - \partial_{\text{grad}^{\mathcal{M}} \eta}^{\mathcal{M}} g^{\mathcal{M}}(\text{grad}^{\mathcal{M}} \phi_{d_2}, \text{grad}^{\mathcal{M}} \phi_{d_1}) + \partial_{\text{grad}^{\mathcal{M}} \phi_{d_2}}^{\mathcal{M}} g^{\mathcal{M}}(\text{grad}^{\mathcal{M}} \eta, \text{grad}^{\mathcal{M}} \phi_{d_1}) \\
& + \partial_{\text{grad}^{\mathcal{M}} \phi_{d_2}}^{\mathcal{M}} g^{\mathcal{M}}(\text{grad}^{\mathcal{M}} \phi_{d_1}, \text{grad}^{\mathcal{M}} \eta) - \partial_{\text{grad}^{\mathcal{M}} \phi_{d_1}}^{\mathcal{M}} g^{\mathcal{M}}(\text{grad}^{\mathcal{M}} \phi_{d_2}, \text{grad}^{\mathcal{M}} \eta) \Big) \mid_{\mathbf{q}} .
\end{aligned} \quad (122)$$

By deleting the same terms, we can get

$$\begin{aligned}
s^i(\mathbf{n}, \mathbf{v}_{d_1}^i, \mathbf{v}_{d_2}^i) = \frac{1}{2} \Big( & \partial_{\text{grad}^{\mathcal{M}} \phi_{d_1}}^{\mathcal{M}} g^{\mathcal{M}}(\text{grad}^{\mathcal{M}} \eta, \text{grad}^{\mathcal{M}} \phi_{d_2}) \\
& + \partial_{\text{grad}^{\mathcal{M}} \phi_{d_2}}^{\mathcal{M}} g^{\mathcal{M}}(\text{grad}^{\mathcal{M}} \eta, \text{grad}^{\mathcal{M}} \phi_{d_1}) - \partial_{\text{grad}^{\mathcal{M}} \eta}^{\mathcal{M}} g^{\mathcal{M}}(\text{grad}^{\mathcal{M}} \phi_{d_1}, \text{grad}^{\mathcal{M}} \phi_{d_2}) \Big) \mid_{\mathbf{q}} .
\end{aligned} \quad (123)$$

For the first term on the right hand side, we can reshape it as

$$\begin{aligned}
\partial_{\text{grad}^{\mathcal{M}} \phi_{d_1}}^{\mathcal{M}} g^{\mathcal{M}}(\text{grad}^{\mathcal{M}} \eta, \text{grad}^{\mathcal{M}} \phi_{d_2}) &= g^{\mathcal{M}}(\text{grad}^{\mathcal{M}} \phi_{d_1}, \text{grad}^{\mathcal{M}} g^{\mathcal{M}}(\text{grad}^{\mathcal{M}} \eta, \text{grad}^{\mathcal{M}} \phi_{d_2})) \\
&= \frac{1}{2} g^{\mathcal{M}}(\text{grad}^{\mathcal{M}} \phi_{d_1}, \text{grad}^{\mathcal{M}} \text{Lbnz}_{\Delta^{\mathcal{M}}}[\eta, \phi_{d_2}]) \\
&= \frac{1}{4} \text{Lbnz}_{\Delta^{\mathcal{M}}}[\phi_{d_1}, \text{Lbnz}_{\Delta^{\mathcal{M}}}[\eta, \phi_{d_2}]) := \frac{1}{4} \mathcal{A}[\phi_{d_1}, \eta, \phi_{d_2}],
\end{aligned} \quad (124)$$

where the last equality defines a new operator $\mathcal{A}$ for notational convenience. Likewise, we can reshape the last two terms of shape operator $s^i$ and finally express it as

$$s^i(\mathbf{n}, \mathbf{v}_{d_1}^i, \mathbf{v}_{d_2}^i) = \frac{1}{8} \Big( \mathcal{A}[\phi_{d_1}, \eta, \phi_{d_2}](\mathbf{q}) + \mathcal{A}[\phi_{d_2}, \eta, \phi_{d_1}](\mathbf{q}) - \mathcal{A}[\eta, \phi_{d_1}, \phi_{d_2}](\mathbf{q}) \Big), \quad (125)$$

which proves the main claim of this proposition.

### E.9 PROOF: CURVATURE ESTIMATION

The shape operator $s^i$ at an arbitrary point $\mathbf{q} = \boldsymbol{\mu}_{\text{obj}}^i \in \mathcal{M}$ is linear in its tangent vector inputs, given a fixed normal vector $\mathbf{n} \in \mathcal{N}_{\mathbf{q}} \mathcal{M}$. In this regard, the operator is commonly represented as a matrix $\mathbf{M}^i$ in extrinsic differential geometry (Lee, 2018). It is a known fact that the eigenvalues and eigenvectors of matrix $\mathbf{M}^i$ respectively correspond to the principle curvature and directions.

For the matrix $\boldsymbol{\mathcal{S}}^i \in \mathbb{R}^{D_{\mathcal{M}} \times D_{\mathcal{M}}}$, it can be represented as

$$\boldsymbol{\mathcal{S}}^i = (\boldsymbol{\mathcal{U}}^i)^\top \mathbf{M}^i \boldsymbol{\mathcal{U}}^i, \tag{126}$$

Here $\boldsymbol{\mathcal{U}}^i$ is an orthonormal matrix as $\{\mathbf{u}_d^i\}_{1 \leqslant d \leqslant D_{\mathcal{M}}}$ is an orthonormal basis in the tangent space $\mathcal{T}_{\mathbf{q}}\mathcal{M}$. Importantly, this means $(\boldsymbol{\mathcal{U}}^i)^\top \boldsymbol{\mathcal{U}}^i = \boldsymbol{\mathcal{U}}^i(\boldsymbol{\mathcal{U}}^i)^\top = \mathbf{I}$. Suppose that $(\tau, \mathbf{w})$ is a pair of eigenvalue and eigenvector for matrix $\mathbf{M}^i$ as $\mathbf{M}^i \mathbf{w} = \tau \mathbf{w}$, then we can get

$$\boldsymbol{\mathcal{S}}^i((\boldsymbol{\mathcal{U}}^i)^\top \mathbf{w}) = (\boldsymbol{\mathcal{U}}^i)^\top \mathbf{M}^i((\boldsymbol{\mathcal{U}}^i(\boldsymbol{\mathcal{U}}^i)^\top)\mathbf{w} = (\boldsymbol{\mathcal{U}}^i)^\top \mathbf{M}^i \mathbf{w} = \tau (\boldsymbol{\mathcal{U}}^i)^\top \mathbf{w}. \tag{127}$$

Therefore, the pair $(\tau, (\boldsymbol{\mathcal{U}}^i)^\top \mathbf{w})$ is the eigenvalue and eigenvector of matrix $\boldsymbol{\mathcal{S}}^i$. In light of this derivation, we can infer that the eigen-decomposition of matrix $\boldsymbol{\mathcal{S}}^i$ corresponds to the principle curvatures and directions.

# F  EXPERIMENT SETTINGS AND MORE RESULTS

In this section, we provide the details of our experiment settings, and show additional results that confirm the effectiveness of our framework: *GeoSplat*.

## F.1  EXPERIMENT SETTINGS

**Benchmark datasets.**  We compare our method with the baselines on two groups of publicly available datasets: Replica (Straub et al., 2019) and ICL (Handa et al., 2014), which respectively consist of $8$ and $4$ 3D scenes. The Replica datasets are collected from commonly seen living room and offices, and the ICL datasets provide similar environments. A key baseline (Li et al., 2025) also adopted these datasets for model evaluation, so we follow their train and test splits.

**Baselines.**  The vanilla Gaussian splatting (i.e., 3DGS) (Kerbl et al., 2023) is a natural baseline in our setting for verifying the performance gains of our framework. In addition to this, an important baseline is GeoGaussian (Li et al., 2025), which adopted a number of regularization strategies (e.g., co-planar constraints) based on low-order geometric priors (e.g., normal information) and covered most of other works (Wang et al., 2024; Turkulainen et al., 2025) on geometric regularization. It is necessary to show that our framework, which is compatible with their low-order regularization and further exploits higher-order geometric information, can achieve better performance. We also compare with Ververas et al. (2025) to show that the mean absolute curvature (MAC) is indeed a better low-curvature area identifier than the mean curvature.

There are other baselines included for diversity. For example, Gaussian-Splatting SLAM (Matsuki et al., 2024) is an end-to-end method that uses Gaussians as the map for incremental localization, reconstruction, and rendering tasks, while Vox-Fusion (Yang et al., 2022) represents a hybrid method that integrates the implicit neural representation (Sitzmann et al., 2020; Mildenhall et al., 2021) into traditional volumetric rendering. For LightGS (Fan et al., 2024), it further refines the vanilla 3DGS by pruning insignificant Gaussian primitives and fine-tuning further. To make fair comparisons, we copy available results from Li et al. (2025), following the same experiment setup.

**Evaluation metrics.**  For quantitatively measuring the quality of novel view rendering, we follow the standard evaluation metrics used in many previous works (Straub et al., 2019; Kerbl et al., 2023; Li et al., 2025), including Peak Signal-to-Noise Ratio (PSNR), Structural Similarity Index Measure (SSIM), and Learned Perceptual Image Patch Similarity (LPIPS). PSNR evaluates on a color-wise basis, while SSIM measures the similarity between two images in terms of structural information, luminance, and distortion. For LPIPS, it compares two images using their features extracted by a pre-trained neural network, like VGG-Net (Simonyan & Zisserman, 2014).

**Hyper-parameter and method configurations.**  For the tiny and large thresholds $\xi_{\min}, \xi_{\max}$, we respectively specify them as $0.001$ and $\xi_{\mathrm{mean}} + 3\xi_{\mathrm{std}}$, where $\xi_{\mathrm{mean}}$ denotes the average of all estimated curvatures in the data and $\xi_{\mathrm{std}}$ represents their standard deviation. The number of nearest neighbors for primitive upsampling is set as $10$. The position vector $\boldsymbol{\mu}_{\mathrm{obj}}^i$ is warmed up through structure from motion (SfM) (Schonberger & Frahm, 2016). For the tangential projection operation $\top$, we implement it as $\mathbf{v}^\top = \langle \mathbf{u}_1^i, \mathbf{v} \rangle \mathbf{u}_1^i + \langle \mathbf{u}_2^i, \mathbf{v} \rangle \mathbf{u}_2^i$ for any vector $\mathbf{v} \in \mathcal{E}$, which is $\mathbf{v}^\perp = \langle \widetilde{\mathbf{u}}_3^i, \mathbf{v} \rangle \widetilde{\mathbf{u}}_3^i$

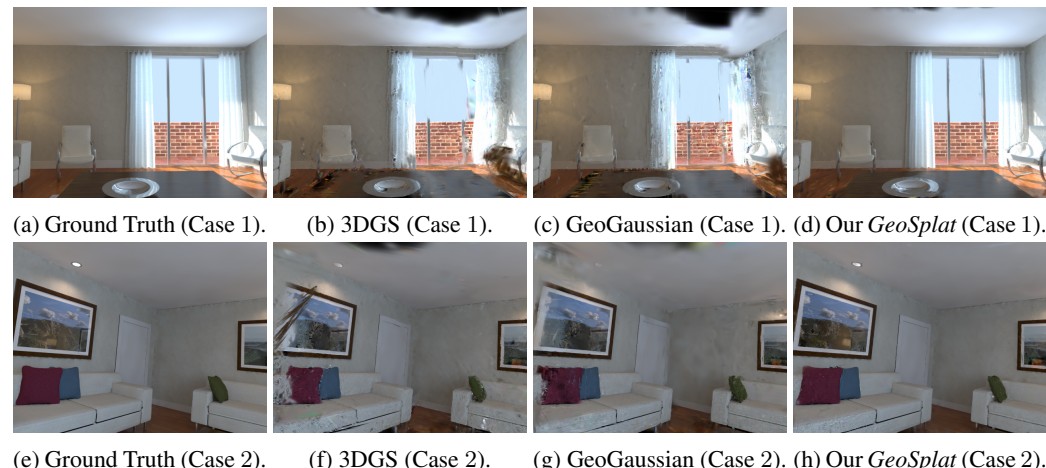

(a) Ground Truth (Case 1).   (b) 3DGS (Case 1).   (c) GeoGaussian (Case 1).   (d) Our *GeoSplat* (Case 1).

(e) Ground Truth (Case 2).   (f) 3DGS (Case 2).   (g) GeoGaussian (Case 2).   (h) Our *GeoSplat* (Case 2).

Figure 4: Ground-truth and rendered images on the low-resource ICL Room-2 dataset. The cases 1, 2 are respectively generated by our varifold-based and manifold-based models.

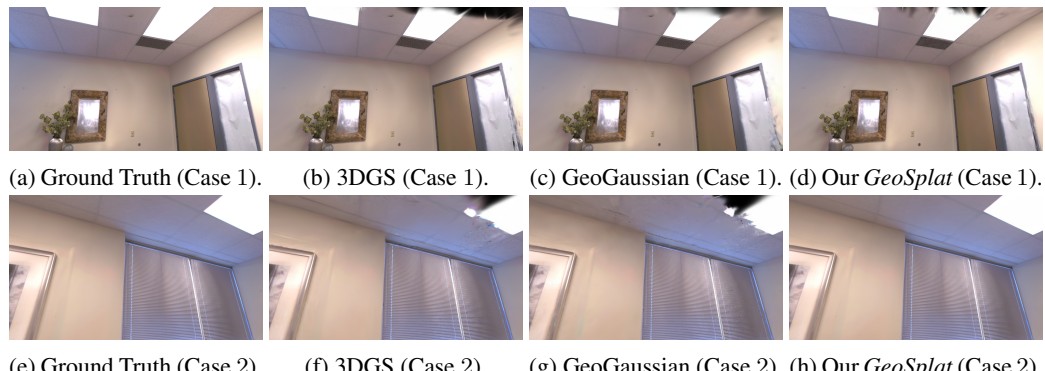

(a) Ground Truth (Case 1).   (b) 3DGS (Case 1).   (c) GeoGaussian (Case 1).   (d) Our *GeoSplat* (Case 1).

(e) Ground Truth (Case 2).   (f) 3DGS (Case 2).   (g) GeoGaussian (Case 2).   (h) Our *GeoSplat* (Case 2).

Figure 5: Ground-truth and rendered images on the low-resource Replica R1 dataset. The cases 1, 2 are respectively generated by our varifold-based and manifold-based models.

for projection $\perp$ to the normal direction. The final loss function for optimization is the original one (e.g., D-SSIM) plus our regularization terms $\mathcal{L}_{\text{scale}}, \mathcal{L}_{\text{rot}}$. We run our models on 3 NVIDIA Tesla V100 GPU devices, with the performance improvements over the baselines are statistically significant with $p < 0.05$ under t-test.

### F.2   MORE CASE STUDIES

We present additional case studies that show our models can render images that contain much fewer artifacts than baselines. The results are provided in Fig. 4, Fig 5, and Fig. 6.

### F.3   ENRICHED GAUSSIAN PRIMITIVES

To show how our curvature-guided primitive upsampling strategy works, we run it on Replica OFF2 and show the enriched primitive cloud in Fig. 7.

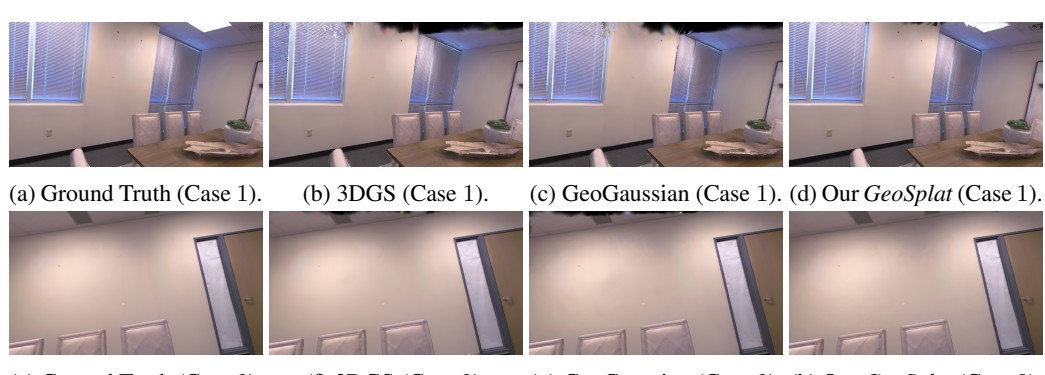

(a) Ground Truth (Case 1). (b) 3DGS (Case 1). (c) GeoGaussian (Case 1). (d) Our *GeoSplat* (Case 1).

(e) Ground Truth (Case 2). (f) 3DGS (Case 2). (g) GeoGaussian (Case 2). (h) Our *GeoSplat* (Case 2).

Figure 6: Ground-truth and rendered images on the low-resource Replica R2 dataset. The cases 1, 2 are respectively generated by our varifold-based and manifold-based models.

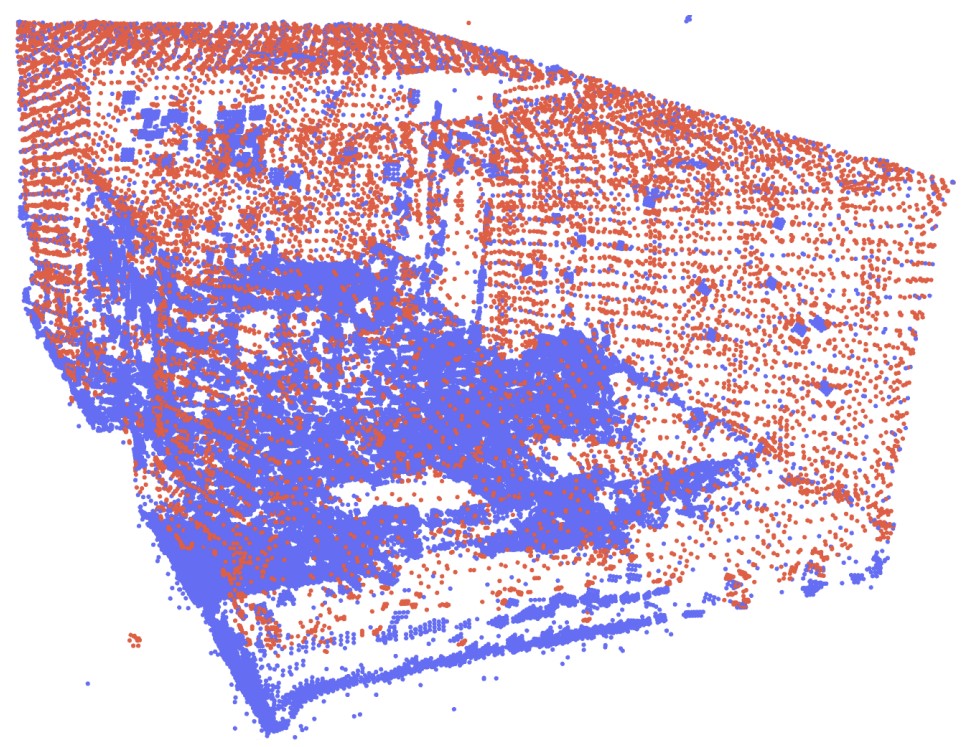

Figure 7: The initial Gaussian primitives of Replica OFF2 are enriched through our curvature-guided primitive upsampling strategy. The points in red are newly added.

