# OpenReview forum: "GeoSplat: A Deep Dive into Geometry-Constrained Gaussian Splatting"
_ICLR.cc/2026/Conference — Submitted to ICLR 2026_

### Official Review · Reviewer_kXse · 2025-10-28

**Soundness:** 2
**Presentation:** 3
**Contribution:** 3
**Rating:** 4
**Confidence:** 3

**Summary:**

This paper extends the 3D Gaussian splatting to constraints Gaussian primitives to reside on smooth manifolds (or varifold), leveraging tangent and curvature priors estimated directly from the Gaussian field. Specifically they introduce curvature-guided initialization, truncated gradient updates on tangent planes, and shape-consistent regularization to maintain geometric stability during training. Experiments shows that GeoSplat achieved imrpoved rendering quality.

**Strengths:**

By formulating Gaussian primitives as samples on continuous surface manifolds, the proposed method introduces a principled geometric framework that goes beyond purely relying on photometric optimization.

The motivation and integration of curvature-guided initialization, and shape regularization, truncated gradient updates are more than acceptable.

The extension to a varifold-based formulation is also a strong point, as it generalizes the approach to handle non-smooth or noisy regions where local manifold assumptions may not hold.

**Weaknesses:**

While the idea of introducing curvature-guided priors is quite interesting, I’m not fully (but certainly partially) convinced it actually brings substantial benefits beyond enforcing smoothness after reading the paper. The major concerns includes followings:

**W1.** The method seems to have a possibility that bias the reconstruction toward overly smooth surfaces, which could suppress fine geometric details or high-curvature regions.
This concern is raised from the fact that the paper doesn’t include explicit geometry reconstruction metrics and only photometric ones like PSNR or SSIM are reported. It would be really helpful to convince benefits by showing geometry reconstruction performance on more diverse benchmarks such as **DTU** or **Mip-NeRF 360**, with quantitative metrics like Chamfer Distance or at least some visual ablations that show how the curvature priors affect geometry quality. Also, the experiments are limited to relatively clean indoor scenes (e.g., Replica, ICL-Office), which don’t stress the method’s generality.

**W2**. I'm concerned about the methods vulnerability to SfM initialization. If the initial SfM point cloud is sparse or noisy, the estimated geometric priors (especially the second-order curvatures which are notoriously sensitive to noise) might be unreliable. This could negatively impact all subsequent geometry-constrained steps, from initialization to densification, potentially leading to artifacts or incorrect surface regularization. The paper claims the estimation methods are noise-robust, but their effectiveness on poorly initialized SfM data is not demonstrated.

**Questions:**

I believe this paper raises important questions about geometric regularization in 3DGS and has significant potential. To better understand the method's contributions and limitations, I would like the authors to clearly address the following points:

**Q1.** Have you observed any cases where the curvature-guided regularization oversmooths fine geometric details or sharp edges? What is your perspective on this potential trade-off between smoothness and detail preservation, especially since the quantitative evaluation (W1) focuses on photometric metrics rather than geometric accuracy?

**Q2.** Following W1, have you evaluated GeoSplat on benchmarks with available ground-truth geometry (like DTU, ScanNet) or more complex, unbounded 360-degree scenes (like Mip-NeRF 360)? Providing quantitative geometric metrics (e.g., Chamfer Distance) on such datasets would be highly valuable to substantiate the claims of improved geometry beyond the indoor scenes presented.

**Q3.** Could you provide a more detailed breakdown of the computational overhead? Specifically, what is the additional training time or memory cost incurred by the dynamic geometric estimations and the geometry-constrained optimization steps compared to the vanilla 3DGS baseline?

**Q4.** How sensitive is GeoSplat to the quality of the initial SfM point cloud? Since the framework relies heavily on geometric priors (normals, curvatures) estimated from these points, how does performance degrade if the initial SfM data is sparse or noisy? This seems particularly critical for higher-order curvature estimates, which are often sensitive to noise.

---

> ### Author Response · Authors · 2025-12-01
>
> Dear Reviewer kXse,
>
> We thank you for your review.
>
> Question-1: Have you observed any cases where the curvature-guided regularization oversmooths fine geometric details or sharp edges…
>
> Answer: Over-regularization might indeed limit the model performance, so we introduced several hyper-parameters to control the strength of regularization during training. For example, the truncated gradient was still allowed to move in the normal direction to a certain degree, and the number of enriched initial Gaussian primitives was also properly configured. We also would like to emphasize that one did not need to spend much effort tuning these hyper-parameters --- the model performance was not sensitive to their specific choices. As shown in the case studies of our paper (e.g., Fig. 2 and Fig. 4), the visual rendering quality, especially in the sharp areas (e.g., edges of sofa and sides of computer screens), remains high. These improvements stem from our regularization-based method rather than being hindered by it. We also refer the reviewer to the varifold paper (Buet et al (2022)), where Fig. 2 (the dragon example) shows that the method accurately estimates highly sharp geometric structures, consistent with our findings in previously mentioned case studies. In other words, estimating high-curvature regions is precisely the strength of such methods, whereas flat regions have trivial curvature.
>
> Question-2: Following W1, have you evaluated GeoSplat on benchmarks … or more complex, unbounded 360-degree scenes (like Mip-NeRF 360)…
>
> Answer: On the one hand, our paper, as well as previous works under the topic of geometric regularization, mainly aimed at making use of the rich geometric information of 3D scenes to further improve Gaussian splatting models, considering that common spatial objects are mostly smooth (at least continuous) and bounded. There might indeed be problematic regions in a 3D scene that contain little or even no geometric priors. This situation means that one is not able to take `free lunch" from geometric regularization, and it also falls out of the research scope of this work. We also refer the reviewer to the varifold paper (Buet et al (2022)), where Fig. 2 (the dragon example) shows that the method accurately estimates highly sharp geometric structures, consistent with our findings in the visualized case studies (e.g., Fig. 2 and Fig. 4).
>
> From another perspective, we would like to emphasize that our experiment setup closely followed that of a very recent related work (Li et al. (ICCV-2025): GeoGaussian), adopting 12 3D scenes of 2 large classes of datasets. The extensive experiment results in Table 1 and Table 2, as well as many case studies, have clearly shown that our method significantly outperforms the baselines. Therefore, we believe the current experiment results are sufficient for verifying the effectiveness of our method in our research scope. It is also interesting how our method, as well as other works in this field, perform and how they can better perform in more general cases. We believe a future work might be more appropriate in this sense.
>
> Question-3:  Could you provide a more detailed breakdown of the computational overhead…
>
> Answer: The time cost of geometric estimation is negligible compared with the overall model training. For example, on the OFF2 dataset, the varifold-based method takes only about 40s on average per run. We perform the estimation five times during training, resulting in a total overhead of around 3m20s which is minimal compared with the full training time of 51m. For the memory, a common CPU server device of 100G is enough for our use. We will clarify these in the revised version.
>
> Question-4: How sensitive is GeoSplat to the quality of the initial SfM point cloud…
>
> Answer: Similar to how neural network parameters are initialized, the goal of initializing a Gaussian primitive is to guide it toward an optimal value, not to reach it immediately. A certain amount of initial bias is therefore acceptable. For example, the original Gaussian covariance is initialized isotropically, which is surely far from its ideal value. By instead introducing a non-isotropic initialization based on geometric priors, we achieve a noticeable performance improvement, as shown in Table 3 of our paper. As mentioned in our first answer to you, there are also hyper-parameters (e.g., the truncated gradient was still allowed to move in the normal direction to a certain degree) to control the strength of regularization during training, so the impact of a certain level of estimation bias is minor to the final performance.

---

### Official Review · Reviewer_t2YG · 2025-10-28

**Soundness:** 2
**Presentation:** 3
**Contribution:** 3
**Rating:** 4
**Confidence:** 4

**Summary:**

This work improves 3DGS optimization from the prior of geometric theory. A set of initialization, regularization, and densification strategies are proposed. Specifically, the Gaussian scales and rotations are better initialized using the neighbor initial points. The training gradients are clip to move mainly on the estimated surface. The Gaussian shapes are regularized to be like 2D circles. The densified Gaussians is sampled more on the tangent plane instead of pure random. All proposed components are verified by ablation study and the overall rendering quality is better than previous work.

**Strengths:**

The proposed components can benefit all optimization-based 3DGS methods. They all grounded by geometric theory and are intuitively helpful. Another good thing is that they only take little extra training time. The improvement with fewer training images is very significant, which makes 3DGS useful even with few observation.

**Weaknesses:**

1. The proposed initialization and regularizations seems more fit with the 2DGS instead of 3DGS. The 2DGS itself can also cover the proposed shape regularization. However, comparison and discussion with 2DGS is missing.
2. The ablation for the proposed Gaussian normal estimation in Sedc.3.2 is missing. As it needs tens of seconds and is the most time consuming from the proposed methods, I expect the gain is good.
3. The evaluation is on 12 synthetic datasets. I wonder the effectiveness of the proposed methods in the real-world datasets. There are many real-world datasets, which are standard for NeRF/3DGS series of works to evaluate like mipnerf360.

**Questions:**

It would be nice if the ablation can also show how each of the proposed component improve the baseline 3DGS. This information is helpful for future work to see the individual improvement from 3DGS so they may can adopt a subset of them based on their need. The current ablation is conducted by removing one component out while keep the other new components activated.

---

> ### Author Response · Authors · 2025-12-01
>
> Dear Reviewer t2YG,
>
> We thank you for your review.
>
> Question: It would be nice if the ablation can also show how each of the proposed component improve the baseline 3DGS…
>
> Answer: To assess the usefulness of each component in our method, we have performed extensive ablation studies, as shown in Table 3—covering 6 × 6 × 2 settings in addition to our main experiments. While it is always possible to explore even more ablations, we believe the current results are already comprehensive and sufficient for this purpose.
>
> For your other comments, please refer to our first answers to Reviewer H9nz and Reviewer BHxw.

---

### Official Review · Reviewer_BHxw · 2025-10-29

**Soundness:** 2
**Presentation:** 2
**Contribution:** 3
**Rating:** 4
**Confidence:** 4

**Summary:**

This paper proposes GeoSplat, a geometry-constrained optimization framework for 3D Gaussian Splatting (3DGS). Unlike prior works that use low-order geometric priors like normals, this paper's core idea is to leverage second-order geometric quantities, specifically principal curvatures. The authors state this high-order prior is systematically applied to the entire 3DGS training pipeline: (1) curvature-guided covariance initialization , (2) curvature-guided primitive upsampling , (3) shape-constrained optimization losses, and (4) curvature-regularized densification. To estimate these geometric quantities, the paper proposes two complex, noise-robust methods: one based on a local manifold assumption and another based on Varifold theory. Experiments on the Replica and ICL datasets show performance gains over baselines like 3DGS and GeoGaussian

**Strengths:**

1.The core idea of using second-order geometry (curvature) to guide the anisotropy of Gaussian ellipsoids is novel. Linking Gaussian scale s inversely to curvature is a strong and intuitive prior .
2.The contribution is systematic. The geometric prior is integrated into all three core stages of 3DGS—initialization, optimization, and densification—rather than being just an isolated loss term.
3.The low-resource experiments in Figure 1 are convincing. The results show that GeoSplat's performance degrades much slower than baselines as training views are reduced , validating that the explicit geometric prior provides robustness.
4.Table 3 provides a complete ablation study, demonstrating that each proposed component, such as covariance warm-up and shape regularization, contributes to the final performance.

**Weaknesses:**

1.Severely Limited Experimental Scope: This is the paper's primary weakness. All experiments are limited to only two indoor datasets, Replica and ICL . For a method claiming to be a general 3DGS framework, this is insufficient.
a.Missing Object-Level Datasets: The method is not evaluated on standard NeRF Blender datasets, such as lego and mic, which are standard for testing fine geometric details.
b.Missing Outdoor 360° Datasets: Critically, the paper completely omits evaluations on unbounded outdoor scenes like Mip-NeRF 360 or Tanks & Temples. A core point of the paper is reducing "floaters" and artifacts . Outdoor 360° scenes are the standard and most difficult testbed for this exact problem. Only showing improvements on indoor ceilings in Figure 2 is not convincing. This omission raises doubts about the method's robustness. Does the heavy reliance on "surface" manifold assumptions  cause the method to fail in unbounded scenes with non-surface elements like the sky or distant backgrounds?
c.Ignoring Non-Surface Objects: The method's assumptions are biased towards solid surfaces. It is unclear how it would perform on volumetric or transparent objects (e.g., hair, smoke), where it might introduce incorrect geometric priors.
2. Section 3.2 on Geometric Estimations is the technical core of the paper. However, the main text provides no intuition or high-level explanation. It only states theorems like Thm. 3.2 and complex final equations like Eq. 12, deferring all derivations to the appendix. This is poor presentation and makes it impossible for a reviewer to assess the method's technical contribution from the main paper.

**Questions:**

1.Why did the authors not experiment on object-level (Blender) and outdoor 360° (Mip-NeRF 360 / Tanks & Temples) datasets? Given the paper's claims about fixing floaters , omitting the 360° benchmark is a major oversight.
2.What is the computational overhead of the dynamic geometry estimation? The paper claims it is efficient, citing tens of seconds for million-level Gaussians. How many times must this be run during training? What is the total training time increase compared to 3DGS and GeoGaussian?
3.In Tables 1 & 2, the Manifold-based and Varifold-based methods perform very similarly . Given the added complexity of the Varifold method in Eq. 12 , is it necessary? Are there cases where it significantly outperforms the manifold approach?

---

> ### Author Response · Authors · 2025-12-01
>
> Dear Reviewer BHxw,
>
> We thank you for your review.
>
> Question-1: Why did the authors not experiment on object-level (Blender) and outdoor 360° (Mip-NeRF 360 / Tanks & Temples) datasets?
>
> Answer: On the one hand, our paper, as well as previous works under the topic of geometric regularization, mainly aimed at making use of the rich geometric information of 3D scenes to further improve Gaussian splatting models, considering that common spatial objects are mostly smooth (at least continuous) and bounded. There might indeed be problematic regions (e.g., transparent) in a 3D scene that contain little or even no geometric priors. This situation means that one is not able to take `free lunch" from geometric regularization, and it also falls out of the research scope of this work.
>
> From another perspective, we also would like to emphasize that our experiment setup closely followed that of a very recent related work (Li et al. (ICCV-2025): GeoGaussian), adopting 12 3D scenes of 2 large classes of datasets. The extensive experiment results in Table 1 and Table 2, as well as many case studies, have clearly shown that our method significantly outperforms the baselines. Therefore, we believe the current experiment results are sufficient for verifying the effectiveness of our method in our research scope. It is also interesting how our method, as well as other works in this field, perform and how they can better perform in more general cases. We believe a future work might be more appropriate in this sense.
>
> Question-2: What is the computational overhead of the dynamic geometry estimation?
>
> Answer: The time cost of geometric estimation is negligible compared with the overall model training. For example, on the OFF2 dataset, the varifold-based method takes only about 40s on average per run. We perform the estimation five times during training, resulting in a total overhead of around 3m20s which is minimal compared with the full training time of 51m. We will clarify this in the revised version.
>
> Question-3: In Tables 1 & 2, the Manifold-based and Varifold-based methods perform very similarly. Given the added complexity of the Varifold method in Eq. 12 , is it necessary?
>
> Answer: Manifolds and varifolds are two major geometric formalisms for characterizing geometric objects, although manifolds are more commonly used in the literature. As discussed in the paper, prior geometric Gaussian splatting methods mainly rely on primitive techniques (e.g., using local PCA for geometric estimation). In this sense, a key contribution of this work is a thorough investigation into how to make such estimation more geometry-driven and scalable. The exploration of varifolds is included to make the study more comprehensive—covering different geometric assumptions, since varifolds offer more flexible and realistic modeling of geometric objects—and may be valuable for future research beyond its effect on model performance.

---

### Official Review · Reviewer_H9nz · 2025-10-30

**Soundness:** 3
**Presentation:** 3
**Contribution:** 2
**Rating:** 4
**Confidence:** 4

**Summary:**

This paper presents GeoSplat, a geometry-constrained optimization framework for Gaussian Splatting that explicitly incorporates higher-order geometric priors—normals and curvatures—throughout the full training pipeline based on the key ideas below.

- Curvature-guided initialization: Curvatures determine Gaussian covariance scales and orientations, producing better surface coverage for initialization.
- Shape-constrained optimization: Truncated gradient updates and curvature-aware regularization reduce floating or needle-like artifacts.
- Curvature-regularized densification: Curvature-weighted split/clone rules prevent outlier Gaussians in high-curvature areas.
- Noise-robust estimation: Proposed two complementary estimators (manifold-based Laplacian-geometry approach and a varifold-based geometric-measure-theory method) that supply dynamic and noise-robust priors during training.

Experiments on Replica and ICL datasets how consistent PSNR/SSIM/LPIPS gains and clear visual improvements, particularly under sparse-view (low-resource) conditions.

**Strengths:**

- Introduces a comprehensive curvature (higher-order geometry) use in Gaussian Splatting
- Formal derivations (Propositions 3.1 & Appendix D, E) connect manifold differential geometry to practical optimization steps.
- Consistent PSNR/SSIM/LPIPS gains and qualitative improvements, especially fewer floating Gaussians on sparse-view settings.

**Weaknesses:**

1. Additional comparison required: Recent works[1][2] curve-oriented Gaussian components are second-order primitives for Gaussian Splatting. A direct quantitative or qualitative comparison with these approaches is required to strengthen the novelty claim.
2. Accessibility of implementation: Heavy geometric derivations may limit reproducibility; a clearer pseudocode summary would help.
3. Runtime overhead: estimation frequency and cost are only qualitatively discussed (“tens of seconds per million Gaussians”)—quantitative runtime table would clarify trade-off.
4. Ablation coverage: while each module is tested, inter-dependencies (e.g., curvature estimator w/o truncated gradient) are not fully explored.

[1] Zhang, Ziyu, et al. "Quadratic Gaussian Splatting: High Quality Surface Reconstruction with Second-order Geometric Primitives." *Proceedings of the IEEE/CVF International Conference on Computer Vision*. 2025.

[2] Gao, Zhirui, et al. "Curve-Aware Gaussian Splatting for 3D Parametric Curve Reconstruction." *Proceedings of the IEEE/CVF International Conference on Computer Vision*. 2025.

**Questions:**

1. Recent works[1][2] also exploit higher-order geometric cues. How does GeoSplat differ from these in terms of formulation or estimation stability? A discussion or comparison table would help clarify the true extent of novelty.
2. Would the proposed method be generalized to outdoor scenes or textureless surfaces?
3. Additional ablation studies on hyperparameters (e.g., nearest neighbor $k$, thresholds $\xi_{\min}, \xi_{\max}$) would support the design choice of the proposed work.
4. Could curvature-guided initialization introduce bias when curvature estimates are noisy early in training?

[1] Zhang, Ziyu, et al. "Quadratic Gaussian Splatting: High Quality Surface Reconstruction with Second-order Geometric Primitives." *Proceedings of the IEEE/CVF International Conference on Computer Vision*. 2025.

[2] Gao, Zhirui, et al. "Curve-Aware Gaussian Splatting for 3D Parametric Curve Reconstruction." *Proceedings of the IEEE/CVF International Conference on Computer Vision*. 2025.

---

> ### Author Response · Authors · 2025-12-01
>
> Dear Reviewer H9nz,
>
> We thank you for your review.
>
> Question-1: Additional comparison required: Recent works[1][2]…. A direct quantitative or qualitative comparison with these approaches is required to strengthen the novelty claim.
>
> Answer: Both papers are largely different from our work in motivation and problem setup. The first work [1] presented a new type of Gaussian primitive: quadratic Gaussian, whose impact domain is defined over a 2D curved region (a generalized version of 2DGS). By contrast, common Gaussian primitives are defined on the entire 3D space. Instead of the local geometry of a single primitive, our work mainly studies the global geometry formed by all Gaussian primitives, and how to incorporate it into improving model optimization. From another perspective, our framework is also applicable to their new type of Gaussian primitives. For the second work [2], they mainly studied how to reconstruct 3D parametric curves in an end-to-end fashion, since previous models that have two stages suffered from error propagation. This work is even distinct from our work in the problem setting --- They studied on curve reconstruction, while we focused on scene rendering
>
> Question-2: Would the proposed method be generalized to outdoor scenes or textureless surfaces?
>
> Answer: Our paper, as well as previous works under the topic of geometric regularization, mainly aimed at making use of the rich geometric information of 3D scenes to further improve Gaussian splatting models, considering that common spatial objects are mostly smooth (at least continuous). There might indeed be regions in a 3D scene that contain little or even no geometric priors. This situation means that one is not able to take "free lunch" from geometric regularization, and it also falls out of the research scope of this work. It is also interesting how this problem can be addressed, and we believe future work might be more appropriate in this sense.
>
> Question-3: Additional ablation studies on hyperparameters… would support the design choice of the proposed work.
>
> Answer: To assess the usefulness of each component in our method, we have performed extensive ablation studies, as shown in Table 3—covering 6 × 6 × 2 settings in addition to our main experiments. While it is always possible to explore even more ablations, we believe the current results are already comprehensive and sufficient for this purpose.
>
> Question-4: Could curvature-guided initialization introduce bias when curvature estimates are noisy early in training?
>
> Answer: Similar to how neural network parameters are initialized, the goal of initializing a Gaussian primitive is to guide it toward an optimal value, not to reach it immediately. A certain amount of initial bias is therefore acceptable. In fact, the original Gaussian covariance is initialized isotropically, which is already far from its ideal value. By instead introducing a non-isotropic initialization based on geometric priors, we achieve a noticeable performance improvement, as shown in Table 3 of our paper.

---

### Comment · Area_Chair_bU2R · 2025-11-25
**Please engage with the reviewers during the rebuttal/discussion phase (if the authors plan to)**

Dear authors,

It has been over one week since the review comments were released.

If you plan to join the rebuttal, please engage sooner rather than later, so that the reviewers can have time engage as well.

Thanks,
AC

---

### Meta-Review · Area_Chair_a1Le · 2026-01-03

**Summary:**

While the paper introduces a novel geometry-constrained optimization framework for Gaussian Splatting with technically sound ideas, the consensus is to reject the submission in its current form. The decision is primarily based on the following critical shortcomings:

- Severely Limited and Insufficient Experimental Validation: The evaluation is confined to two indoor datasets, omitting critical benchmarks like object-level (NeRF Blender) and, most importantly, challenging unbounded outdoor 360° scenes (e.g., Mip-NeRF 360). This fails to substantiate the paper's claims of general applicability and robustness in addressing floaters—a problem most prevalent and critical in the very benchmarks it ignores. The omission raises significant doubts about the method's performance in real-world scenarios.

- Lack of Technical Accessibility and Clarity: The presentation of the core geometric estimation module (Sec. 3.2) is opaque. It relies heavily on theorem statements and complex equations in the main text without providing an intuitive explanation, making it difficult to assess the technical contribution. This hinders reproducibility and understanding.

- Incomplete Ablation and Analysis: Critical analyses are missing, including: the computational overhead of the proposed estimators on total training time; a justification for the added complexity of the varifold-based method given its similar performance to the manifold-based one; and the impact of the most time-consuming component (normal estimation). Furthermore, a necessary discussion and comparison with relevant works like 2DGS is absent.

Based on the overall information, I  rate the work below the threshold of acceptance.

**Reviewer Concerns:**

Some of concerns are not addressed sufficiently:
1. No solid extra experiments(NeRF Blender/ Mip-NeRF360 and etc.) are provided.
2. No sufficient dicussion about the relationships between this work and 2dgs.
3. Good geometry does not inherently guarantee high rendering quality, a point this paper completely overlooks in its discussion! Since 2DGS has previously explored this issue in depth, the lack of such discussion here leads me to conclude that the authors lack a clear understanding of the relationship between geometry and rendering quality.

**Reviewer Scores:**

I believe no reviewer would change their score for reasons mentioned above.

---

### Decision · Program_Chairs · 2026-01-26

Reject